# SELF-EVOLVING NEURAL RADIANCE FIELDS

## ABSTRACT

Recently, neural radiance field (NeRF) has shown remarkable performance in novel view synthesis and 3D reconstruction. However, it still requires abundant high-quality images, limiting its applicability in real-world scenarios. To overcome this limitation, recent works have focused on training NeRF only with sparse viewpoints by giving additional regularizations, often called few-shot NeRF. We observe that due to the under-constrained nature of the task, solely using additional regularization is not enough to prevent the model from overfitting to sparse viewpoints. In this paper, we propose a novel framework, dubbed Self-Evolving Neural Radiance Fields (`SE-NeRF`), that applies a self-training framework to NeRF to address these problems. We formulate few-shot NeRF into a teacher-student framework to guide the network to learn a more robust representation of the scene by training the student with additional pseudo labels generated from the teacher. By distilling ray-level pseudo labels using distinct distillation schemes for reliable and unreliable rays obtained with our novel reliability estimation method, we enable NeRF to learn a more accurate and robust geometry of the 3D scene. We show and evaluate that applying our self-training framework to existing models improves the quality of the rendered images and achieves state-of-the-art performance in multiple settings.

## 1 INTRODUCTION

Novel view synthesis that aims to generate novel views of a 3D scene from given images is one of the essential tasks in computer vision fields. Recently, neural radiance field (NeRF) (Mildenhall et al., 2021) has shown remarkable performance for this task, modeling highly detailed 3D geometry and specular effects solely from given image information. However, the requirement of abundant high-quality images with accurate poses restricts its application to real-world scenarios, as reducing the input views causes NeRF to produce broken geometry and undergo severe performance degradation.

Numerous works (Kim et al., 2022; Jain et al., 2021; Wang et al., 2023; Niemeyer et al., 2022; Yu et al., 2021) tried to address this problem, known as few-shot NeRF, whose aim is to robustly optimize NeRF in scenarios where only a few and sparse input images are given. To compensate for the few-shot NeRF's under-constrained nature, they either utilize the prior knowledge of a pre-trained model (Jain et al., 2021; Yu et al., 2021) such as CLIP (Radford et al., 2021) or 2D CNN (Yu et al., 2021) or introduce an additional regularization (Niemeyer et al., 2022; Kim et al., 2022; Kwak et al., 2023), showing compelling results. However, these works show limited success in addressing the fundamental issue of overfitting as NeRF tends to *memorize* the input known viewpoints instead of *understanding* the geometry of the scene.

In our toy experiment, this behavior is clearly shown in Figure 1, where existing methods (even with regularization (Fridovich-Keil et al., 2023; Niemeyer et al., 2022; Kim et al., 2022)) trained with 3-views show a noticeable drop in PSNR even with slight changes of viewpoints. Utilizing additional ground truth data for viewpoints that were unknown to the few-shot setting, we compare the rendered images from few-shot NeRF with the ground truth images and verify that there are accurately modeled regions even in *unknown* viewpoints that are *far* from known ones.

This indicates that if we can accurately identify reliable regions, the rendered regions can be utilized as additional data achieved with no extra cost. Based on these facts, we formulate the few-shot NeRF task into the self-training framework by considering the rendered images as pseudo labels and training a new NeRF network with confident pseudo labels as additional data.

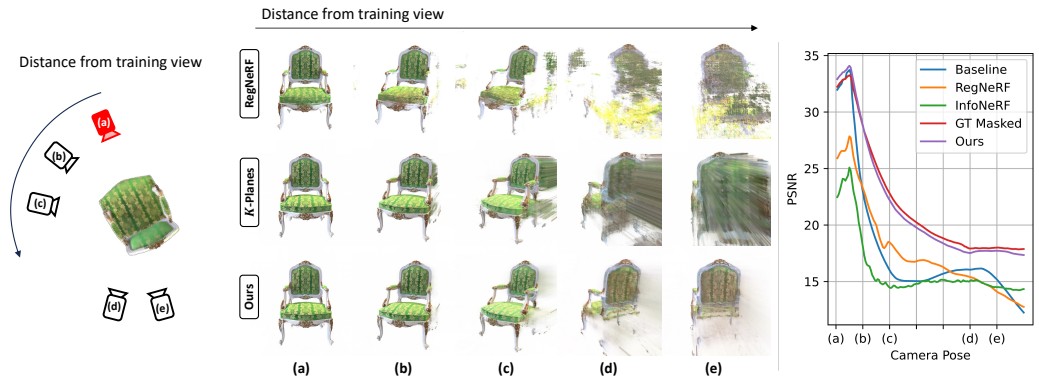

Figure 1: **Toy experiment to verify the robustness of models trained with sparse views. (Left)** The red camera (a) indicates the camera position used for training and cameras from (b-e) are used to verify the robustness of models when the novel viewpoint gets further from the known viewpoint. **(Middle)** For each viewpoint (a-e), we visualize the rendered images by RegNeRF (Niemeyer et al., 2022), baseline ($K$-Planes (Fridovich-Keil et al., 2023)), and SE-NeRF from top to bottom rows. **(Right)** Starting from viewpoint (a), we show the PSNR graph of the rendered images as the viewpoint moves gradually from (a-e). Existing models show extreme PSNR drops, even with slight movements.

Expanding upon this idea, we introduce a novel framework, dubbed Self-Evolving Neural Radiance Fields (SE-NeRF), which enables a more robust training of few-shot NeRF in a self-supervised manner. We train the few-shot NeRF under an iterative teacher-student framework, in which pseudo labels for geometry and appearance generated by the teacher NeRF are distilled to the student NeRF, and the trained student serves as the teacher network in the next iteration for progressive improvement. To estimate the reliability of the pseudo labels, we utilize the semantic features of a pre-trained 2D CNN to measure the consistency of the pseudo labels within multiple viewpoints. We also apply distinct distillation schemes for reliable and unreliable rays, in which reliable ray labels are directly distilled to the student, while unreliable rays undergo a regularization process to distill more robust geometry.

Our experimental results show that our framework successfully guides existing NeRF models towards a more robust geometry of the 3D scene in the few-shot NeRF setting without using any external 3D priors or generative models (Xu et al., 2022). Also, we show the versatility of our framework, which can be applied to any existing models without changing their structure. We evaluate our approach on synthetic and real-life datasets, achieving state-of-the-art results in multiple settings.

## 2 RELATED WORK

**Neural radiance fields (NeRF).** Synthesizing images from novel views of a 3D scene given multi-view images is a long-standing goal of computer vision. Recently, neural radiance fields (NeRF) (Mildenhall et al., 2021) has achieved great success by optimizing a single MLP that learns to estimate the radiance of the queried coordinates. The MLP learns the density $\sigma \in \mathbb{R}$ and color $\mathbf{c} \in \mathbb{R}^3$ of continuous coordinates $\mathbf{x} \in \mathbb{R}^3$, and is further utilized to explicitly render the volume of the scene using ray marching (Kajiya & Von Herzen, 1984). Due to it's impressive performance in modeling the 3D scene, various follow-ups (Deng et al., 2022; Jain et al., 2021; Kim et al., 2022; Fridovich-Keil et al., 2023; Niemeyer et al., 2022; Wang et al., 2023; Roessle et al., 2022; Yang et al., 2023) adopted NeRF as their baseline model to solve various 3D tasks.

**Few-shot NeRF.** Although capable of successfully modeling 3D scenes, NeRF requires abundant high-quality images with accurate poses, making it hard to apply in real-world scenarios. Several methods have paved the way to circumvent these issues by showing that the network can be successfully trained even when the input images are limited. One approach addresses the problem using prior knowledge from pre-trained local CNNs (Yu et al., 2021; Chibane et al., 2021; Kwak et al., 2023). PixelNeRF (Yu et al., 2021), for instance, employs a NeRF conditioned with features extracted by a pre-trained encoder. Another line of research introduces a geometric or depth-based regularization to the network (Jain et al., 2021; Kim et al., 2022; Niemeyer et al., 2022; Deng et al.,

2022; Wang et al., 2023; Roessle et al., 2022). DietNeRF (Jain et al., 2021) proposes an auxiliary semantic consistency loss to encourage realistic renderings at novel poses. RegNeRF (Niemeyer et al., 2022) regularizes the geometry and appearance of patches rendered from unobserved viewpoints. DS-NeRF (Deng et al., 2022) introduces additional depth supervision from sparse point clouds obtained in the COLMAP (Schonberger & Frahm, 2016) process.

**Self-training.**  Self-training is one of the earliest semi-supervised learning methods (Fralick, 1967; Scudder, 1965) mainly used in settings where obtaining sufficient labels is expensive (e.g., Instance segmentation). Self-training exploits the *unlabeled* data by pseudo labeling with a teacher model, which is then *combined* with the labeled data and used in the student training process. Noisy student (Xie et al., 2020) succeeds in continually training a better student by initializing a larger model as the student, and injecting noise into the data and network. Meta pseudo labels (Pham et al., 2021), on the other hand, optimizes the teacher model by evaluating the student's performance on labeled data, guiding the teacher to generate better pseudo labels. We bring self-training to NeRFs by formulating the few-shot NeRF task as a semi-supervised learning task. Our approach can be seen as an analogous method of noisy student (Xie et al., 2020) that exploits NeRF as the teacher and student model, with teacher-generated *unknown* views as the *unlabeled* data.

## 3 PRELIMINARIES AND MOTIVATION

### 3.1 PRELIMINARIES

Given a set of training images $\mathcal{S} = \{I_i \mid i \in \{1, \ldots, N\}\}$, NeRF (Mildenhall et al., 2021) represents the scene as a continuous function $f(\cdot; \theta)$, a neural network with parameters $\theta$. The network renders images by querying the 3D points $\mathbf{x} \in \mathbb{R}^3$ and view direction $\mathbf{d} \in \mathbb{R}^2$ transformed by a positional encoding $\gamma(\cdot)$ to output a color value $\mathbf{c} \in \mathbb{R}^3$ and a density value $\sigma \in \mathbb{R}$ such that $\{\mathbf{c}, \sigma\} = f(\gamma(\mathbf{x}), \gamma(\mathbf{d}); \theta)$. The positional encoding transforms the inputs into Fourier features (Tancik et al., 2020) that facilitate learning high-frequency details. Given a ray parameterized as $\mathbf{r}(t) = \mathbf{o} + t\mathbf{d}$, starting from camera center $\mathbf{o}$ along the direction $\mathbf{d}$, the expected color value $C(\mathbf{r}; \theta)$ along the ray $\mathbf{r}(t)$ from $t_n$ to $t_f$ is rendered as follows:

$$C(\mathbf{r}; \theta) = \int_{t_n}^{t_f} T(t)\sigma(\mathbf{r}(t); \theta)\mathbf{c}(\mathbf{r}(t), \mathbf{d}; \theta)dt, \quad T(t) = \exp\left(-\int_{t_n}^{t} \sigma(\mathbf{r}(s); \theta)ds\right), \quad (1)$$

where $T(t)$ denotes the accumulated transmittance along the ray from $t_n$ to $t$.

To optimize the network $f(\cdot; \theta)$, the photometric loss $\mathcal{L}_{\text{photo}}(\theta)$ enforces the rendered pixel color value $C(\mathbf{r}; \theta)$ to be consistent with the ground-truth pixel color value $C^{\text{gt}}(\mathbf{r})$:

$$\mathcal{L}_{\text{photo}}(\theta) = \sum_{\mathbf{r} \in \mathcal{R}} \|C^{\text{gt}}(\mathbf{r}) - C(\mathbf{r}; \theta)\|_2^2, \quad (2)$$

where $\mathcal{R}$ is the set of rays corresponding to each pixel in the image set $\mathcal{S}$.

### 3.2 MOTIVATION

Despite its impressive performance, NeRF has the critical drawback of requiring large amounts of posed input images $\mathcal{S}$ for robust scene reconstruction. Naïvely optimizing NeRF in a few-shot setting (e.g., $|\mathcal{S}| < 10$) results in NeRF producing erroneous artifacts and undergoing major breakdowns in the geometry due to the task's under-constrained nature (Niemeyer et al., 2022; Kim et al., 2022).

A closer look reveals important details regarding the nature of the few-shot NeRF optimization. As described by the PSNR graph in Figure 1, all existing methods show a noticeable PSNR drop even with slight viewpoint changes, which indicates the tendency of NeRF to *memorize* the given input views. Such a tendency results in broken geometry that looks perfect in known viewpoints but progressively degenerates as the rendering view gets further away from known views. Although training with additional data directly solves this problem, obtaining high-quality images with accurate poses is extremely expensive. Instead, we notice that although images (rendered from NeRF trained with only sparse viewpoints) contain artifacts and erroneous geometry, there are reliable pixels of the image that are close to the corresponding ground truth pixels, which can be used as additional data.

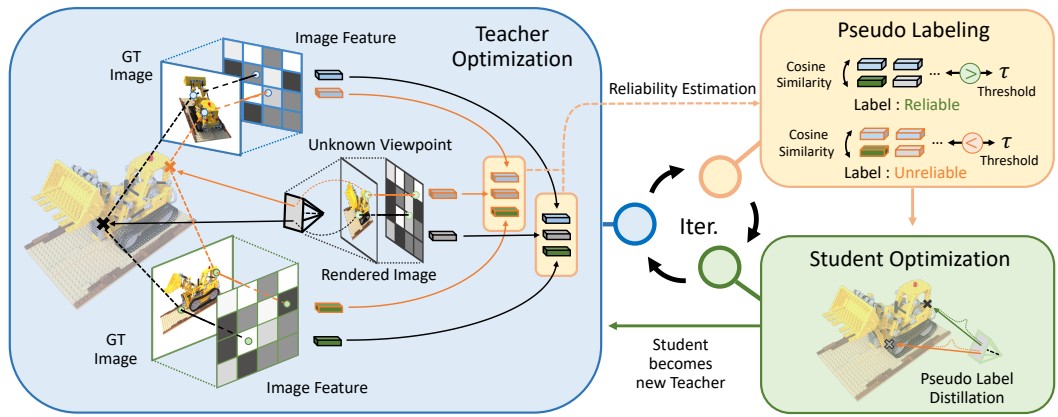

Figure 2: **Illustration of our overall framework for applying self-training to NeRF.** `SE-NeRF` utilizes the self-training framework to distill the knowledge of learned appearance and 3D geometry from teacher to student. The process is done iteratively as the student becomes the new teacher.

To check the feasibility that using reliable pixels from the rendered images as additional data can help prevent NeRF from overfitting, we conduct an experiment of first optimizing NeRF under the identical few-shot setting. After training a teacher NeRF with three images, we train a new student NeRF with the extended set of images $\mathcal{S} \cup \mathcal{S}^+$ where $\mathcal{S}^+$ is the set of rendered images. To train with only the reliable pixels of $\mathcal{S}^+$, we define a binary reliability mask $M(\mathbf{r})$, which masks out pixels where the difference between the rendered color value $C(\mathbf{r}; \theta^{\mathbb{T}})$ and its ground truth color value $C^{\text{gt}}(\mathbf{r})$ is above a predetermined threshold. Training the student NeRF network to follow the reliably rendered color values $\{C(\mathbf{r}; \theta^{\mathbb{T}}) \,|\, M(\mathbf{r}) = 1\}$ of the teacher can be seen as a weak distillation from the teacher to the student. The new student NeRF is trained with the following loss function:

$$\mathcal{L}_{\text{photo}}(\theta) + \lambda \sum_{\mathbf{r} \in \mathcal{R}^+} M(\mathbf{r}) \| C(\mathbf{r}; \theta^{\mathbb{T}}) - C(\mathbf{r}; \theta) \|_2^2, \tag{3}$$

where $\mathcal{R}^+$ is a set of rays corresponding to each pixel in the rendered image set $\mathcal{S}^+$, and $\lambda$ denotes the weight parameter.

The result of this experiment, described in "GT Masked" of the PSNR graph in Figure 1, shows that the student trained with $K$-Planes (Fridovich-Keil et al., 2023) as the baseline, displays staggering improvement in performance, with *unknown* viewpoints showing higher PSNR values and their rendered geometry remaining highly robust and coherent. This leads us to deduce that a major cause of few-shot NeRF geometry breakdown is its tendency to *memorize* the given sparse viewpoints and that selected distillation of additional reliable rays is crucial to enhance the robustness and coherence of 3D geometry. Based on this observation, our concern now moves on to how to estimate the reliability mask $M$ for the rendered novel images of $\mathcal{S}^+$ to develop a better few-shot NeRF model.

## 4 METHOD

### 4.1 TEACHER-STUDENT FRAMEWORK

**Teacher network optimization.** A teacher network is trained naïvely by optimizing the standard NeRF photometric loss where the number of known viewpoints is $|\mathcal{S}| < 10$. During this process, NeRF recovers accurate geometry for certain regions and inaccurate, broken geometry in other regions. The parameters of teacher network $\theta^{\mathbb{T}}$ is optimized as the following equation:

$$\theta^{\mathbb{T}} = \underset{\theta}{\arg\min} \, \mathcal{L}_{\text{photo}}(\theta). \tag{4}$$

**Pseudo labeling with teacher network.** By evaluating the optimized teacher NeRF representation $\theta_{\mathbb{T}}$, we can generate per-ray pseudo labels $\{C(\mathbf{r}; \theta^{\mathbb{T}}) \,|\, \mathbf{r} \in \mathcal{R}^+\}$ from the rendered images $\mathcal{S}^+$ from unknown viewpoints. To accurately identify and distill the reliable regions of $\mathcal{S}^+$ to the student model, we assess the reliability of every pseudo label in $\mathcal{R}^+$ to acquire a reliability mask $M(\mathbf{r})$ using a novel reliability estimation method we describe in detail in Section 4.2.

**Student network optimization.** The student network $\theta^{\mathbb{S}}$ is then trained with the extended training set of $\mathcal{S} \cup \mathcal{S}^+$, with the reliability mask $M$ taken into account. In addition to the photometric loss with the initial image set $\mathcal{S}$, the student network is also optimized with a distillation loss that encourages it to follow the robustly reconstructed parts of the teacher model in $\mathcal{S}^+$. In the distillation process, the estimated reliability mask $M$ determines how each ray should be distilled, a process which we explain further in Section 4.3. In summary, student network $\theta^{\mathbb{S}}$ is optimized by the following equation:

$$\theta^{\mathbb{S}} = \underset{\theta}{\arg\min} \left\{ \mathcal{L}_{\text{photo}}(\theta) + \lambda \sum_{\mathbf{r} \in \mathcal{R}^+} M(\mathbf{r}) \| C(\mathbf{r}; \theta^{\mathbb{T}}) - C(\mathbf{r}; \theta) \|_2^2 \right\}, \tag{5}$$

where $C(\mathbf{r}; \theta^{\mathbb{T}})$ and $C(\mathbf{r}; \theta)$ is the rendered color of the teacher and student model, respectively and $\lambda$ denotes the weight parameter.

**Iterative labeling and training.** After the student network is fully optimized, the trained student network becomes the teacher network of the next iteration for another distillation process to a newly initialized NeRF, as described in Figure 2. We achieve improvement of the NeRF's quality and robustness every iteration with the help of the continuously extended dataset.

## 4.2 RAY RELIABILITY ESTIMATION

To estimate the reliability of per-ray pseudo labels $\{C(\mathbf{r}; \theta^{\mathbb{T}}) | \ \mathbf{r} \in \mathcal{R}^+\}$ from the rendered images $\mathcal{S}^+$, we expand upon an important insight that if a ray has accurately recovered a surface location and this location is projected to multiple viewpoints, the semantics of the projected locations should be consistent except for occlusions between viewpoints. This idea has been used in previous works that formulate NeRF for refined surface reconstruction (Chibane et al., 2021), but our work is the first to leverage it for explicitly modeling ray reliability in a self-training setting.

The surface location recovered by a ray $\mathbf{r}$ corresponding to pixel $\mathbf{p}_i$ of the viewpoint $i$ can be projected to another viewpoint $j$ with the extrinsic matrix $R_{i \rightarrow j}$, intrinsic matrix $K$, and the estimated depth $D_i$ from viewpoint $i$ with the following projection equation:

$$\mathbf{p}_{i \rightarrow j} \sim K R_{i \rightarrow j} D_i(\mathbf{r}) K^{-1} \mathbf{p}_i. \tag{6}$$

Using the projection equation, we can make corresponding pixel pairs between viewpoint $i$ and $j$ such as $(\mathbf{p}_i, \mathbf{p}_j)$ where $\mathbf{p}_j = \mathbf{p}_{i \rightarrow j}$. Similarly, if we acquire pixel-level feature maps from viewpoint $i$ and $j$ using a pre-trained 2D CNN, we can make corresponding feature pairs as $(f_{\mathbf{p}}^i, f_{\mathbf{p}}^j)$. In our case, by projecting the feature vector of the corresponding pseudo label $\{C(\mathbf{r}; \theta^{\mathbb{T}}) | \ \mathbf{r} \in \mathcal{R}^+\}$ to all given input viewpoints, we can achieve $|\mathcal{S}|$ feature pairs for every pseudo label. To generate a reliability mask for each ray, if a ray has at least one feature pair whose similarity value is higher than the threshold value $\tau$, it indicates that the feature consistency of the ray's rendered geometry has been confirmed and classify such rays as reliable. Summarized in equation, the binary reliability mask $M(\mathbf{r})$ for the ray $\mathbf{r}$ rendered from viewpoint $i$ can be defined as follows:

$$M(\mathbf{r}) = \min \left\{ \sum_{j \in |\mathcal{S}|} \mathbb{1} \left[ \frac{\mathbf{f}_{\mathbf{p}}^i \cdot \mathbf{f}_{\mathbf{p}}^j}{\left\| \mathbf{f}_{\mathbf{p}}^i \right\| \left\| \mathbf{f}_{\mathbf{p}}^j \right\|} > \tau \right], 1 \right\}. \tag{7}$$

To prevent the unreliable rays from being misclassified as reliable, we must carefully choose the threshold $\tau$. Although using a fixed value for the $\tau$ is straightforward, we find that choosing the adequate value is extremely cumbersome as the similarity distribution for each scene varies greatly. Instead, we adopt the adaptive thresholding method, which chooses the threshold by calculating the $(1 - \alpha)^{th}$ percentile of the similarity distribution where $\alpha$ is a hyperparameter in the range $\alpha \in [0, 1]$. This enables the threshold $\tau$ to be dynamically adjusted to each scene, leading to a better classification of the reliable rays.

## 4.3 RELIABILITY-BASED DISTILLATION

To guide the student network to learn a more robust representation of the scene, we distill the label information from the teacher to the student with two distinct losses based on the ray's reliability. By remembering the rays evaluated in the teacher network and re-evaluating the same rays in the student network, the geometry and color information of reliable rays is directly distilled into the student network through distillation loss, while the rays classified as unreliable are regularized with nearby reliable rays for improved geometry before applying the distillation loss.

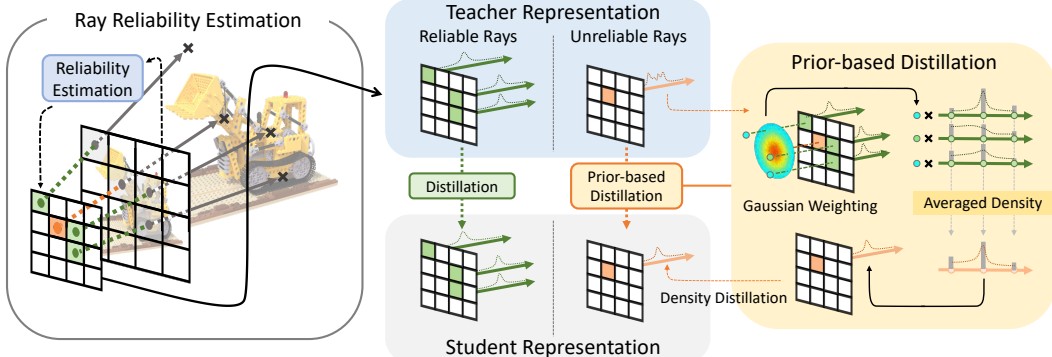

Figure 3: **Distillation of pseudo labels.** After estimating the reliability of the rays from unknown views, we apply distinct distillation schemes for reliable and unreliable rays. Reliable rays are directly distilled to the student while we aggregate the nearby reliable rays to regularize the unreliable rays.

**Reliable ray distillation.** Since we assume the reliable rays' appearance and geometry have been accurately predicted by the teacher network, we directly distill their rendered color so that the student network faithfully follows the outputs of the teacher for these reliable rays. With the teacher-generated per-ray pseudo labels $\{C(\mathbf{r}; \theta^{\mathbb{T}}) | \ \mathbf{r} \in \mathcal{R}^+\}$ from the rendered images $\mathcal{S}^+$ and the estimated reliability mask $M$, the appearance of a reliable ray is distilled by the reformulated photometric loss $\mathcal{L}_c^{\mathcal{R}}$:

$$\mathcal{L}_c^{\mathcal{R}}(\theta) = \sum_{\mathbf{r} \in \mathcal{R}^+} M(\mathbf{r}) \|C(\mathbf{r}; \theta^{\mathbb{T}}) - C(\mathbf{r}; \theta)\|_2^2. \tag{8}$$

In addition to the photometric loss $\mathcal{L}_c^{\mathcal{R}}$, we follow Deng et al. (2022); Roessle et al. (2022) of giving the depth-supervision together to NeRF. As the teacher network $\theta^{\mathbb{T}}$ also outputs the density $\sigma(\mathbf{r}; \theta^{\mathbb{T}})$ for each of the rays, we distill the density weights of the sampled points of the reliable rays to the student network. Within the same ray, we select an identical number of points randomly sampled from evenly spaced bins along the ray. This allows us to follow the advantages of injecting noise to the student as in Xie et al. (2020) as randomly sampling points from each bin induces each corresponding point to have slightly different positions, which acts as an additional noise to the student.

The density distillation is formulated by the geometry distillation loss $\mathcal{L}_g^{\mathcal{R}}$, which is L2 loss between accumulated density values of corresponding points within the teacher and student rays, with teacher rays' density values $\sigma^{\mathbb{T}}$ serving as the pseudo ground truth labels. Therefore, for reliable rays, our distillation loss along the camera ray $\mathbf{r}(t) = \mathbf{o} + t\mathbf{d}$ is defined as follows:

$$\mathcal{L}_g^{\mathcal{R}}(\theta) = \sum_{\mathbf{r} \in \mathcal{R}^+} \sum_{t,t' \in T} M(\mathbf{r}) \|\sigma(\mathbf{r}(t); \theta^{\mathbb{T}}) - \sigma(\mathbf{r}(t'); \theta)\|_2^2. \tag{9}$$

where $T$ refers to the evenly spaced bins from $t_n$ to $t_f$ along the ray, $t$ and $t'$ indicate randomly selected points from each bins.

**Unreliable ray distillation.** In traditional semi-supervised methods, unreliable labels are ignored to prevent the confirmation bias problem. Similarly, unreliable rays must not be directly distilled as they are assumed to have captured inaccurate geometry. However, stemming from the prior knowledge that depth changes smoothly above the surface, we propose a novel method for regularizing the unreliable rays with geometric priors of nearby reliable rays, dubbed prior-based distillation.

To distill the knowledge of nearby reliable rays, we calculate a weighted average of nearby reliable rays' density distribution and distill this density to the student. As described in Figure 3, we apply a Gaussian mask to unreliable ray r to calculate per-ray weights for nearby reliable rays. The intuition behind this design choice is straightforward: the closer a ray is to an unreliable ray, the more likely it is to be that the geometry of the two rays will be similar. Based on these facts, we apply the prior-based geometry distillation loss $\mathcal{L}_g^{\mathcal{P}}$, which is the L2 loss between the weighted-average density $\tilde{\sigma}(\mathbf{r}; \theta^{\mathbb{T}})$ and the student density outputs $\sigma(\mathbf{r}; \theta)$, is described in the following equation:

$$\mathcal{L}_g^{\mathcal{P}}(\theta) = \sum_{\mathbf{r} \in \mathcal{R}^+} \sum_{t,t' \in T} (1 - M(\mathbf{r})) \|\tilde{\sigma}(\mathbf{r}(t); \theta^{\mathbb{T}}) - \sigma(\mathbf{r}(t'); \theta)\|_2^2. \tag{10}$$

We apply the prior-based geometry distillation loss to the unreliable rays only when adjacent reliable rays exist. A more detailed explanation can be found in Appendix B.3

Table 1: **Quantitative comparison on NeRF Synthetic and LLFF.**

| Methods | NeRF Synthetic Extreme | | | | NeRF Synthetic | | | | LLFF | | | |
|---|---|---|---|---|---|---|---|---|---|---|---|---|
| | PSNR↑ | SSIM↑ | LPIPS↓ | Avg.↓ | PSNR↑ | SSIM↑ | LPIPS↓ | Avg.↓ | PSNR↑ | SSIM↑ | LPIPS↓ | Avg.↓ |
| NeRF | 14.85 | 0.73 | 0.32 | 0.27 | 19.38 | 0.82 | 0.17 | 0.20 | 17.50 | 0.50 | 0.47 | 0.40 |
| $K$-Planes | 15.45 | 0.73 | 0.28 | 0.28 | 17.99 | 0.82 | 0.18 | 0.21 | 15.77 | 0.44 | 0.46 | 0.41 |
| DietNeRF | 14.46 | 0.72 | 0.28 | 0.28 | 15.42 | 0.73 | 0.31 | 0.20 | 14.94 | 0.37 | 0.50 | 0.44 |
| InfoNeRF | 14.62 | 0.74 | 0.26 | 0.27 | 18.44 | 0.80 | 0.22 | 0.12 | 13.57 | 0.33 | 0.58 | 0.48 |
| RegNeRF | 13.73 | 0.70 | 0.30 | 0.30 | 13.71 | 0.79 | 0.35 | 0.21 | 19.08 | 0.59 | 0.34 | 0.15 |
| SE-NeRF (NeRF) | 17.41 (+2.56) | 0.78 (+0.05) | 0.21 (-0.11) | 0.22 (-0.05) | 20.53 (+1.15) | 0.84 (+0.02) | 0.16 (-0.01) | 0.19 (-0.01) | 18.10 (+0.60) | 0.54 (+0.04) | 0.45 (-0.02) | 0.38 (-0.02) |
| SE-NeRF ($K$-Planes) | 17.49 (+2.04) | 0.78 (+0.05) | 0.23 (-0.05) | 0.24 (-0.04) | 19.93 (+1.94) | 0.83 (+0.01) | 0.17 (-0.01) | 0.20 (-0.01) | 16.30 (+0.53) | 0.49 (+0.05) | 0.44 (-0.02) | 0.39 (-0.02) |

**Total distillation loss.** Finally, our entire distillation loss can be formulated as follows:

$$\theta^{\mathbb{S}} = \underset{\theta}{\arg\min} \left\{ \mathcal{L}_{\text{photo}}(\theta) + \lambda_c^{\mathcal{R}} \mathcal{L}_c^{\mathcal{R}}(\theta) + \lambda_g^{\mathcal{R}} \mathcal{L}_g^{\mathcal{R}}(\theta) + \lambda_g^{\mathcal{P}} \mathcal{L}_g^{\mathcal{P}}(\theta) \right\}, \qquad (11)$$

where $\lambda_c^{\mathcal{R}}$, $\lambda_g^{\mathcal{R}}$, and $\lambda_g^{\mathcal{P}}$ denotes the weight parameters.

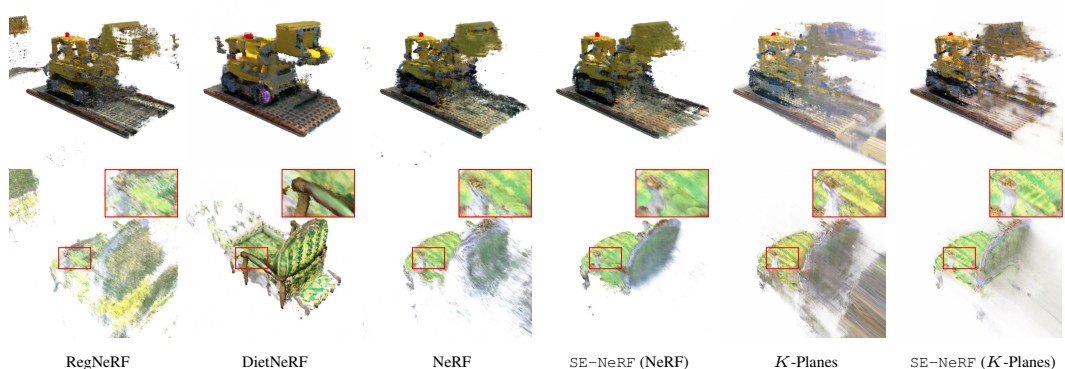

| RegNeRF | DietNeRF | NeRF | SE-NeRF (NeRF) | $K$-Planes | SE-NeRF ($K$-Planes) |

Figure 4: **Qualitative comparison on NeRF Synthetic Extreme.** The results show the rendered images from viewpoints far away from the seen views. A noticeable improvement over existing models regarding artifacts and distortion removal can be observed in SE-NeRF.

## 5 EXPERIMENTS

### 5.1 SETUPS

**Datasets and metrics.** We evaluate our methods on NeRF Synthetic (Mildenhall et al., 2021) and LLFF dataset (Mildenhall et al., 2019). For the NeRF Synthetic dataset, we randomly select 4 views in the train set and use 200 images in the test set for evaluation. For LLFF, we chose every 8-th image as the held-out test set and randomly select 3 views for training from the remaining images. In addition, we find that all existing NeRF models' performance on the NeRF Synthetic dataset is largely affected by the randomly selected views. To explore the robustness of our framework and existing methods, we introduce a novel evaluation protocol of training every method with an extreme 3-view setting (NeRF Synthetic Extreme) where all the views are selected from one side of the scene. The selected views can be found in Appendix C. We report PSNR, SSIM (Wang et al., 2004), LPIPS (Zhang et al., 2018) and geometric average (Barron et al., 2021) values for qualitative comparison.

**Implementation details.** Although any NeRF representation is viable, we adopt $K$-Planes (Fridovich-Keil et al., 2023) as our main baseline to leverage its memory and time efficiency. Also, we conduct experiments using our framework with NeRF (Mildenhall et al., 2021) and Instant-NGP[1] (Müller et al., 2022) to demonstrate the applicability of our framework. For our reliability estimation method, we use VGGNet (Simonyan & Zisserman, 2014), specifically VGG-19, and utilize the first 4 feature layers located before the pooling layers. We train $K$-Planes for 20 minutes on NeRF Synthetic and 60 minutes on LLFF using a single RTX 3090, and NeRF is trained for 90 minutes on NeRF Synthetic and 120 minutes on LLFF using 4 RTX 3090 GPUs for each iteration.

[1]For Instant-NGP, we train the model for 5 minutes on NeRF Synthetic Extreme.

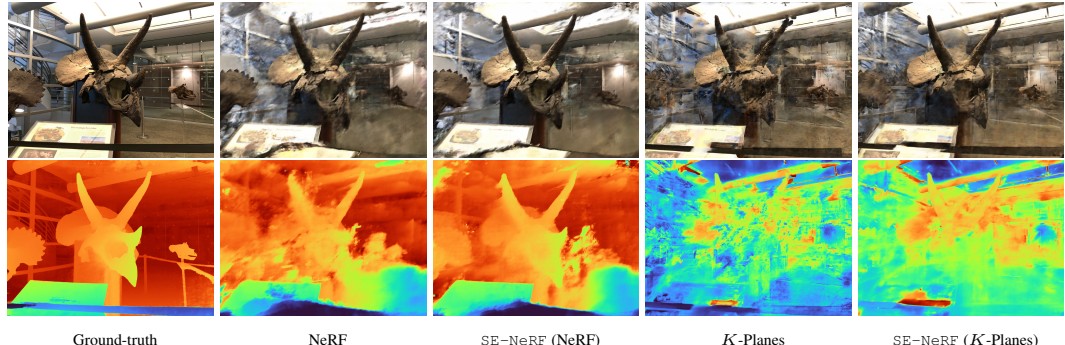

| Ground-truth | NeRF | SE-NeRF (NeRF) | $K$-Planes | SE-NeRF ($K$-Planes) |

Figure 5: **Qualitative improvement from baselines.**

**Hyper-parameters.** We set the adaptive threshold value at $\alpha = 0.15$ for the first iteration. To enable the network to benefit from more reliable rays for each subsequent iteration, we employ a curriculum labeling Cascante-Bonilla et al. (2021) approach that increases $\alpha$ by 0.05 every iteration. As images rendered from views near the initial inputs include more reliable regions, we progressively increase the range of where the pseudo labels should be generated. We start by selecting views that are inside the range of 10 degrees in terms of $\phi, \theta$ of the initial input and increase range after iterations. For the weights for our total distillation loss, we use $\lambda_c^{\mathcal{R}} = 1.0$, $\lambda_g^{\mathcal{R}} = 1.0$, and $\lambda_g^{\mathcal{P}} = 0.005$.

Table 2: **Quantitative comparison per-scene on NeRF Synthetic Extreme.**

| Methods | chair | drums | ficus | hotdog | lego | mater. | ship | mic |
|---------|-------|-------|-------|--------|------|--------|------|-----|
| NeRF | 15.08 | 11.98 | 17.16 | 13.83 | 16.31 | 17.31 | 10.84 | 16.29 |
| $K$-Planes | 15.61 | 13.23 | 18.29 | 12.45 | 14.67 | 16.30 | 13.35 | 19.74 |
| Instant-NGP | 17.66 | 12.75 | 18.44 | 13.67 | 13.17 | 16.83 | 13.82 | 19.05 |
| DietNeRF | 16.60 | 8.09 | 18.32 | 19.00 | 11.45 | 16.97 | 15.26 | 10.01 |
| InfoNeRF | 15.38 | 12.48 | 18.59 | 19.04 | 12.27 | 15.25 | 7.23 | 16.76 |
| RegNeRF | 15.92 | 12.09 | 14.83 | 14.06 | 14.86 | 10.53 | 11.44 | 16.12 |
| SE-NeRF (NeRF) | 19.96 (+4.88) | 14.72 (+2.74) | 19.29 (+2.13) | 16.06 (+2.23) | 16.45 (+0.14) | 17.51 (+0.20) | 14.20 (+3.36) | 21.09 (+4.80) |
| SE-NeRF ($K$-Planes) | 20.54 (+4.93) | 13.38 (+0.15) | 18.33 (+0.04) | 20.14 (+7.69) | 16.65 (+1.98) | 17.01 (+0.71) | 13.72 (+0.37) | 20.13 (+0.39) |
| SE-NeRF (Instant-NGP) | 20.40 (+2.74) | 13.34 (+0.59) | 19.07 (+0.63) | 18.15 (+4.48) | 15.99 (+2.82) | 17.94 (+1.11) | 14.61 (+0.79) | 20.23 (+1.18) |

## 5.2 COMPARISON

**Qualitative comparison.** Figure 4 and Figure 5 illustrates the robustness of our model to unknown views, even when the pose differs significantly from the training views. Our model demonstrates robust performance on unknown data, surpassing the baselines. This is particularly evident in the "chair" scene, where all existing methods exhibit severe overfitting to the training views, resulting in heavy artifacts when the pose significantly changes from those used during training. RegNeRF (Niemeyer et al., 2022) fails to capture the shape and geometry in unknown views and although DietNeRF (Jain et al., 2021) is capable of capturing the shape of the object accurately, it produces incorrect information, such as transforming the armrests of the chair into wood. In contrast, SE-NeRF maintains the shape of an object even from further views with less distortion, resulting in the least artifacts and misrepresentation.

**Quantitative comparison.** Table 1 and Table 2 show quantitative comparisons of applying our framework against other few-shot NeRFs and our baseline models on NeRF synthetic and LLFF datasets. As shown in Table 1, SE-NeRF outperforms previous few-shot NeRF models in the NeRF synthetic Extreme and the conventional 4-view setting. By applying SE-NeRF, we observe an general improvement in performance over different methods and different datasets, demonstrating that our framework successfully guides networks of existing methods to learn more robust knowledge of the 3D scene.

## 5.3 ABLATION STUDY.

**Iterative training.** As shown in Figure 6, which presents the quantitative results for each iteration, a significant improvement in performance can be observed after the first iteration. The performance

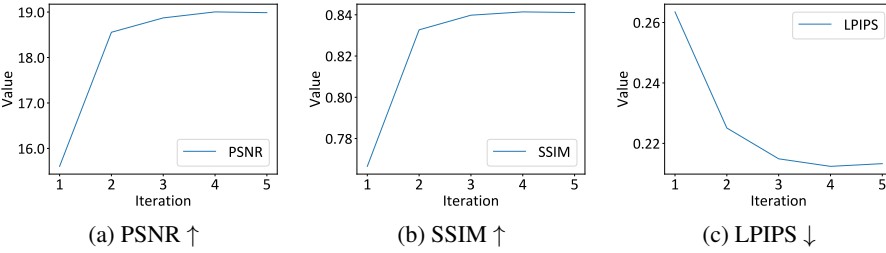

|  (a) PSNR ↑ | (b) SSIM ↑ | (c) LPIPS ↓ |

Figure 6: **Quantitative improvement from baseline after multiple iterations.**

continues to be boosted with each subsequent iteration until the convergence. Based on our experimental analysis, we find that after the simultaneous distillation of reliable rays and regularization of unreliable rays in the first iteration, there is much less additional knowledge to distill to the student in certain scenes which leads to a smaller performance gain from the second iteration. However, although the performance gain in terms of metrics is small, the remaining artifacts and noise in the images continue to disappear after the first iteration, which is important in perceptual image quality.

**Prior-based ray distillation.** In Table 3, we conduct an ablation study on the "lego" scene of the NeRF Synthetic Extreme setting and show that using both reliable and unreliable ray distillation is crucial to guide the net-

Table 3: **Ray distillation ablation.**

| Method | PSNR ↑ | SSIM ↑ | LPIPS ↓ | Average ↓ |
|---|---|---|---|---|
| $K$-Planes | 14.67 | 0.68 | 0.31 | 0.30 |
| $K$-Planes + Reliable | 16.15 (+1.48) | 0.72 (+0.04) | 0.27 (-0.04) | 0.27 (-0.03) |
| $K$-Planes + Reliable/Unreliable | **16.65 (+1.98)** | **0.75 (+0.07)** | **0.24 (-0.07)** | **0.25 (-0.05)** |

work to learn a more robust representation of the scene, showing the highest results in all metrics. This stands in contrast to existing semi-supervised appraoches (Xie et al., 2020; Amini et al., 2023), which typically discard unreliable pseudo labels to prevent the student learning from erroneous information Arazo et al. (2020). We show that when applying self-training to NeRF, the unreliable labels can be further facilitated by the prior knowledge that depth within a 3D space exhibits smoothness.

**Thresholding.** In Table 4, we show the results of `SE-NeRF` trained on the NeRF Synthetic Extreme setting with different thresholding strategies. Following traditional semi-supervised approaches (Tur et al., 2005; Cascante-Bonilla et al., 2021; Zhang et al., 2021a; Chen et al., 2023), we conducted experiments using a pre-

Table 4: **Thresholding ablation.**

| Threshold | PSNR ↑ | SSIM ↑ | LPIPS ↓ | Avg. ↓ |
|---|---|---|---|---|
| Fixed | 17.02 | 0.77 | 0.25 | 0.25 |
| Unified | 15.95 | 0.73 | 0.28 | 0.27 |
| Adaptive | **17.49** | **0.78** | **0.23** | **0.24** |

defined fixed threshold, adaptive threshold (ours), and a unified threshold which does not classify psuedo labels as reliable and unreliable but uses the similarity value to decide how much the distillation should be made from the teacher to the student. The adaptive thresholding method resulted in the most performance gain, showing the rationale of our design choice. A comprehensive and detailed analysis regarding the threshold selection process is provided in Appendix B.4.

## 6 CONCLUSION AND LIMITATIONS

In this paper, we present a novel self-training framework Self-Evolving Neural Radiance Fields (`SE-NeRF`), specifically designed for few-shot NeRF. By employing a teacher-student framework in conjunction with our unique implicit distillation method, which is based on the estimation of ray reliability through feature consistency, we demonstrate that our self-training approach yields a substantial improvement in performance without the need for any 3D priors or modifications to the original architecture. Our approach is able to achieve state-of-the-art results on multiple settings and shows promise for further development in the field of few-shot NeRF.

However, our framework also shares similar limitations to existing semi-supervised approaches. 1) Sensitivity to inappropriate pseudo labels: when unreliable labels are classified as reliable and used to train the student network, this leads to performance degradation of the student model. 2) Teacher initialization: if the initialized teacher network in the first iteration is too poor, our framework fails to enhance the performance of the models even after several iterations. Even with these limitations, our framework works robustly in most situations, and we leave the current limitations as future work.

## 7 REPRODUCIBILITY STATEMENT

For the reproducibility of our work, we will release all the source codes and checkpoints used in our experiments. For those who want to try applying our self-training framework to existing works, we provide the pseudo codes for our reliability estimation method for the per-ray pseudo labels and the overall self-training pipeline.

---

**Algorithm 1** Reliability estimation method for per-ray pseudo labels

---

1: **Input:** Labeled Image $I$, rendered Image $I^+$, rendered depth $D^+$, threshold $\tau$
2: **Output:** Mask $M$ for $I^+$
3: $f \leftarrow \text{VGG19}(I)$
4: $f^+ \leftarrow \text{VGG19}(I^+)$
5: **for** $i \leftarrow 0$ to (Height - 1) **do**
6:     **for** $j \leftarrow 0$ to (Width - 1) **do**
7:         $(i', j') \leftarrow \text{Warp}(I^+, D^+, I, i, j)$      $\triangleright$ $I_{i,j}^+$ is warped to $I_{i',j'}$ using rendered depth $D^+$
8:         $S \leftarrow \text{CosineSimilarity}(f_{i,j}^+, f_{i',j'})$
9:         **if** $S > \tau$ **then**
10:             $M_{i,j} \leftarrow 1$
11:         **else**
12:             $M_{i,j} \leftarrow 0$
13:         **end if**
14:     **end for**
15: **end for**

---

**Algorithm 2** Self-Training

---

1: **Input:** Teacher Network $\mathbb{T}$, set of labeled ray $\mathcal{R}$, set of rendered ray $\mathcal{R}^+$
2: **Output:** Teacher Network $\mathbb{T}$ for next iteration
3: **for** each $step$ **do**
4:     Initialize $\mathbb{S}$                    $\triangleright$ Initialize Student Network
5:     Loss $\leftarrow 0$
6:     **for** each $\mathbf{r}$ in $R$ **do**
7:         Loss $\leftarrow$ Loss $+ \text{L2}(c, \text{Color}(\mathbb{S}, r))$
8:     **end for**
9:     **for** each $\mathbf{r}$ in $R^+$ **do**
10:         Evaluate $M(\mathbf{r})$
11:         **if** $M(\mathbf{r}) = 1$ **then**
12:             Loss $\leftarrow$ Loss $+ \text{L2}(\text{Color}(\mathbb{T}, \mathbf{r}), \text{Color}(\mathbb{S}, \mathbf{r}))$      $\triangleright$ Reliable RGB Loss
13:             Loss $\leftarrow$ Loss $+ \text{L2}(\text{Weight}(\mathbb{T}, \mathbf{r}), \text{Weight}(\mathbb{S}, \mathbf{r}))$      $\triangleright$ Reliable Density Loss
14:         **else**
15:             Loss $\leftarrow$ Loss $+ \text{L2}(\text{GaussianWeight}(\mathbb{T}, \mathbf{r}), \text{Weight}(\mathbb{S}, \mathbf{r}))$ $\triangleright$ Unreliable Density Loss
16:         **end if**
17:         Update $\mathbb{T}$ with Loss
18:     **end for**
19: **end for**
20: $\mathbb{T} \leftarrow \mathbb{S}$

---

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

In this supplementary document, we provide a more detailed analysis of our experiments and implementation details, together with additional results of images rendered using our framework.

## A  PRELIMINARY: $K$-PLANES

$K$-Planes (Fridovich-Keil et al., 2023) is a model that uses $\binom{d}{2}$ ("d-choose-2") planes to represent radiance fields in $d$-dimensional scene. This planar factorization makes adding dimension-specific priors easy and induces a natural decomposition of static and dynamic components of a scene. In the static scene, $K$-Planes obtains the features of a 3D coordinate $\mathbf{x} \in \mathbb{R}^3$ from $\binom{3}{2} = 3$ planes which are $\mathbf{P}_{xy}, \mathbf{P}_{yz}$, and $\mathbf{P}_{xz}$. These planes have shape $N \times N \times M$, where $N$ is the spatial resolution and $M$ is the size of stored features that represent the scene and will be decoded into density and view-dependent color of the scene. The features $q(\mathbf{x})_k$ of a 3D coordinate $\mathbf{x} \in \mathbb{R}^3$ can be obtained by normalizing its entries between $[0, N)$ and projecting it onto the three planes by

$$q(\mathbf{x})_k = \psi(\mathbf{P}_k, \pi_k(\mathbf{x})), \tag{12}$$

where $\pi_k$ projects $\mathbf{x}$ onto the $k$-th plane and $\psi$ denotes bilinear interpolation of a point into a regularly spaced 2D grid. After repeating the process in Equation 12 for each $k \in K$, the features are combined using the Hadamard product (elementwise multiplication) over the three planes to produce a final feature vector $q(\mathbf{x}) \in \mathbb{R}^M$ with the following equation:

$$q(\mathbf{x}) = \prod_{k \in K} q(\mathbf{x})_k. \tag{13}$$

The final features $q(\mathbf{x})$ are further decoded into color and density using either an explicit linear decoder or a hybrid MLP decoder. We use the hybrid model as our baseline that utilizes the spherical harmonic (SH) basis and a shallow MLP decoder. In the hybrid model, the final features are decoded with two small MLP layers. The first MLP $f_\sigma$ maps $q(\mathbf{x})$ to view-independent density $\sigma \in \mathbb{R}$ and additional features $\hat{q}$ as follows:

$$\sigma(\mathbf{x}), \hat{q}(\mathbf{x}) = f_\sigma(q(\mathbf{x})). \tag{14}$$

The second MLP $f_{\text{RGB}}$ maps additional features $\hat{q}$ and the embedded view direction $\gamma(\mathbf{d})$ to view-dependent color value $\mathbf{c} \in \mathbb{R}^3$.

$$\mathbf{c}(\mathbf{x}, \mathbf{d}) = f_{\text{RGB}}(\hat{q}(\mathbf{x}), \gamma(\mathbf{d})) \tag{15}$$

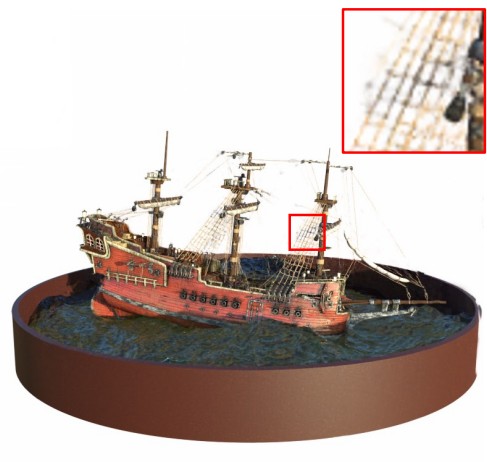 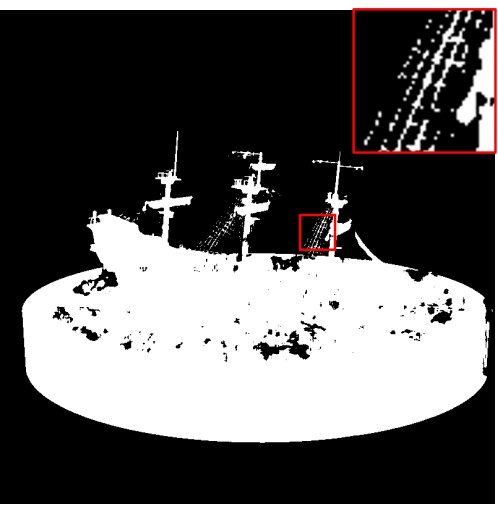

Figure 7: **Pixel-level reliability estimation (Left)** This figure represents an unknown viewpoint rendered by the Teacher network in the "ship" scene of the NeRF Synthetic dataset. **(Right)** This is a binary reliability mask generated through our reliability estimation method. As can be seen in the red box, it is evident that the reliability of the net connecting the sails of the ship is well determined at the pixel-level.

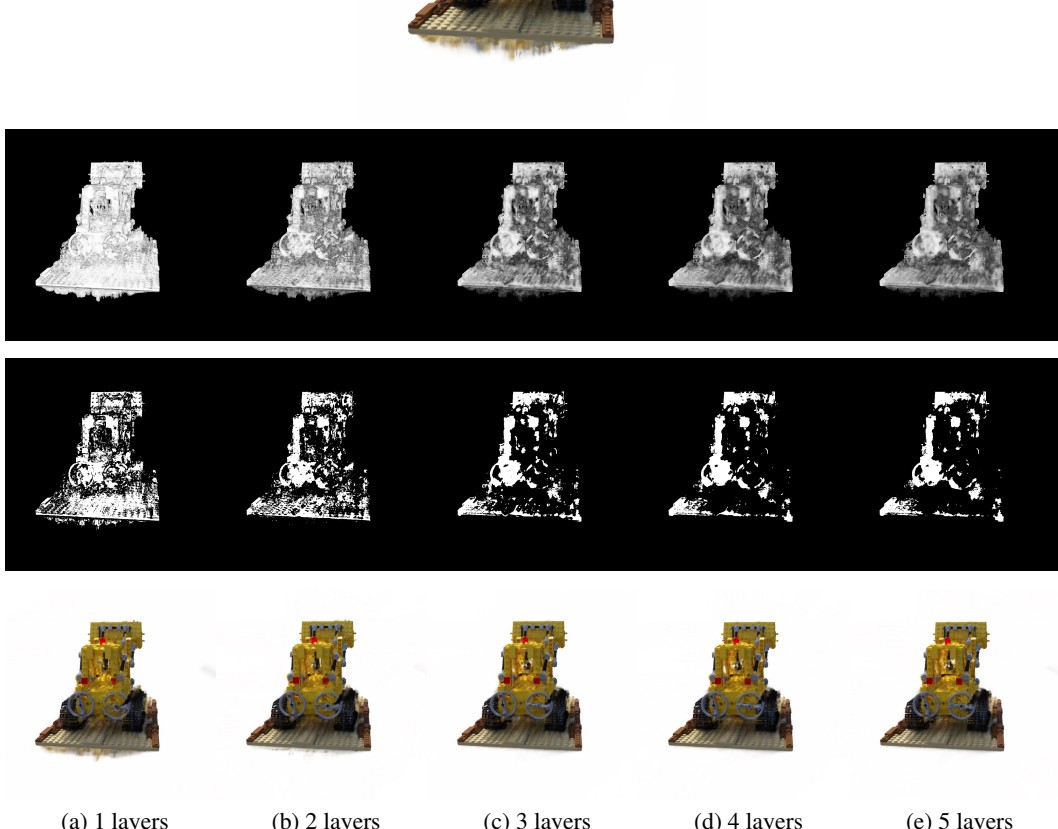

(a) 1 layers     (b) 2 layers     (c) 3 layers     (d) 4 layers     (e) 5 layers

Figure 8: **Feature map of the lego scene.**

# B ANALYSIS

## B.1 RELIABILITY ESTIMATION

In this section, we explain the details of our implementation and the results of our proposed method for estimating the reliability of *unknown* rays generated by the teacher model. To assess the reliability of *unknown* rays, we expand upon an essential insight that when a point located at a surface is projected to multiple viewpoints, the semantics of the projected viewpoints should be consistent unless the point is non-visible due to certain occlusions. This idea has been shown in multiple previous works that formulate NeRF for refined surface reconstruction (Chibane et al., 2021; Truong et al., 2022).

We check this semantic consistency between the *unknown* rays and *known* viewpoints to estimate the reliability of the ray. Though any architectural framework that can provide pixel-level feature maps can be used, we utilize a pre-trained 2D CNN image encoder to extract the semantics of the rays, which is well known for the expressiveness of its features and thus has been adopted as a feature extractor in various works (Chibane et al., 2021; Yu et al., 2021; Wang et al., 2021; Zhang et al., 2021b) that leverage semantic information. Following them, we have conducted experiments employing VGG-19 (Simonyan & Zisserman, 2014), ResNet50 (He et al., 2016), and U-Net (Ronneberger et al.,

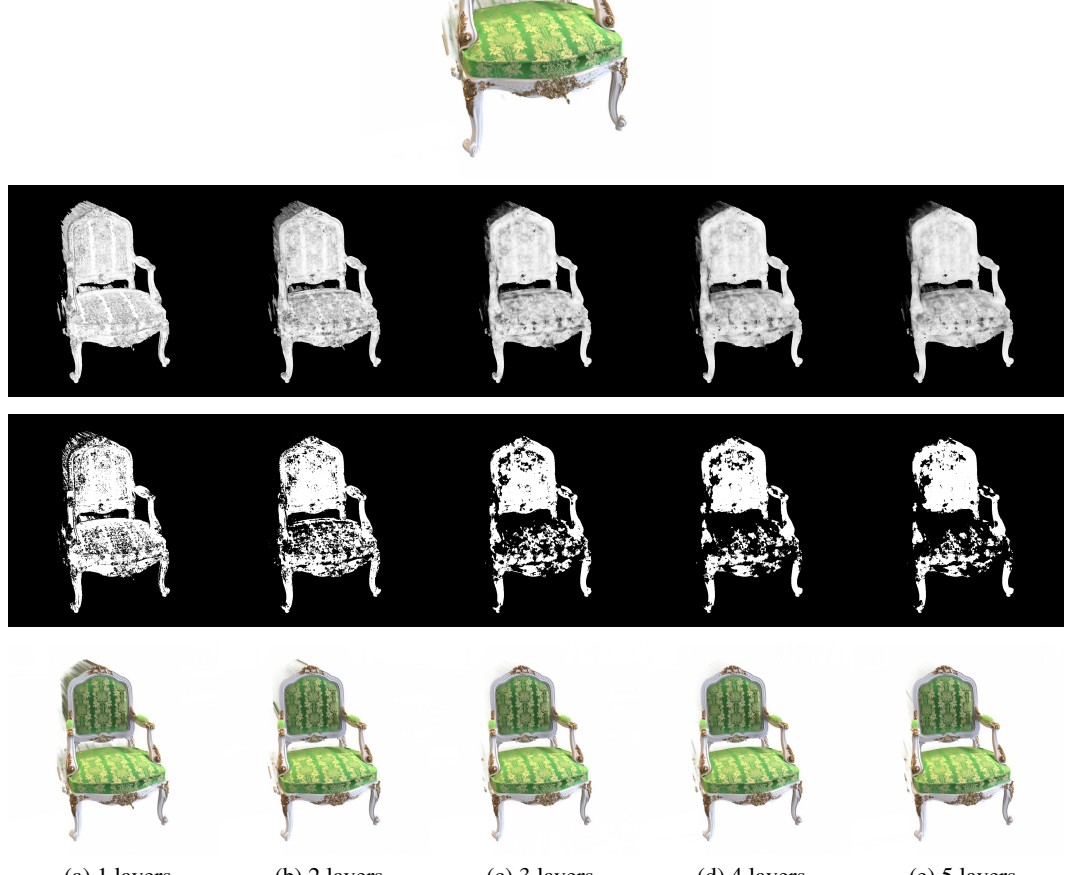

(a) 1 layers        (b) 2 layers        (c) 3 layers        (d) 4 layers        (e) 5 layers

Figure 9: **Feature map of the chair scene.**

Table 5: **PSNR comparison of various feature extractors on NeRF Synthetic Extreme.**

| Methods | chair | drums | ficus | hotdog | lego | mater. | ship | mic | Average |
|---|---|---|---|---|---|---|---|---|---|
| $K$-Planes | 15.61 | 13.23 | 18.29 | 12.45 | 14.67 | 16.30 | 13.35 | 19.74 | 15.45 |
| SE-NeRF ($K$-Planes) w/ U-Net | 19.81 (+4.20) | 13.37 (+0.14) | 18.33 (+0.04) | 20.19 (+7.74) | 16.29 (+1.62) | 16.74 (+0.44) | 13.47 (+0.12) | 19.78 (+0.04) | 17.25 (+1.80) |
| SE-NeRF ($K$-Planes) w/ ResNet50 | 19.96 (+4.35) | 13.50 (+0.27) | 18.42 (+0.13) | 20.36 (+7.91) | 16.28 (+1.61) | 16.89 (+0.59) | 13.96 (+0.61) | 19.77 (+0.03) | 17.39 (+1.94) |
| SE-NeRF ($K$-Planes) w/ VGG-19 | 20.54 (+4.93) | 13.38 (+0.15) | 18.33 (+0.04) | 20.14 (+7.69) | 16.65 (+1.98) | 17.01 (+0.71) | 13.72 (+0.37) | 20.13 (+0.39) | 17.49 (+2.04) |

2015) feature maps in our framework. For VGG-19 and U-Net, features are extracted prior to the first 4 pooling layers whose dimensions are $H \times W$, $H/2 \times W/2$, $H/4 \times W/4$, $H/8 \times W/8$. For ResNet50, features are extracted prior to the first 3 pooling layers whose dimensions are $H/2 \times W/2$, $H/4 \times W/4$, $H/8 \times W/8$. They are upsampled to $H \times W$ using bilinear interpolation and then concatenated to form latent vectors aligned to each pixel. Though all cases resulted in noticeable performance enhancements across different networks as shown in Table 5, we empirically selected VGG-19 as our default feature extractor, which shows the highest performance improvement.

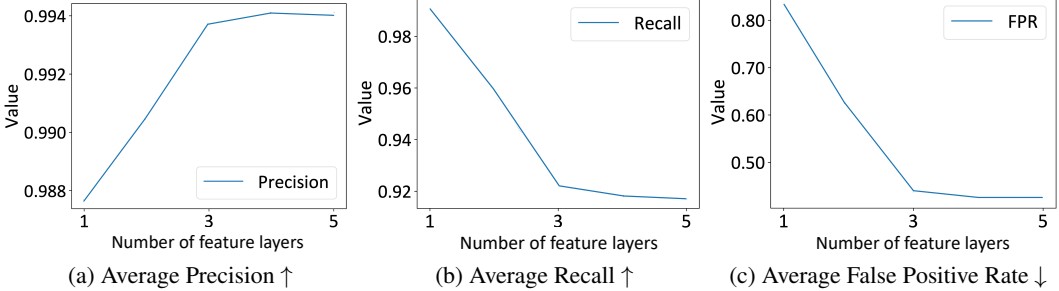

(a) Average Precision ↑     (b) Average Recall ↑     (c) Average False Positive Rate ↓

Figure 10: **Evaluation of masks generated using different numbers of feature layers.** This figure shows the evaluation results of the generated masks using 1∼5 feature layers. Each figure shows the average Precision, average Recall, average False Positive Rate (FPR) of the generated binary mask compared with the binary mask constructed with ground truth RGB values.

As can be seen in Figure 7, our reliability estimation method using VGG-19 generates a binary reliability mask that accurately predicts reliability at the pixel-level. In the following section, we show the comparison results of leveraging different numbers of feature layers.

### B.2 USING DIFFERENT NUMBERS OF FEATURES

To figure out how many features we should take, we compare the results of the generated binary reliability mask using different numbers of feature layers of 1 to 5. Specifically, we select layers 3, 8, 17, 26, and 35 of the VGG-19 network. The results are shown in Figure 8 and Figure 9. The figures show the result of the similarity maps in first row and estimated binary reliability masks in second row by leveraging different numbers of feature layers. The binary reliability mask is constructed by applying a threshold at the 0.85 percentile of the overall scene similarity values. The third row of Figure 8 and Figure 9 represents the results of one iteration of training using each reliability mask. As the number of feature layers increases, the mask is improved by calculating lower similarity values for erroneous artifacts thus effectively removing the artifacts.

As we leverage reliable rays from the binary reliability mask to distill the density weights and colors and also regularize unreliable rays, it is important to lower the False Positive Ratio (FPR) to prevent the confirmation bias problem (Arazo et al., 2020). In Figure 10, we can observe that although the average Recall decreases as the number of feature layers increases, the FPR decreases much more sharply. Our analysis of this phenomenon is that the shallow layers of the CNN only model highly local features (e.g., edges, corners). This leads the features from different locations to have similar semantics, shown in the high Recall when using only 1 feature layer.

**Reliability estimation of pseudo labels.**  To evaluate the results of our proposed ray reliability estimation method, we compare the reliability binary masks generated by our feature consistency method with the binary masks created using ground-truth RGB values. The results in Table 6 demonstrate that our method provides a reasonable estimation of the reliability of the rays.

Table 6: **Reliability mask evaluation.**

| | |
|---|---|
| Precision | 98.03 |
| Recall | 92.25 |

### B.3 UNRELIABLE RAY DISTILLATION

**Applying Gaussian mask.**  Given a rendered image in $\mathcal{S}^+$, let the ray going through pixel coordinate $(i, j)$ be $\mathbf{r}_{ij}$. For each reliable ray going through a pixel near $(i, j)$ in the image, we derive the weight from 2D isotropic Gaussian distribution centered at $(i, j)$ with standard deviation 1, whose probability density function is $G(x, y, i, j) = \frac{1}{\sqrt{2\pi}}\exp(-\frac{(x-i)^2+(y-j)^2}{2})$. Specifically, we use $Q \times Q$ discretely approximated version whose probability mass function is $g(x, y, i, j)$. Then, the Gaussian weighted-average density $\tilde{\sigma}(\mathbf{r}_{ij}(t); \theta^{\mathbb{T}})$ is calculated as:

$$\tilde{\sigma}(\mathbf{r}_{ij}(t); \theta^{\mathbb{T}}) = \sum_{x\in\Omega_x}\sum_{y\in\Omega_y}\frac{M(\mathbf{r}_{xy}) \cdot g(x, y, i, j)}{\sum_{x\in\Omega_x}\sum_{y\in\Omega_y}M(\mathbf{r}_{xy}) \cdot g(x, y, i, j)}\sigma(\mathbf{r}_{xy}(t); \theta^{\mathbb{T}}) \qquad (16)$$

where $\Omega_x = [i - \lfloor \frac{Q}{2} \rfloor, i + \lfloor \frac{Q}{2} \rfloor] \backslash \{i\}$ and $\Omega_y = [j - \lfloor \frac{Q}{2} \rfloor, j + \lfloor \frac{Q}{2} \rfloor] \backslash \{j\}$. We apply the above process for unreliable rays in $\mathcal{R}^+$ only when reliable rays exist in the adjacent $Q \times Q$ kernel which leads to our prior-based geometry distillation loss $\mathcal{L}_g^{\mathcal{P}}$.

Table 7: **PSNR comparison between different kernel size.**

| Methods | chair | drums | ficus | hotdog | lego | mater. | ship | mic |
|---|---|---|---|---|---|---|---|---|
| SE-NeRF ($K$-Planes) w/ kernel 5 | 20.27 | 13.29 | 18.03 | 20.33 | 16.60 | 16.88 | 13.86 | 20.47 |
| SE-NeRF ($K$-Planes) w/ kernel 3 | 20.54 | 13.38 | 18.33 | 20.14 | 16.65 | 17.01 | 13.72 | 20.13 |

**Technical design for kernel size.** The necessity to handle unreliable rays arises when employing Fixed or Adaptive thresholding. We adopt a Gaussian mask with a kernel size of 3 as the default unreliable ray distillation method in our framework. The technical design for this choice is that although incorporating the depth smoothness prior is intuitive, we find that this assumption can be too strong in cases when modeling high-frequency regions. As we directly distill the point-wise weights of points sampled from the ray, we must apply the depth-smoothness prior exclusively in highly adjacent regions. In Table 7, we compare SE-NeRF ($K$-Planes) using different sizes of kernels of 3 and 5 in the NeRF Synthetic Extreme setting. Although the overall performance boost does not show a big difference, scenes that contain high-frequency regions such as 'ficus' from the NeRF Synthetic dataset exhibit suboptimal results. To ensure the robust operation of our framework in extreme settings, we have selected three as the default kernel size for unreliable ray distillation.

### B.4  PSEUDO LABELING

As mentioned in Section 4.2, there are three methods frequently used in traditional semi-supervised frameworks: fixed thresholding, adaptive thresholding, and using a unified equation. In this section, we explain each of the methods in detail about how the reliable and unreliable rays can be classified using each of the methods and show that using the adaptive thresholding method shows the best results. The equation to classify reliable and unreliable rays is as follows:

$$M_l(\mathbf{r_p}) = \min \left\{ \sum_{i \in \mathcal{S}} \mathbb{1} \left[ \frac{\mathbf{f_p^i} \cdot \mathbf{f_p^l}}{\|\mathbf{f_p^i}\| \|\mathbf{f_p^l}\|} > \tau \right], 1 \right\}. \tag{17}$$

**Fixed thresholding.** Fixed thresholding is the most naïve and straightforward way of thresholding which was mainly used in early semi-supervised approaches (Tur et al., 2005). By predefining a specific threshold, it exploits unlabeled examples with confidence values that are above the predefined threshold. Although simple and intuitive, the predefined threshold has to be carefully chosen which needs to be done empirically. Also, only taking values over the fixed threshold sometimes results in having no reliable values after the thresholding process, which lowers the performance boost of semi-supervised frameworks. When it comes to applying self-training to NeRF, we found that the distribution of similarities estimated by our proposed method is significantly different for each of the scenes, which makes choosing a predefined threshold tricky. For our experiments, we have empirically set the threshold value as 0.6, after observing the similarity distributions of all scenes.

**Adaptive thresholding.** To resolve the aforementioned problems, we try taking the top-K confidence value or the $(1 - \alpha)^{th}$ percentile value. This approach has the advantage of always allowing the semi-supervised framework to utilize reliable pseudo labels to improve the performance of the network. However, this approach also has a disadvantage in the sense that even if a majority of pseudo labels are actually reliable, only the top-K labels can be classified as reliable. This constrains the network to learn from as many reliable pseudo labels as it can which constrains the overall performance boost. To mitigate this problem, recent methods (Cascante-Bonilla et al., 2021; Zhang et al., 2021a) which utilize curriculum learning (Bengio et al., 2009) adaptively increase the value K of top-K, allowing the model to learn from more labels as the iteration progresses. We also found that adopting adaptive thresholding together with curriculum learning resulted in the best results in our setting. Specifically, we use the $(1 - \alpha)^{th}$ percentile of the confidence distribution starting from $\alpha = 0.15$ and increase the value by 0.05 at each step. This strategy allows us to maximize the advantages gained from expanding the labels through the process of self-training.

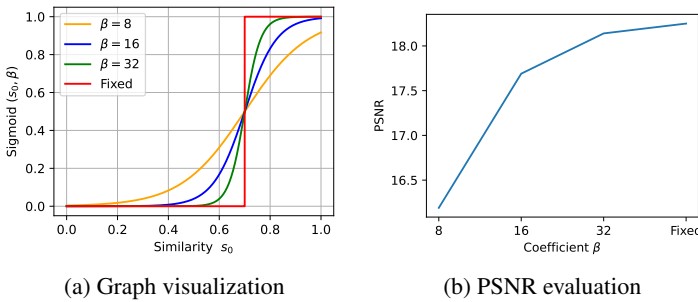

(a) Graph visualization    (b) PSNR evaluation

Figure 11: **Performance evaluation according to $\beta$.**

**Unified Equation.**    Instead of using an explicit threshold to classify reliable and unreliable pseudo labels, some methods (Chen et al., 2023) adopt a unified equation that uses all pseudo labels with their confidence values taken into account. The advantage of using the unified equation is that it does not require any initial value such as the fixed threshold or the K for the top-K method. By giving higher weights to pseudo labels with higher confidence values and vice versa, these methods show that utilizing the unified equation successfully guides the framework to benefit mostly from the reliable pseudo labels.

Following this approach, we propose how the unified equation can be designed when it comes to our reliable and unreliable ray distillation method. As we want to distill the pseudo labels of the reliable rays directly to the student and regularize the unreliable rays with nearby reliable rays, we first construct our equation of giving higher weights to rays whose similarity values are high and regularize the rays with information from nearby rays by Gaussian weighting proposed in Section B.3.

Given a rendered image in $\mathcal{S}^+$, let the ray going through pixel coordinate $(i, j)$ is $\mathbf{r}_{ij}$. Then, the geometry distillation loss $\mathcal{L}_g(\mathbf{r}_{ij}, \theta)$ for each ray $\mathbf{r}_{ij}$ is formulated as :

$$\mathcal{L}_g(\mathbf{r}_{ij}, \theta) = s_{ij} \Delta \sigma_{ij}^2 + (1 - s_{ij}) \sum_{x \in \Omega_x} \sum_{y \in \Omega_y} \frac{\max(\Delta s_{xy}, 0) \cdot g(x, y, i, j)}{\sum_{x \in \Omega_x} \sum_{y \in \Omega_y} \max(\Delta s_{xy}, 0) \cdot g(x, y, i, j)} \Delta \sigma_{xy}^2 \quad (18)$$

where $\Delta \sigma_{xy}^2 = \sum_{t,t' \in T} \|\sigma(\mathbf{r}_{xy}(t); \theta^{\mathbb{T}}) - \sigma(\mathbf{r}_{ij}(t'); \theta)\|_2^2$, $\Delta s_{xy} = s_{xy} - s_{ij}$, $\Omega_x = [i - \lfloor \frac{Q}{2} \rfloor, i + \lfloor \frac{Q}{2} \rfloor]\backslash\{i\}$ and $\Omega_y = [j - \lfloor \frac{Q}{2} \rfloor, j + \lfloor \frac{Q}{2} \rfloor]\backslash\{j\}$.

The first term of the equation describes how much the ray should learn directly from the identical ray of the teacher and the second term describes how much the the ray should learn from nearby reliable rays. To let reliable rays of the teacher network be distilled directly to the student the first term is multiplied by the similarity value $s_{xy}$. Also, to guide the ray to learn from only the nearby rays that are more reliable, we perform the weighted sum on $\Delta \sigma_{xy}^2$ with $\max(\Delta s_{xy}, 0)$ not $\Delta s_{xy}$.

Similarly, photometric loss for each ray $\mathbf{r}_{ij}$ is formulated as:

$$\mathcal{L}_c(\mathbf{r}_{ij}, \theta) = s_{ij} \|C(\mathbf{r}_{ij}; \theta^{\mathbb{T}}) - C(\mathbf{r}_{ij}; \theta)\|_2^2 \quad (19)$$

Then, the student network is optimized by following equation:

$$\theta^{\mathbb{S}} = \underset{\theta}{\arg\min} \left\{ \mathcal{L}_{\text{photo}}(\theta) + \sum_{\mathbf{r} \in \mathcal{R}^+} \mathcal{L}_c(\mathbf{r}, \theta) + \mathcal{L}_g(\mathbf{r}, \theta) \right\} \quad (20)$$

Although the unified equation has the advantage of avoiding the need to manually search for an appropriate threshold value, we find that using the unified equation does not perform as well as initially anticipated. In detail, the unified equation resulted in the smallest performance gain even after several iterations. To further analyze this phenomenon, we conduct additional experiments applying several sigmoid functions with different coefficient $\beta$ to similarity $s_{ij}$, for comparison with fixed thresholding method.

$$\text{Sigmoid}(s_{ij}, \beta) = \frac{1}{1 + \exp(-\beta(s_{ij} - 0.7))} \quad (21)$$

Experiment is conducted on chair and lego scene with NeRF Synthetic Extreme setting. Quantitative results are shown at Figure 11 and qualitative results are shown at Figure 12. Both shows that using a sigmoid function with larger $\beta$ (which means getting closer to the fixed thresholding) leads to higher performance gains. As a result, we deduce that using ambiguous boundaries for reliable and unreliable rays leads to unwanted distillation to the student causing confirmation bias, which constraints the overall performance gain of the model.

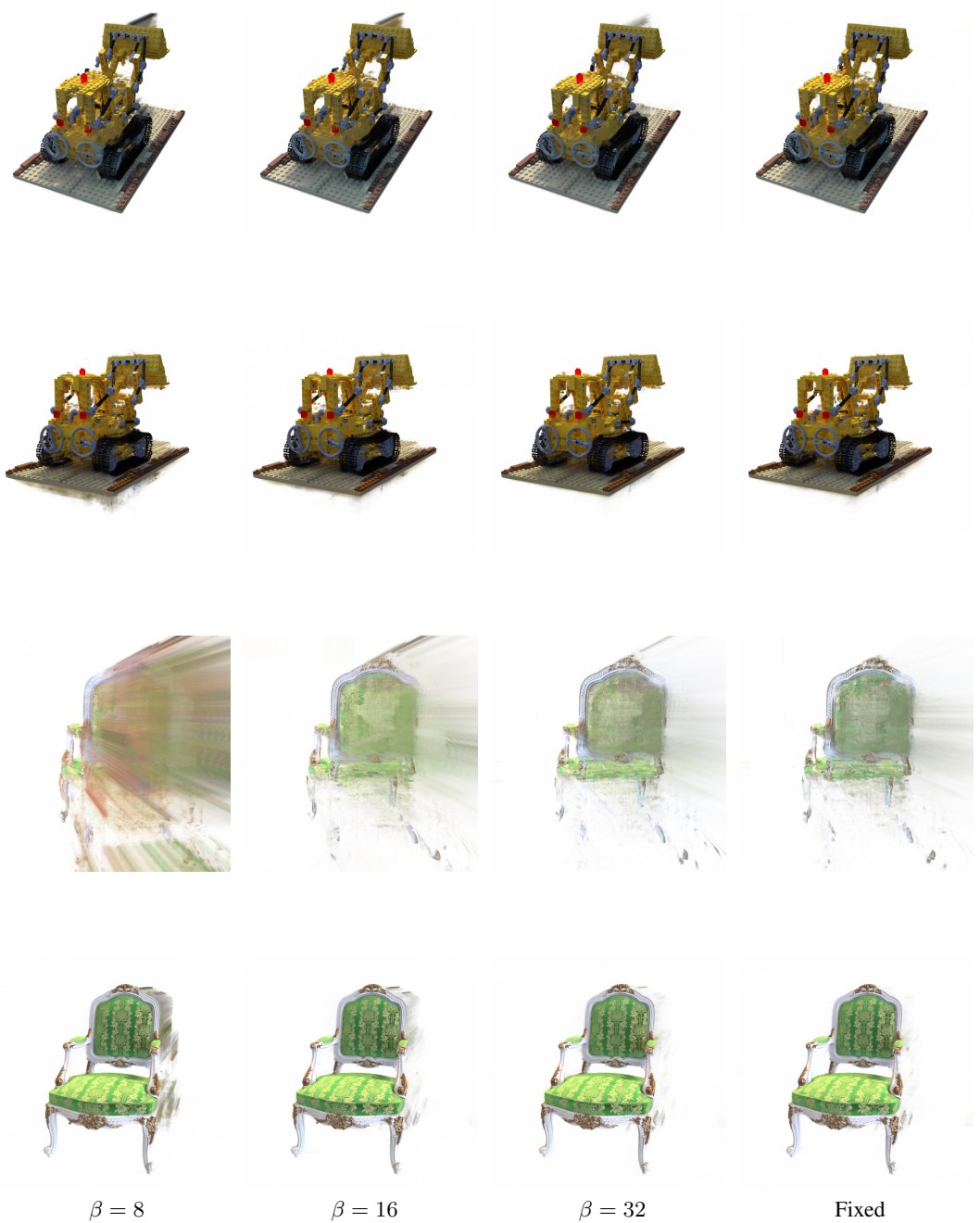

$\beta = 8$        $\beta = 16$        $\beta = 32$        Fixed

Figure 12: **Qualitative results according to $\beta$.**

### B.5 Limitations and Future Work

Although our framework successfully guides existing NeRF models to learn a more robust representation of the scene, our framework shares similar limitations to existing semi-supervised frameworks. 1) Sensitivity to inappropriate pseudo labels: when unreliable labels are classified as reliable and used to train the student network, this leads to performance degradation of the student model. 2) Teacher initialization: if the initialized teacher network in the first iteration is too poor, our framework fails to enhance the performance of the models even after several iterations.

**Sensitivity to inappropriate pseudo labels** As known as the confirmation bias problem in semi-supervised frameworks (Arazo et al., 2020), our framework also fails to guide the student network to learn a more robust representation of the scene in cases where a majority of unreliable rays are misclassified as reliable. The following cases show when the network was trained with a completely wrong reliability mask, where the first two images show the initial rendered image and estimated depth, and the last two images show the rendered image and estimated depth after 4 iterations. The wrongly estimated mask in the middle leads to performance degradation.

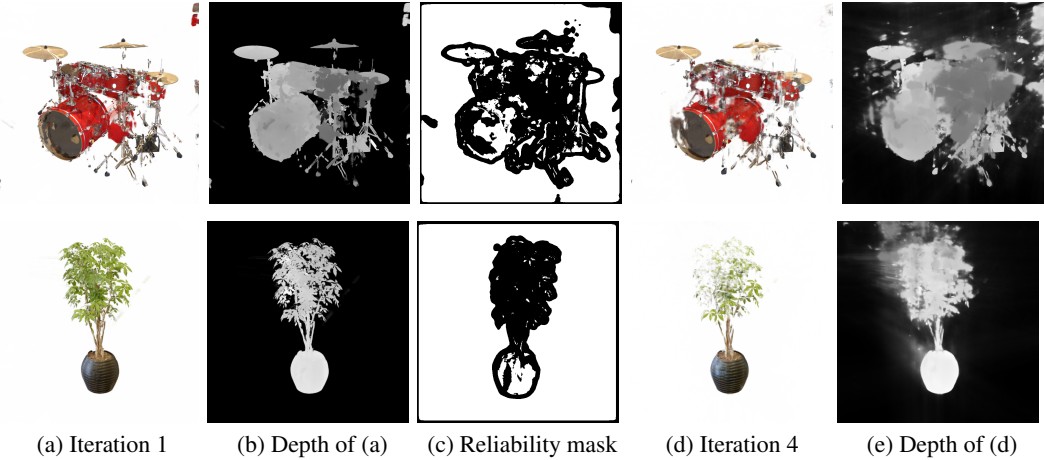

(a) Iteration 1          (b) Depth of (a)          (c) Reliability mask          (d) Iteration 4          (e) Depth of (d)

**Teacher initialization** If the rendered images from the initialized teacher model only have a small portion of the reliable regions or too many artifacts, our framework struggles to guide the student to capture the geometry of the scene successfully. In these cases, we notice that the small portion of correct geometry also becomes incorrect after iterations. The first two images are the rendered images from the first network and the last two images are the rendered images after 4 iterations.

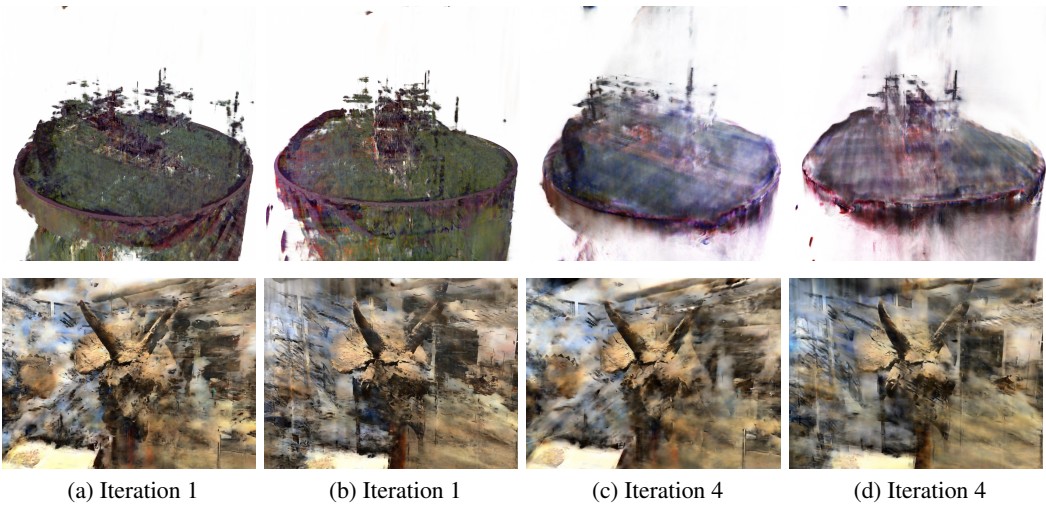

(a) Iteration 1          (b) Iteration 1          (c) Iteration 4          (d) Iteration 4

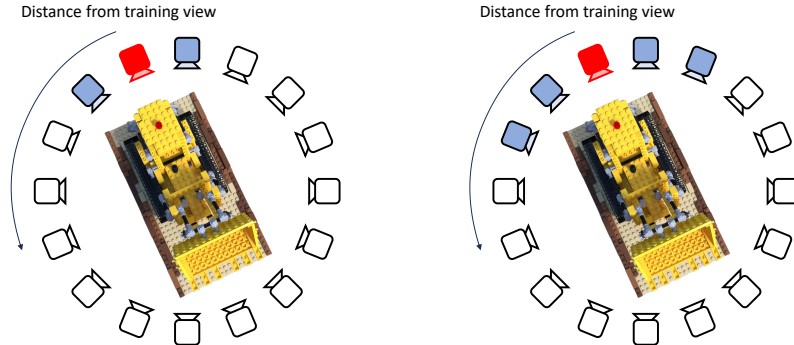

Figure 13: **Progressive view selection (Left)** This figure visualizes the camera poses in early iterations steps. The red camera indicates the position of the given input view, and the blue camera indicates the positions we select to generate pseudo labels. **(Right)** After iterations, we increase the range of where the pseudo labels are generated and utilize rendered images from further views of the given view as pseudo labels.

To enable our framework to guide the student network more robustly, we incorporated several methods for training. For both $K$-Planes (Fridovich-Keil et al., 2023) and NeRF (Mildenhall et al., 2021), we start with generating the pseudo labels from viewpoints that are near the given input views. This is straightforward as the closer the views are to the known views, the more robust the rendered images are. By calculating the pose $\phi, \theta$ of the input viewpoints, we start using the rendered images in the range of 10 degrees ($-10 \leq \phi \leq 10, -10 \leq \theta \leq 10$) as the pseudo labels. After the images rendered from near viewpoints do not give any more additional information about the scene, we increase the range to guide the student network with new information, as shown in Figure 13.

In addition to the progressive pseudo view selection, we find that training NeRF (Mildenhall et al., 2021) directly with few-shot images leads to poor initializations, making our framework struggle to guide the student to a better representation. This was caused mostly by the heavy artifacts that are located between the scene and the camera, and to mitigate this problem, Park et al. (2021) proposes a coarse-to-fine frequency annealing strategy that forces the network to learn the low-frequency details first and the high-frequency details after the coarse features are successfully learned. We follow Park et al. (2021) and define the weight for each frequency band j as:

$$w_j(\eta) = \frac{(1 - \cos(\pi \mathrm{clamp}(\eta - j, 0, 1)))}{2}, \tag{22}$$

where $\eta$ is the parameter in the range of $\eta \in [0, m]$ when $m$ is the number of frequency bands used for the positional encoding $\gamma(\cdot)$. The $\eta$ is a hyper-parameter defined as $\eta(t) = \frac{mt}{N}$ where $t$ is the current training iteration, and $N$ is a hyper-parameter that defines when the network should utilize the entire frequency bands. For our experiments, we set $N$ to $\frac{1}{4}$ of the total steps. We also compare the performance of NeRF in the NeRF Synthetic Extreme setting with and without using frequency annealing and `SE-NeRF` applied to NeRF using frequency annealing in Table 8.

Table 8: **Frequency annealing ablation.**

| Method | PSNR ↑ | SSIM ↑ | LPIPS ↓ | Average ↓ |
|---|---|---|---|---|
| NeRF w/o frequency annealing | 12.91 | 0.68 | 0.33 | 0.32 |
| NeRF w frequency annealing | 14.85 (**+1.94**) | 0.73 (**+0.05**) | 0.32 (**-0.01**) | 0.27 (**-0.05**) |
| `SE-NeRF` (NeRF) | 17.15 (**+4.24**) | 0.79 (**+0.11**) | 0.21 (**-0.12**) | 0.22 (**-0.10**) |

With these additional methods incorporated into our framework, `SE-NeRF` can successfully guide existing models to learn a more robust representation of the scene without changing the structure of the models. However, progressive view selection and frequency annealing can be difficult to apply to all existing models, and removing these additional methods are left for future work.

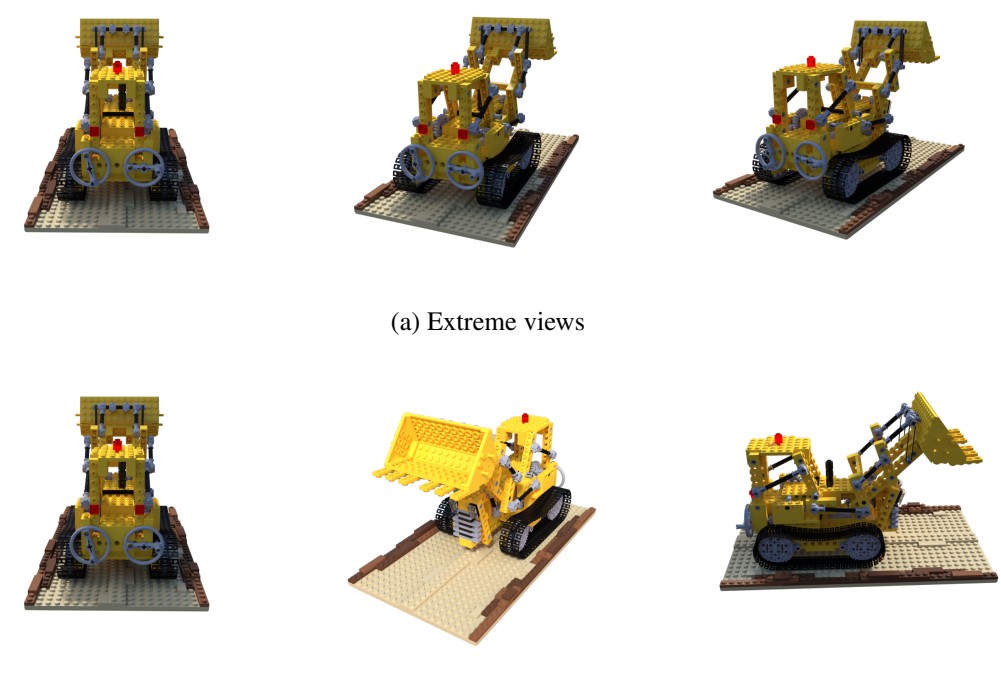

(a) Extreme views

(b) Ideally random views

Figure 14: **Comparison of extreme views and ideally random views.** (a) represents the extreme 3 views that can be randomly selected from the "lego" training set and (b) represents the 3 views that can be ideally selected. While the extreme views only contain information on the backside of the lego object, the ideally random views contain information from all sides of the lego which provides more information to the network. The comparison of (a) and (b) highlights the need for an evaluation protocol to fairly compare each of the methods.

## C VIEW SELECTION

In this section, we show the specific views we selected to compare the performance of multiple methods in Section 5.1 for the extreme views case. The motivation for the extreme view case is that when evaluating the NeRF Synthetic dataset (Mildenhall et al., 2021), although current methods do not show or explain the views used for the evaluation of their models, we found that all methods are very sensitive to how the random views are selected which makes it hard to evaluate and compare their performances. Also, to evaluate how much our framework can resolve the model from overfitting to the given sparse views, we tried to choose the most challenging 3 views to successfully model the geometry of the scene. In Figure 14, we first show how different extreme views and ideally random(where 3 views contain near sufficient information to reconstruct the 3D scene) views for the "lego" scene and then show all the views we selected for each of the scenes in the NeRF Synthetic dataset (Mildenhall et al., 2021).

### C.1 SELECTED VIEWS

Figure 15 shows each of the extreme views we selected for each of the scenes. Specifically, we used view 26,31,32 for the "chair" scene, view 3,9,14 for the "drums" scene, view 12,36,56 for the "ficus" scene, view 31,33,48 for the "hotdog" scene, view 0,1,51 for the "lego" scene, view 47,63,73 for the "materials" scene, view 36,55,66 for the "ship" scene, and view 35,65,82 for the "mic" scene.

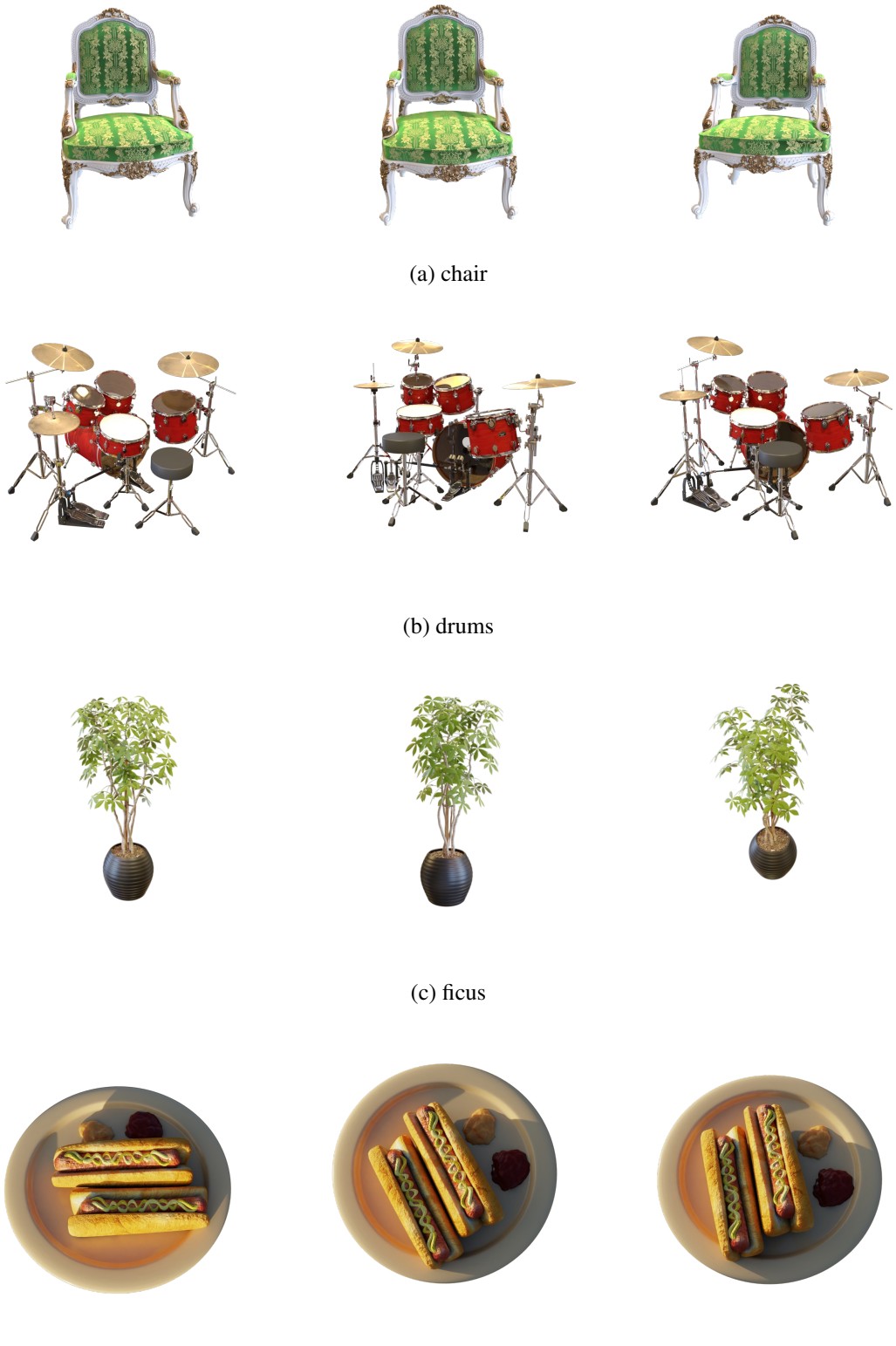

(a) chair

(b) drums

(c) ficus

(d) hotdog

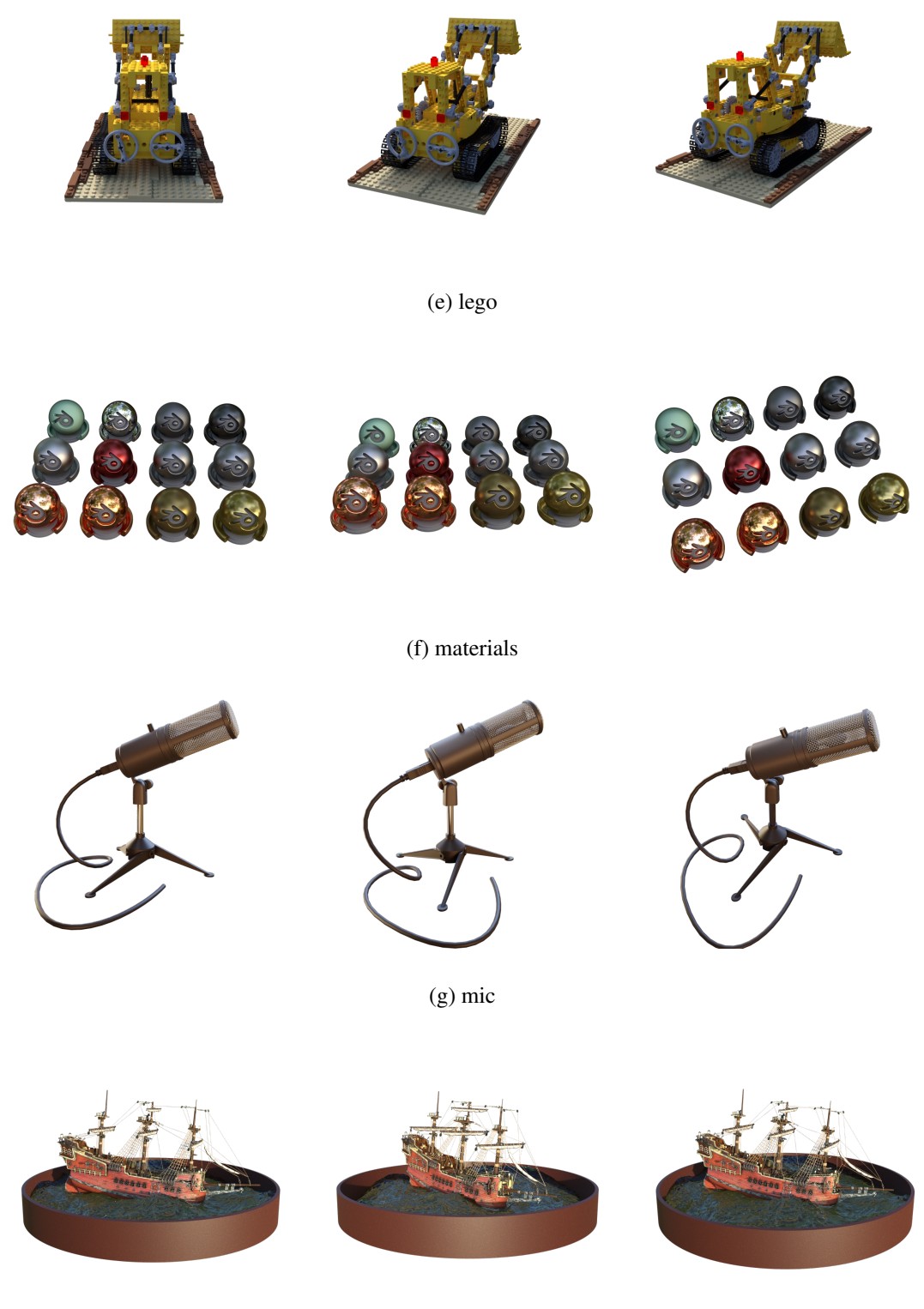

(e) lego

(f) materials

(g) mic

(h) ship

Figure 15: **View selection for NeRF Synthetic Extreme.**

Table 9: **Quantitative comparison on LLFF.**

| Methods | PSNR↑ | | | SSIM↑ | | | LPIPS↓ | | | Average↓ | | |
|---|---|---|---|---|---|---|---|---|---|---|---|---|
| | 3-view | 6-view | 9-view | 3-view | 6-view | 9-view | 3-view | 6-view | 9-view | 3-view | 6-view | 9-view |
| $K$-Planes | 15.77 | 19.58 | 21.72 | 0.44 | 0.66 | 0.73 | 0.46 | 0.30 | 0.24 | 0.41 | 0.29 | 0.25 |
| DietNeRF | 14.94 | 21.75 | 24.28 | 0.370 | 0.717 | 0.801 | 0.496 | 0.248 | 0.183 | 0.240 | 0.105 | 0.073 |
| RegNeRF | 19.08 | 23.10 | 24.86 | 0.587 | 0.760 | 0.820 | 0.336 | 0.206 | 0.161 | 0.149 | 0.086 | 0.067 |
| SE-NeRF ($K$-Planes) | 16.30 (+0.53) | 20.30 (+0.72) | 22.31 (+0.59) | 0.49 (+0.05) | 0.69 (+0.03) | 0.75 (+0.02) | 0.44 (-0.02) | 0.30 (-0.00) | 0.25 (+0.01) | 0.39 (-0.02) | 0.28 (-0.01) | 0.25 (-0.00) |

Table 10: **Quantitative per-scene results on LLFF.**

| Scene | | PSNR↑ | | | SSIM↑ | | | LPIPS↓ | | | Average↓ | | |
|---|---|---|---|---|---|---|---|---|---|---|---|---|---|
| | | 3-view | 6-view | 9-view | 3-view | 6-view | 9-view | 3-view | 6-view | 9-view | 3-view | 6-view | 9-view |
| $K$-Planes | fern | 18.22 | 21.84 | 22.59 | 0.55 | 0.71 | 0.76 | 0.39 | 0.28 | 0.23 | 0.36 | 0.28 | 0.25 |
| | orchids | 12.52 | 16.41 | 17.92 | 0.19 | 0.48 | 0.56 | 0.57 | 0.36 | 0.32 | 0.51 | 0.37 | 0.33 |
| | horns | 14.40 | 18.37 | 21.21 | 0.35 | 0.63 | 0.74 | 0.55 | 0.35 | 0.27 | 0.46 | 0.33 | 0.26 |
| | leaves | 14.44 | 17.21 | 18.51 | 0.37 | 0.57 | 0.65 | 0.47 | 0.32 | 0.27 | 0.43 | 0.33 | 0.29 |
| | trex | 15.04 | 18.95 | 21.57 | 0.45 | 0.69 | 0.79 | 0.49 | 0.31 | 0.23 | 0.42 | 0.29 | 0.23 |
| | room | 14.55 | 20.73 | 22.66 | 0.55 | 0.81 | 0.87 | 0.44 | 0.23 | 0.17 | 0.38 | 0.22 | 0.18 |
| | fortress | 19.74 | 21.33 | 26.71 | 0.65 | 0.76 | 0.84 | 0.30 | 0.26 | 0.17 | 0.30 | 0.25 | 0.19 |
| | flower | 17.25 | 21.81 | 22.62 | 0.41 | 0.67 | 0.70 | 0.49 | 0.29 | 0.28 | 0.42 | 0.29 | 0.28 |
| SE-NeRF ($K$-Planes) | fern | 18.83 | 22.82 | 23.28 | 0.59 | 0.73 | 0.77 | 0.38 | 0.28 | 0.25 | 0.35 | 0.27 | 0.25 |
| | orchids | 13.79 | 17.14 | 18.67 | 0.29 | 0.52 | 0.59 | 0.50 | 0.36 | 0.32 | 0.46 | 0.36 | 0.33 |
| | horns | 14.90 | 19.39 | 21.82 | 0.40 | 0.67 | 0.75 | 0.54 | 0.35 | 0.28 | 0.45 | 0.31 | 0.26 |
| | leaves | 14.92 | 17.59 | 18.80 | 0.40 | 0.59 | 0.65 | 0.47 | 0.34 | 0.28 | 0.43 | 0.33 | 0.30 |
| | trex | 15.32 | 20.05 | 22.28 | 0.50 | 0.73 | 0.80 | 0.47 | 0.29 | 0.23 | 0.40 | 0.27 | 0.23 |
| | room | 14.83 | 21.59 | 23.27 | 0.58 | 0.84 | 0.88 | 0.40 | 0.22 | 0.18 | 0.36 | 0.21 | 0.18 |
| | fortress | 19.99 | 21.45 | 26.87 | 0.69 | 0.77 | 0.85 | 0.31 | 0.27 | 0.18 | 0.29 | 0.25 | 0.19 |
| | flower | 17.84 | 22.33 | 23.51 | 0.45 | 0.70 | 0.73 | 0.49 | 0.29 | 0.27 | 0.41 | 0.29 | 0.26 |

# D    ADDITIONAL RESULTS

In this section, we show additional results of the estimated binary reliability mask using our reliability estimation method. In Figure 16, we compare the ground truth (GT) masks generated by the difference between a fully trained model with 100 training views and a model trained with only 3 training views to the SE-NeRF masks generated by our method. Our binary reliability masks, created using only 3 training views, effectively mask unreliable artifacts in unknown views, similar to the masks generated with GT RGB values. In Figure 17,18,19,20, we show additional results of the improved rendered images from novel viewpoints. In Figure 17, we also present the estimated binary reliability mask, which is used to apply separate distillation schemes for reliable and unreliable rays that significantly improves the quality of images rendered by the baseline model. For the LLFF dataset (Mildenhall et al., 2019), we additionally show the results of $K$-Planes Fridovich-Keil et al. (2023) and SE-NeRF ($K$-Planes) trained with 3, 6, and 9 views. The performance metrics are reported in Table 9 and Table 10.

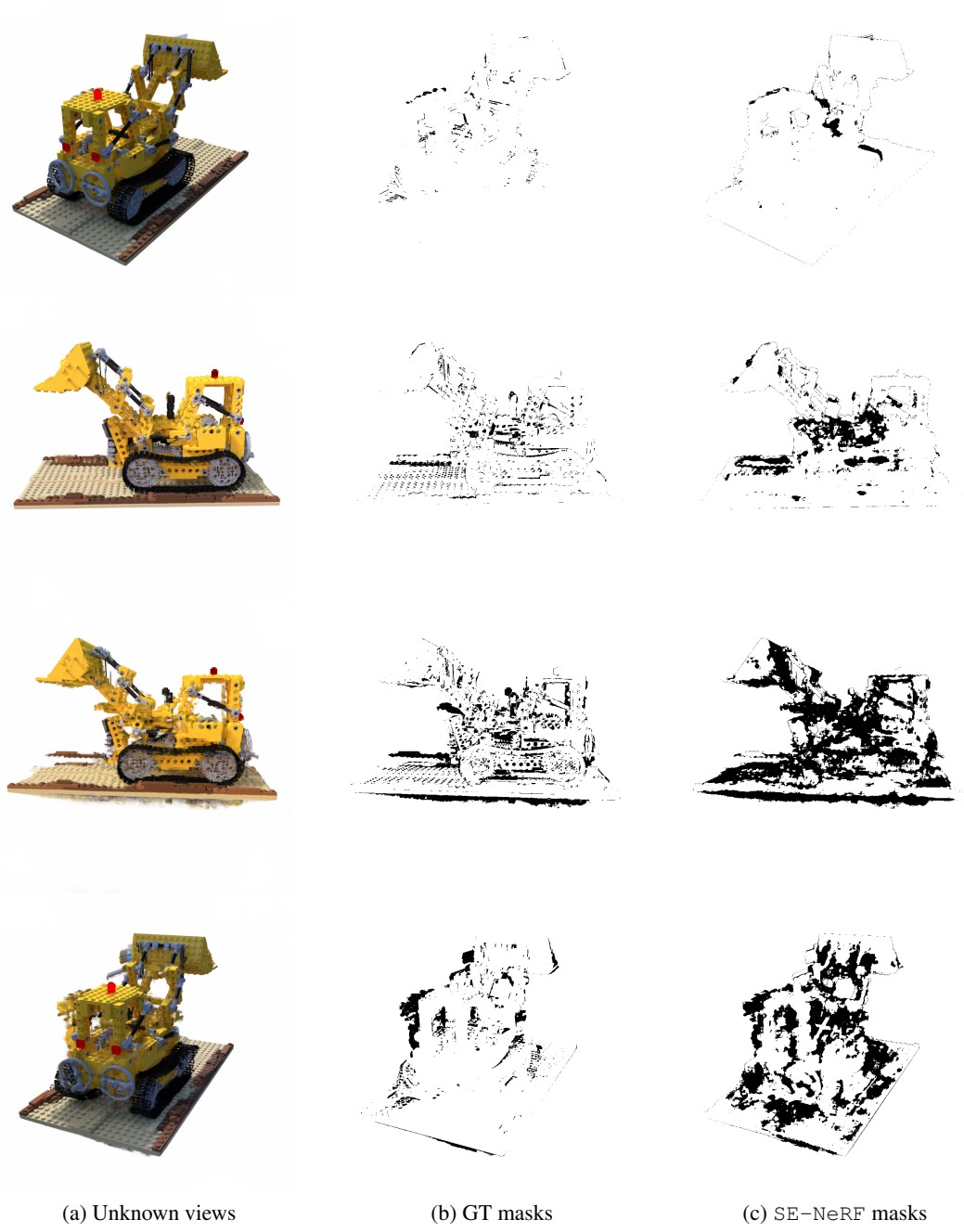

(a) Unknown views        (b) GT masks        (c) `SE-NeRF` masks

Figure 16: **Binary reliability masks.** (b) and (c) represent binary reliability masks for the rendered unknown views in (a). (b) shows the masks generated by training on 100 views, while (c) shows the masks generated by training on 3 views and using our reliability estimation method. Note that for efficient comparison and visualization, we altered the background of the masks generated from our framework to white as shown in (c).

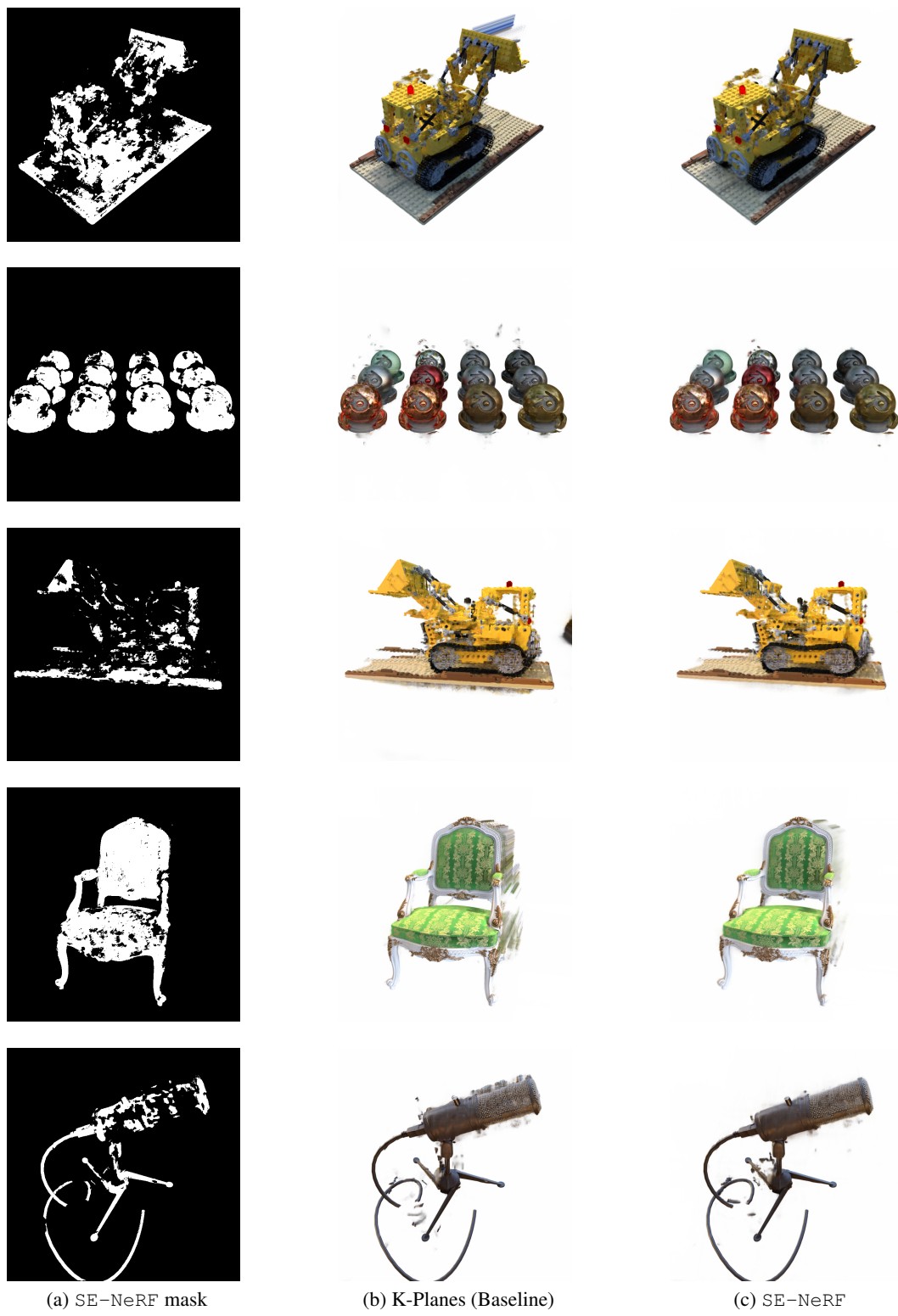

(a) SE-NeRF mask      (b) K-Planes (Baseline)      (c) SE-NeRF

Figure 17: **Qualitative improvement from baseline in 3-view setting (NeRF Synthetic Extreme).** (a) shows the mask generated by training our model in the 3-view setting using our reliability estimation method. As shown in (c), which presents the results of applying the mask from (a) to (b) in the subsequent iterations of training, we are able to remove the artifacts present in (b) and achieve improved performance.

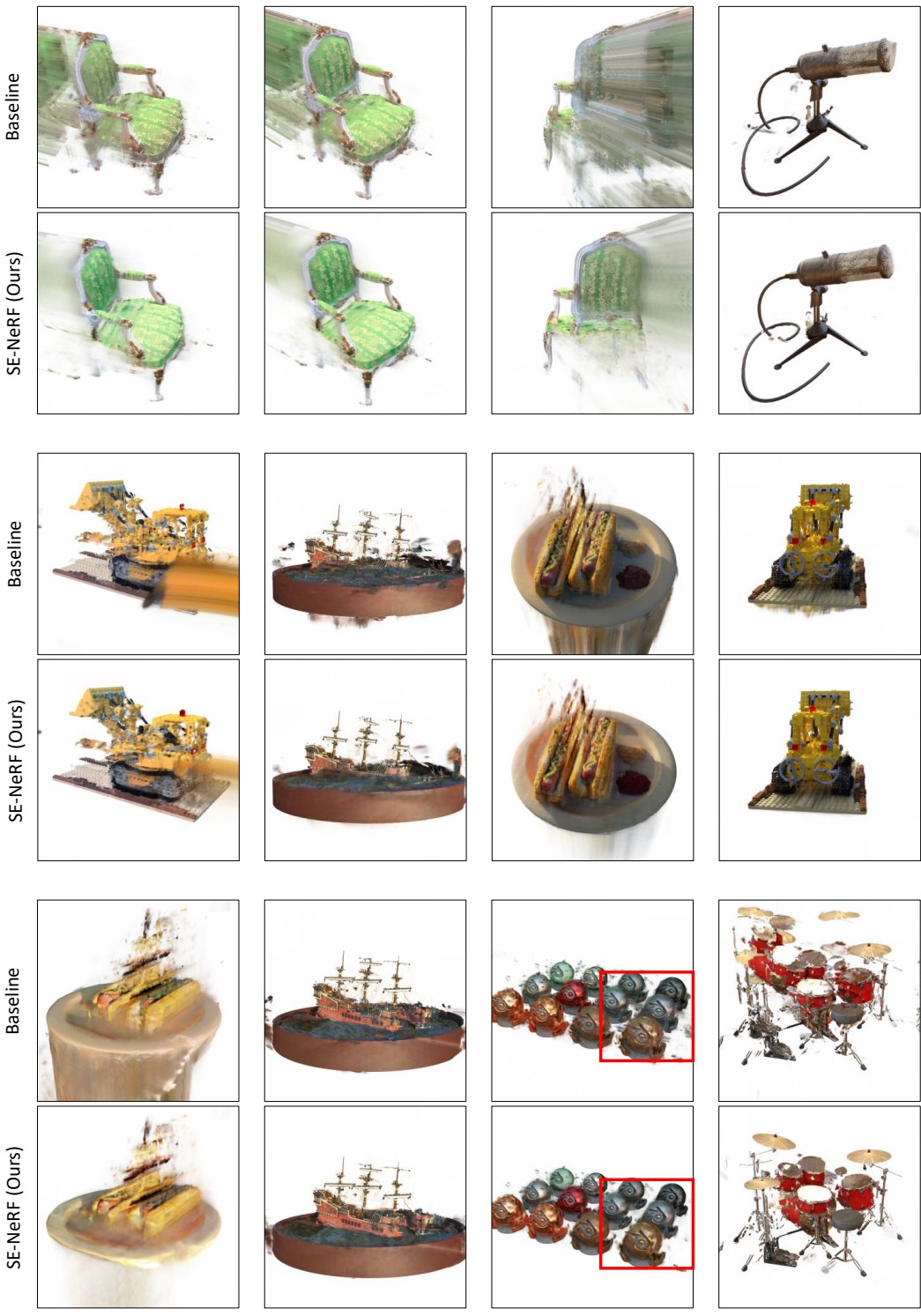

Figure 18: **Additional results in 3-view setting (NeRF Synthetic Extreme).**

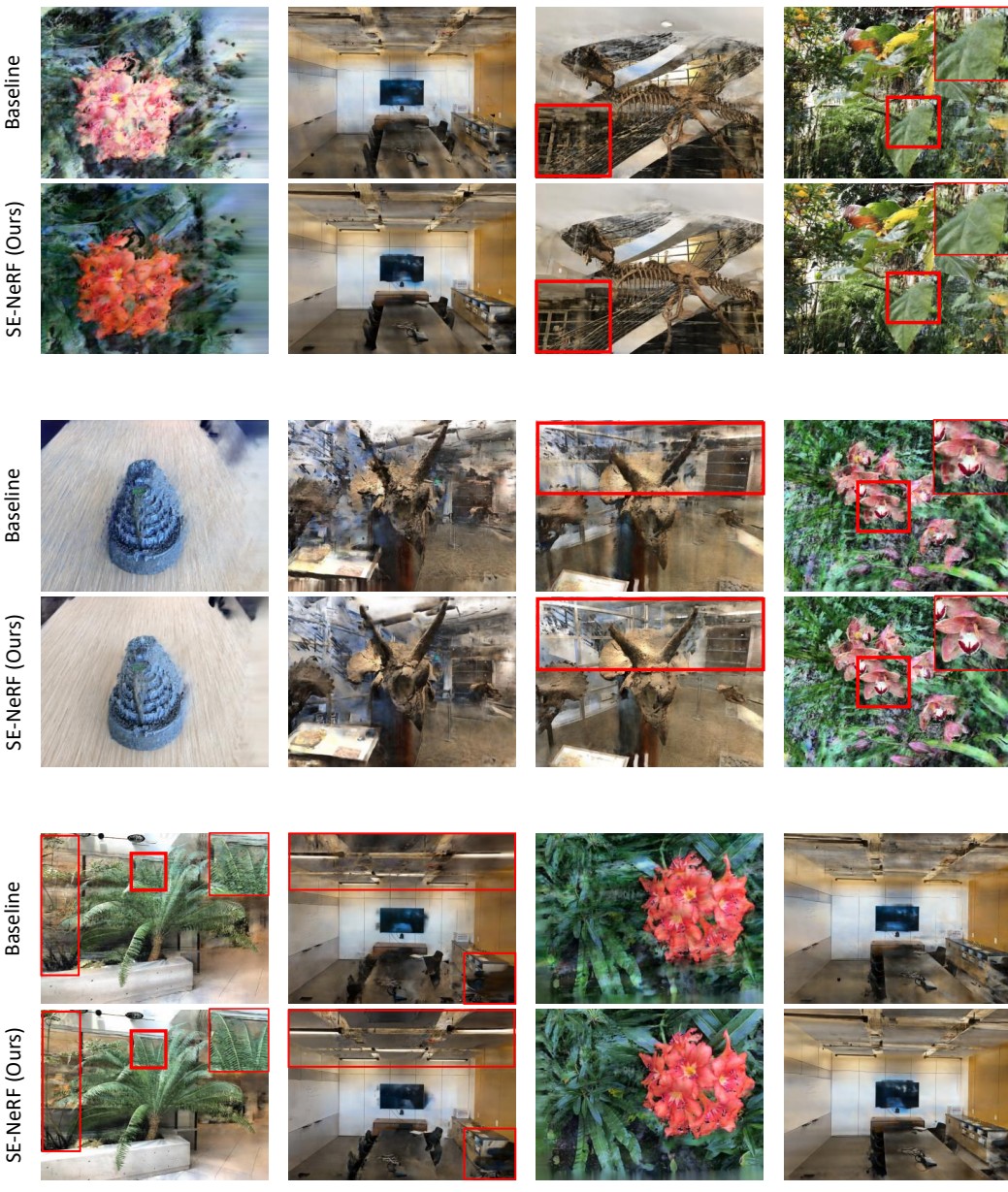

Figure 19: **Additional results in 3-view setting (LLFF).**

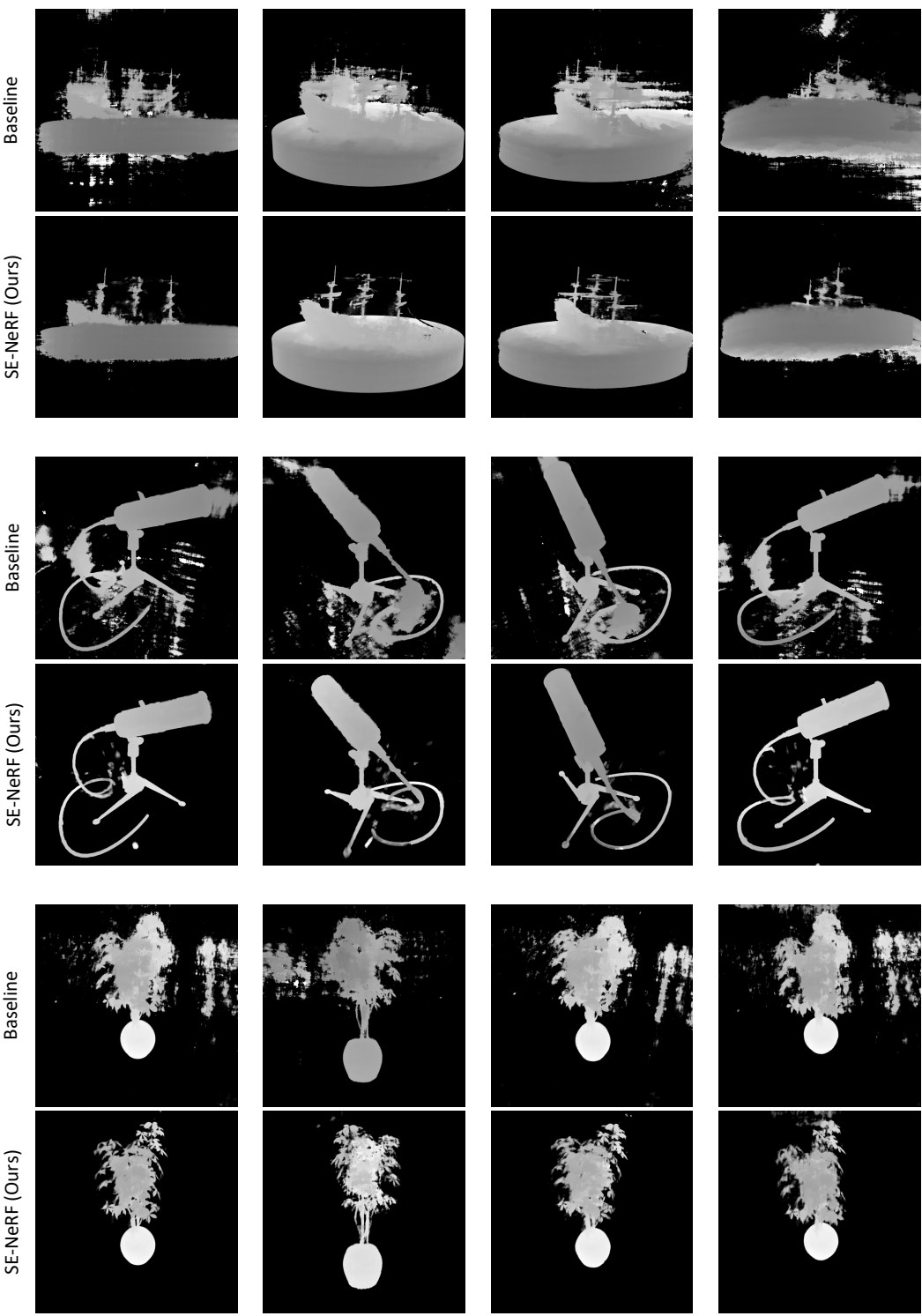

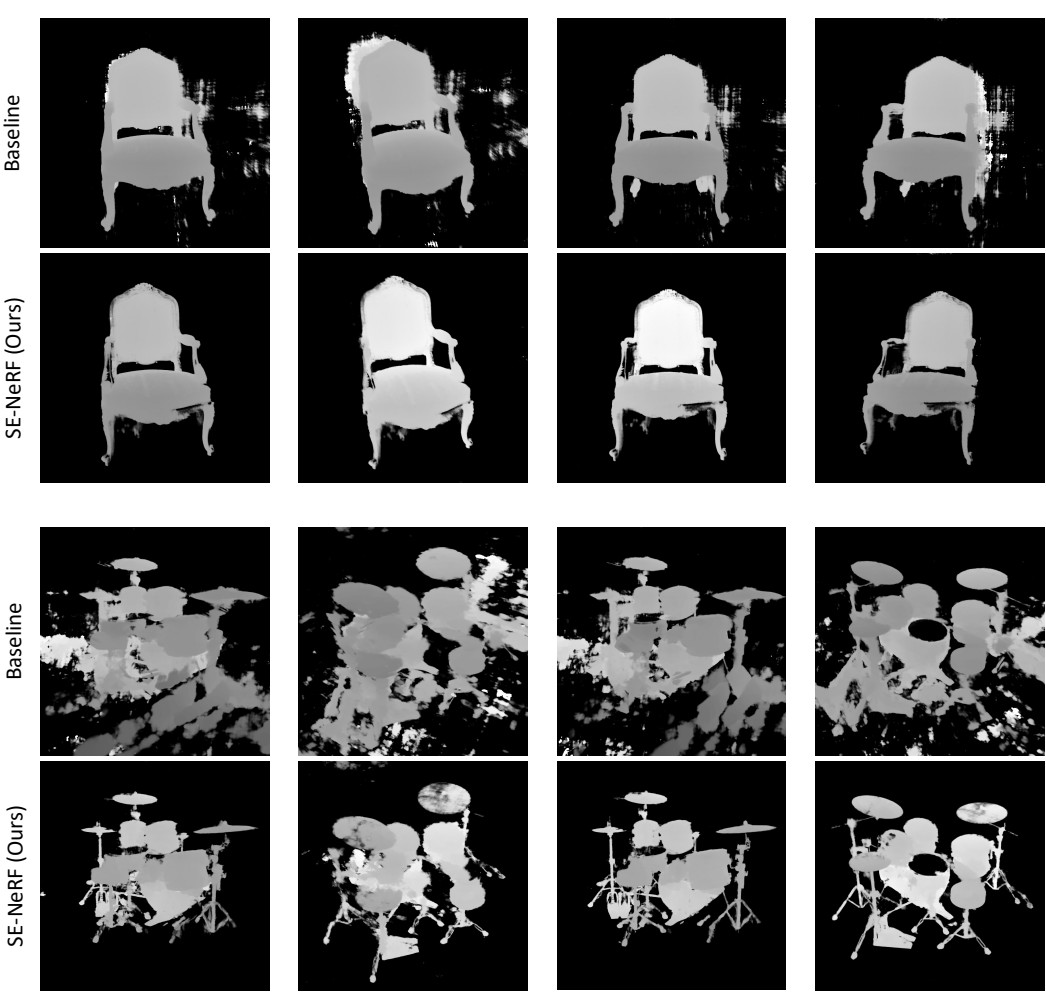

Figure 20: **Additional depth improvements in 3-view setting (NeRF Synthetic Extreme).**

