# OpenReview forum: "Self-Evolving Neural Radiance Fields"
_ICLR.cc/2024/Conference — Submitted to ICLR 2024_

### Official Review · Reviewer_qP3j · 2023-10-25

**Soundness:** 2 fair
**Presentation:** 2 fair
**Contribution:** 2 fair
**Rating:** 5
**Confidence:** 3

**Summary:**

This paper introduces Self-Evolving Neural Radiance Fields (SE-NeRF), a teacher-student framework for the few-shot Neural Radiance Fields task. The authors propose an innovative self-training approach to tackle the challenge of overfitting scenes with sparse viewpoints. The key idea is to refine the model iteratively by generating and using pseudo labels, allowing it to learn a more robust representation of the scene. A novel reliability estimation method has been proposed to identify which rays can be trusted and assist in the distillation process. The experimental results demonstrate state-of-the-art performance on multiple datasets.

**Strengths:**

- The proposal of a self-evolving framework for Neural Radiance Fields is innovative. The integration of a self-training approach into NeRF is a significant and original development.
- The quality of work appears high. The proposed method of reliability estimation for distilling reliable and unreliable rays is well thought out, and its application in a self-training framework is a noteworthy achievement.
- The paper is well-written with a clear and comprehensive explanation. The authors provided detailed derivations of their methodology, as well as thorough analysis and justification of design choices in the ablation study.
- The results show that SE-NeRF improves the quality of rendered images and achieves state-of-the-art performance, which is a meaningful contribution to the novel view synthesis and 3D reconstruction fields.

**Weaknesses:**

- The authors mention various methods for distilling the knowledge of nearby reliable rays, but the specific difference in performance between these methods is not sufficiently discussed.
- The experimental section could have benefited from additional comparisons against other few-shot learning methods or self-supervised training methods. For example, SparseNeRF [1], SimpleNeRF [2], and Vip-NeRF [3].
- The authors mention that the performance of the model on the NeRF Synthetic dataset is highly affected by the randomly selected views, which indicates an aspect of fragility in the model. This limitation could have been discussed in more detail.

[1] Wang, Guangcong, et al. "Sparsenerf: Distilling depth ranking for few-shot novel view synthesis." ICCV 2023.
[2] Somraj, Nagabhushan, Adithyan Karanayil, and Rajiv Soundararajan. "SimpleNeRF: Regularizing Sparse Input Neural Radiance Fields with Simpler Solutions." SIGGRAPH Asia 2023.
[3] Somraj, Nagabhushan, and Rajiv Soundararajan. "ViP-NeRF: Visibility Prior for Sparse Input Neural Radiance Fields." SIGGRAPH 2023.

**Questions:**

- Could the authors elaborate on their choice of VGGNet for estimating the reliability of pseudo labels? How would using another architecture affect the estimation?
- How does the method perform with larger or more complex datasets?
- Could the authors clarify how they determine that some viewpoints are reliable and others are not? How generalizable is this approach to diverse real-world cases with more pronounced viewpoint variety?
- How does the magnitude of performance improvement compare between the first iteration and subsequent iterations? Does the model reach a plateau after a certain number of iterations?

---

> ### Author Response · Authors · 2023-11-17
> **Response to Reviewer qP3j (1/5)**
>
> **General reply:**
>
> Thank you for your positive feedback on our work! We are so grateful for your opinion of our work as presenting an innovative self-training approach for training NeRF in sparse view settings, which is a meaningful contribution to the novel view synthesis and 3D reconstruction. Also, we are thankful for your judgment that our paper is well-written with a clear and comprehensive explanation, including high-quality works. We have carefully reviewed your concerns and provided detailed explanations below. Please let us know if you have any further questions or suggestions.
>
> ### Weakness1. Performance difference between various distilling methods
>
> Thank you for your suggestion of the need for additional discussion of differences between various methods for distilling the knowledge of nearby reliable rays. Following works in self-supervised training[1][2][3][4][5], we have **done multiple experiments to decide how the threshold should be determined.** Specifically, we use fixed thresholding, adaptive thresholding, and a unified equation for the distillation. The difference in performance between these methods is shown in **Table 4. Thresholding ablation of our paper.** The results of this experiment show that the adaptive thresholding method results in the most performance gain, which is used as the default method in our framework. We also made a more specific explanation and thorough analysis for each of the methods in Section B.3 of the supplementary materials. We hope this addresses your concern.
>
> [1] Gokhan Tur, Dilek Hakkani-Tür, and Robert E Schapire. Combining active and semi-supervised learning for spoken language understanding. Speech Communication, 45(2):171–186, 2005.
>
> [2] Yoshua Bengio, Jérôme Louradour, Ronan Collobert, and Jason Weston. Curriculum learning. In Proceedings of the 26th annual international conference on machine learning, pp. 41–48, 2009.
>
> [3] Paola Cascante-Bonilla, Fuwen Tan, Yanjun Qi, and Vicente Ordonez. Curriculum labeling: Revisiting pseudolabeling for semi-supervised learning. In Proceedings of the AAAI conference on artificial intelligence, volume 35, pp. 6912–6920, 2021.
>
> [4] Bowen Zhang, Yidong Wang, Wenxin Hou, Hao Wu, Jindong Wang, Manabu Okumura, and Takahiro Shinozaki. Flexmatch: Boosting semi-supervised learning with curriculum pseudo labeling. Advances in Neural Information Processing Systems, 34:18408–18419, 2021.
>
> [5] Hao Chen, Ran Tao, Yue Fan, Yidong Wang, Jindong Wang, Bernt Schiele, Xing Xie, Bhiksha Raj, and Marios Savvides. Softmatch: Addressing the quantity-quality trade-off in semi-supervised learning. 2023.

---

> ### Author Response · Authors · 2023-11-17
> **Response to Reviewer qP3j (2/5)**
>
> ### Weakness2. Additional comparisons against other few-shot learning methods or self-supervised training methods. For example, SparseNeRF [2], SimpleNeRF [3], and Vip-NeRF [4]
>
> Thank you for your suggestions! We would like to first point out that our work tackles the task of training NeRF with a limited number of views **where sparse SfM [1] depth points are not available**. This is due to the fact that SfM algorithms generally require sufficient images of large overlapping images to successfully reconstruct the 3D scene and estimate the camera poses. Therefore, we consider the setting where external sensors or a pre-calibrated fixed camera array are employed to obtain accurate camera poses of the limited number of views, which is used from prior works in multiple settings such as virtual or augmented reality, autonomous driving, and few-shot NeRF. In contrast, the works you have mentioned, such as SparseNeRF [2], SimpleNeRF [3], and Vip-NeRF [4] are methods relying on additional depth supervision achieved by the sparse point clouds generated from the SfM process or incorporating external prior models such as DPT [5]. However, training external models requires extra long training time with the need for sufficient data and also suffers from performance degradation when the scenes are not included in the domain of the training data. We would like to emphasize that our work is aligned with the prior works(InfoNeRF [6], DietNeRF [7], RegNeRF [8]) that solve the task of **training NeRF with sparse views without any depth information,** and the works you have mentioned are aligned with the prior works(DS-NeRF[9] that solve the task of **training NeRF with sparse views with additional depth information which will guide the network with accurate geometry information.**
>
> However, we agree that extra comparison with such results would help us in demonstrating our models’ performance, and as ViP-NeRF provides the results of their model trained solely using the visibility prior mask and not the sparse point clouds on the LLFF dataset using three views, we have made a comparison with their results. We show that the sparse depth supervision of Vip-NeRF is crucial for their performance.
>
> | $\text{Method}$ | $\text{Average}$ | $\text{fern}$ | $\text {orchids}$ | $\text{horns}$ | $\text{leaves}$ | $\text{trex}$ | $\text{room}$ | $\text{fortress}$ | $\text{flower}$ |
> | --- | --- | --- | --- | --- | --- | --- | --- | --- | --- |
> | $\text{ViP-NeRF w/o sparse depth}$ | $17.71$ | $16.31$  | $14.37$ | $16.34$ | $15.12$ | $16.60$ | $19.21$ | $23.66$ | $18.15$ |
> | $\text{ViP-NeRF with sparse depth}$ | $18.92$ | $17.49$  | $14.24$ | $18.27$ | $12.61$ | $18.16$ | $21.97$ | $24.12$ | $20.82$ |
> | $\text{SE-NeRF (NeRF) w/o sparse depth}$ | $18.10$ | $19.70$  | $14.10$ | $17.92$ | $15.46$ | $15.53$ | $19.03$ | $22.86$ | $20.16$ |
>
> [1] Johannes L Schonberger and Jan-Michael Frahm. Structure-from-motion revisited. In Proceedings of the IEEE conference on computer vision and pattern recognition, pp. 4104–4113, 2016.
>
> [2] Guangcong Wang, Zhaoxi Chen, Chen Change Loy, and Ziwei Liu. Sparsenerf: Distilling depth ranking for few-shot novel view synthesis. In *Proceedings of the IEEE/CVF International Conference on Computer Vision (ICCV)*, pp. 9065–9076, October 2023a.
>
> [3] Nagabhushan Somraj, Adithyan Karanayil, and Rajiv Soundararajan. SimpleNeRF: Regularizing sparse input neural radiance fields with simpler solutions. In *SIGGRAPH Asia*, December 2023. doi: 10.1145/3610548.3618188.
>
> [4] Nagabhushan Somraj and Rajiv Soundararajan. Vip-nerf: Visibility prior for sparse input neural radiance fields. In *ACM SIGGRAPH 2023 Conference Proceedings*, SIGGRAPH ’23, New York, NY, USA, 2023. Association for Computing Machinery. ISBN 9798400701597. doi: 10.1145/3588432.3591539. URL https://doi.org/10.1145/3588432.3591539.
>
> [5] René Ranftl, Alexey Bochkovskiy, and Vladlen Koltun. Vision transformers for dense prediction. In , pp. 12179–12188, October 2021.
>
> [6] Mijeong Kim, Seonguk Seo, and Bohyung Han. Infonerf: Ray entropy minimization for few-shot neural volume rendering. In *Proceedings of the IEEE/CVF Conference on Computer Vision and Pattern Recognition*, pp. 12912–12921, 2022.
>
> [7] Ajay Jain, Matthew Tancik, and Pieter Abbeel. Putting nerf on a diet: Semantically consistent few-shot view synthesis. In Proceedings of the IEEE/CVF International Conference on Computer Vision, pp. 5885–5894, 2021.
>
> [8] Michael Niemeyer, Jonathan T Barron, Ben Mildenhall, Mehdi SM Sajjadi, Andreas Geiger, and Noha Radwan. Regnerf: Regularizing neural radiance fields for view synthesis from sparse inputs. In *Proceedings of the IEEE/CVF Conference on Computer Vision and Pattern Recognition*, pp. 5480–5490, 2022.
>
> [9] Kangle Deng, Andrew Liu, Jun-Yan Zhu, and Deva Ramanan. Depth-supervised nerf: Fewer views and faster training for free. In *Proceedings of the IEEE/CVF Conference on Computer Vision and Pattern Recognition*, pp. 12882–12891, 2022.

---

> ### Author Response · Authors · 2023-11-17
> **Response to Reviewer qP3j (3/5)**
>
> ### Weakness3. The performance of the model on the NeRF Synthetic dataset is highly affected by the randomly selected views
>
> Thank you for your review. We appreciate your feedback and would like to address your concern regarding the fragility of the NeRF model on the NeRF Synthetic dataset. As you have pointed out, the performance of the model is highly dependent on the randomly selected views. However, we would like to clarify that the **problem is generally shared among all NeRF-like models not only ours**, especially when dealing with sparse views in complex scene settings. Thus, we have taken great care to minimize the effect of this problem in our experiments. Specifically, we randomly select views in each scene and keep them fixed while evaluating all comparison models. This ensures that our **“score improvement” is independent of the fluctuating effect caused by random view selection**. Additionally, **we introduce a novel evaluation protocol, "NeRF Synthetic Extreme,**" specifically tailored for evaluating the robustness of few-shot NeRF scenarios. In this protocol, we deliberately select the most challenging three views for each scene, strategically placing them in locations that have access to only limited information about the entire scene. This intentional selection aims to provide accurate reconstruction information for only the limited viewpoints while providing only partial information for the overall scene, creating a situation where models are prone to overfitting. Consequently, the protocol offers a fair environment for testing how well each model overcomes the overfitting challenge.
>
> We apologize for any ambiguity in the paper, and we will upload a revised paper that clarifies the ambiguity. We hope that this explanation addresses your concern and clarifies our approach to dealing with the fragility of the NeRF model.

---

> ### Author Response · Authors · 2023-11-17
> **Response to Reviewer qP3j (4/5)**
>
> ### Q1. Selecting feature extractor
>
> We appreciate the reviewer’s interest in our work and their insightful question. In our study, **we utilized pixel-level feature maps** to evaluate the reliability of pseudo labels. As a result, **any architectural framework** that can provide **pixel-level feature maps** can be seamlessly integrated into our proposed methodology. Traditionally, 2D CNN is well-known for the expressiveness of its features, which is the reason it has been adopted as a feature extractor in various works [1][2][3][4]. Following prior works, we selected VGG-19 [5] as our default feature extractor **after conducting experiments** employing different feature extractors such as VGG-19, ResNet50 [6], and U-Net [7] in our framework.
>
> For VGG-19 and U-Net, we selected the extracted feature map prior to the first 4 pooling layers, whose dimensions are HxW, H/2xW/2, H/4xW/4, and H/8xW/8. For ResNet50, we selected the feature maps prior to the first 3 pooling layers, whose dimensions are H/2xW/2, H/4xW/4, and H/8xW/8. They are upsampled to HxW using bilinear interpolation and then concatenated to form latent vectors aligned to each pixel. All cases resulted in noticeable performance enhancements across the networks under consideration. For our framework, we selected the model that results in the highest performance improvement among those. The result of experiments we have done using different feature extractors is attached.
>
> | $\text{Methods}$ | $\text{Average}$ | $\text{chair}$ | $\text {drums}$ | $\text{ficus}$ | $\text{hotdog}$ | $\text{lego}$ | $\text{mater.}$ | $\text{ship}$ | $\text{mic}$ |
> | --- | --- | --- | --- | --- | --- | --- | --- | --- | --- |
> | $\text{K-Planes}$ | $15.45$ | $15.61$ | $13.23$ | $18.29$ | $12.45$ | $14.67$ | $16.30$ | $13.35$ | $19.74$ |
> | $\text{SE-NeRF (K-Planes) with U-Net}$ | $17.25$$\scriptsize\color{green}(+1.80)$ | $19.81$$\scriptsize\color{green}(+4.20)$ | $13.37$$\scriptsize\color{green}(+0.14)$ | $18.33$$\scriptsize\color{green}(+0.04)$ | $20.19$$\scriptsize\color{green}(+7.74)$ | $16.29$$\scriptsize\color{green}(+1.62)$ | $16.74$$\scriptsize\color{green}(+0.44)$ | $13.47$$\scriptsize\color{green}(+0.12)$ | $19.78$$\scriptsize\color{green}(+0.04)$ |
> | $\text{SE-NeRF (K-Planes) with ResNet50}$ | $17.39$$\scriptsize\color{green}(+1.94)$ | $19.96$$\scriptsize\color{green}(+4.35)$ | $13.50$$\scriptsize\color{green}(+0.27)$ | $18.42$$\scriptsize\color{green}(+0.13)$ | $20.36$$\scriptsize\color{green}(+7.91)$ | $16.28$$\scriptsize\color{green}(+1.61)$ | $16.89$$\scriptsize\color{green}(+0.59)$ | $13.96$$\scriptsize\color{green}(+0.61)$ | $19.77$$\scriptsize\color{green}(+0.03)$ |
> | $\text{SE-NeRF (K-Planes) with VGG-19}$ | $17.49$$\scriptsize\color{green}(+2.04)$ | $20.54$ $\scriptsize\color{green}(+4.93)$ | $13.38$ $\scriptsize\color{green}(+0.15)$ | $18.33$ $\scriptsize\color{green}(+0.04)$ | $20.14$ $\scriptsize\color{green}(+7.69)$ | $16.65$ $\scriptsize\color{green}(+1.98)$ | $17.01$ $\scriptsize\color{green}(+0.71)$ | $13.72$ $\scriptsize\color{green}(+0.37)$ | $20.13$ $\scriptsize\color{green}(+0.39)$ |
>
> [1] Qianqian Wang, Zhicheng Wang, Kyle Genova, Pratul P Srinivasan, Howard Zhou, Jonathan T Barron, Ricardo Martin-Brualla, Noah Snavely, and Thomas Funkhouser. Ibrnet: Learning multi-view image-based rendering. In Proceedings of the IEEE/CVF Conference on Computer Vision and Pattern Recognition, pp. 4690–4699, 2021.
>
> [2] Jingyang Zhang, Yao Yao, and Long Quan. Learning signed distance field for multi-view surface reconstruction. In Proceedings of the IEEE/CVF International Conference on Computer Vision, pp. 6525–6534, 2021b.
>
> [3] Alex Yu, Vickie Ye, Matthew Tancik, and Angjoo Kanazawa. pixelnerf: Neural radiance fields from one or few images. In Proceedings of the IEEE/CVF Conference on Computer Vision and Pattern Recognition, pp. 4578–4587, 2021.
>
> [4] Minseop Kwak, Jiuhn Song, and Seungryong Kim. Geconerf: Few-shot neural radiance fields via geometric consistency. arXiv preprint arXiv:2301.10941, 2023.
>
> [5] Karen Simonyan and Andrew Zisserman. Very deep convolutional networks for large-scale image recognition. *arXiv preprint arXiv:1409.1556*, 2014.
>
> [6] Kaiming He, Xiangyu Zhang, Shaoqing Ren, and Jian Sun. Deep residual learning for image recognition. In Proceedings of the IEEE conference on computer vision and pattern recognition, pp. 770–778, 2016.
>
> [7] Olaf Ronneberger, Philipp Fischer, and Thomas Brox. U-net: Convolutional networks for biomedical image
> segmentation. In Nassir Navab, Joachim Hornegger, William M. Wells, and Alejandro F. Frangi (eds.),
> Medical Image Computing and Computer-Assisted Intervention – MICCAI 2015, pp. 234–241, Cham, 2015.
> Springer International Publishing. ISBN 978-3-319-24574-4.

---

> ### Author Response · Authors · 2023-11-17
> **Response to Reviewer qP3j (5/5)**
>
> ### Q3. Selecting Rendering View and Application to Real-World Images
>
> Thank you for sharing your concerns. We apologize for any confusion caused by our incomplete explanation of how we select rendering viewpoints within our framework. We select rendering viewpoints by perturbing the views given for the sparse inputs. Although any random perturbation to the views is possible, we find that initially selecting rendering viewpoints from regions near known viewpoints and then selecting rendering viewpoints further from the known viewpoints in a progressive manner acts most robustly, as the probability of containing reliable rays is higher in adjacent regions. By repeating this process iteratively, we increase the range of the training view to contain larger regions of the scene. **As the assumption of close regions containing more reliable regions is very natural in real-world cases**, we believe that this type of approach can be seamlessly applied to real-world images. An additional explanation can be found in **Figure 12. of Section B.4 in the supplementary materials.** We hope that this revised explanation is more clear and helpful.
>
> ### Q2. Applying to a large dataset
>
> Thank you for your interest in our work! As the current baseline models lack the capacity of modeling large scenes such as [1] and [2], we did not conduct an experiment on these datasets in our paper. However, as existing large-scale NeRFs [3,4] also utilize ray-based volume rendering, **our framework can also be easily applied to these settings.** Also, our proposed rendering view selection of first selecting rendering views near the known viewpoints will also result in the rendering views containing reliable regions and therefore **our distillation scheme can be similarly applied.**
>
> ### Q4. Convergence
>
> In general, we observe the most significant performance improvement during the first to second and second to third iterations, after which the model tends to converge. However, since the magnitude of performance improvement per iteration under our framework is i**nfluenced by the underlying architecture and its own training strategy**, there is a slight difference according to the adopted baseline models. For instance, the $K$-Planes [5] baseline, which learns rapidly in relatively fewer steps than vanilla NeRF [6], accounts for 80% of the overall performance improvement within just the first two iterations.
>
> [1] Andreas Geiger, Philip Lenz, and Raquel Urtasun. Are we ready for autonomous driving? the kitti vision benchmark suite. In 2012 IEEE conference on computer vision and pattern recognition, pp. 3354–3361. IEEE, 2012.
>
> [2] Marius Cordts, Mohamed Omran, Sebastian Ramos, Timo Rehfeld, Markus Enzweiler, Rodrigo Benenson, Uwe Franke, Stefan Roth, and Bernt Schiele. The cityscapes dataset for semantic urban scene understanding. In Proceedings of the IEEE conference on computer vision and pattern recognition, pp. 3213–3223, 2016.
>
> [3] Matthew Tancik, Vincent Casser, Xinchen Yan, Sabeek Pradhan, Ben Mildenhall, Pratul P Srinivasan, Jonathan T Barron, and Henrik Kretzschmar. Block-nerf: Scalable large scene neural view synthesis. In Proceedings of the IEEE/CVF Conference on Computer Vision and Pattern Recognition, pp. 8248–8258, 2022.
>
> [4] Yuanbo Xiangli, Linning Xu, Xingang Pan, Nanxuan Zhao, Anyi Rao, Christian Theobalt, Bo Dai, and Dahua
> Lin. Bungeenerf: Progressive neural radiance field for extreme multi-scale scene rendering. In European
> conference on computer vision, pp. 106–122. Springer, 2022.
>
> [5] Sara Fridovich-Keil, Giacomo Meanti, Frederik Rahbæk Warburg, Benjamin Recht, and Angjoo Kanazawa. K-planes: Explicit radiance fields in space, time, and appearance. In CVPR, 2023.
>
> [6] Ben Mildenhall, Pratul P Srinivasan, Matthew Tancik, Jonathan T Barron, Ravi Ramamoorthi, and Ren Ng. Nerf: Representing scenes as neural radiance fields for view synthesis. Communications of the ACM, 65(1): 99–106, 2021.

---

> > ### Comment · Reviewer_qP3j · 2023-11-22
> > **Lower rating**
> >
> > Thank you so much for addressing my concerns. However, after reading the comments from other reviewers and corresponding responses, I would like to lower my rating to * marginally below the acceptance threshold*. Specifically, due to the approach's similarity to existing works like Self-NeRF, though the authors argue distinct features in their methodology. Additionally, some parts lack details, particularly in handling unreliable rays and the absence of camera correction in sparse views.

---

> > > ### Author Response · Authors · 2023-11-22
> > > **Major differences with existing work**
> > >
> > > Thank you for your reply and for raising your concern about a similar work Self-NeRF [1]. We would like to emphasize that our work is **concurrent** to their work as their work is an **unpublished arxiv paper unable to make any experimental comparisons with their codes being unreleased**. Additionally, we would like to re-emphasize some **major key differences** in our work with Self-NeRF.
> > >
> > > Although the **concept** of using self-training (making the student the new teacher after every iteration) is similar, Self-NeRF and our work make entirely different approaches in terms of generalizability. They design a **completely** **new architecture** **tailored for** modeling the confidence of the scene(adding a new branch), l**imiting their applications to other existing methods** such as $K$-Planes [2], Vanilla NeRF [3], and Instant-NGP [4]. On the other hand, our approach is more flexible and versatile. We design **an explicit module** that can estimate the reliability of unseen rays. The advantage of our design is that it can be **seamlessly integrated with any existing ray-based volume rendering models**, thereby enhancing their performance **without the need for significant architectural modifications**. This makes our method a more **universally applicable solution**, offering improvements to a wide range of existing models.
> > >
> > > Additionally, our novel method of distilling the knowledge of reliable and unreliable rays to the student networks **has never been proposed in existing works** demonstrating the novelty and contribution of our work. The importance of this ray-based distillation method has been comprehensively **covered in Section 5.3 of our paper**. Furthermore, the feature-based reliability estimation is a much **more robust method** compared to using the warped image directly as the scene becomes closer to real-world images, **including reflectant non-Lambertian surfaces** [5], which makes our method **more generally applicable to any scene** regardless of the lighting condition.
> > >
> > > In summary, we ensure an **incomparable level of generalizability** in terms of applying existing models and adapting to various scenes compared to Self-NeRF. We are confident in our work’s contribution, and we would really appreciate it if you could understand the key contributions and differences of our work.
> > >
> > > [1] Jiayang Bai, Letian Huang, Wen Gong, Jie Guo, and Yanwen Guo. Self-nerf: A self-training pipeline for few-shot neural radiance fields. arXiv preprint arXiv:2303.05775, 2023.
> > >
> > > [2] Sara Fridovich-Keil, Giacomo Meanti, Frederik Rahbæk Warburg, Benjamin Recht, and Angjoo Kanazawa. K-planes: Explicit radiance fields in space, time, and appearance. In CVPR, 2023.
> > >
> > > [3] Ben Mildenhall, Pratul P Srinivasan, Matthew Tancik, Jonathan T Barron, Ravi Ramamoorthi, and Ren Ng. Nerf: Representing scenes as neural radiance fields for view synthesis. Communications of the ACM, 65(1): 99–106, 2021.
> > >
> > > [4] Thomas Müller, Alex Evans, Christoph Schied, and Alexander Keller. Instant neural graphics primitives with a multiresolution hash encoding. ACM Transactions on Graphics (ToG), 41(4):1–15, 2022.
> > >
> > > [5] Huangying Zhan, Ravi Garg, Chamara Saroj Weerasekera, Kejie Li, Harsh Agarwal, and Ian Reid. Unsupervised learning of monocular depth estimation and visual odometry with deep feature reconstruction. In Proceedings of the IEEE conference on computer vision and pattern recognition, pp. 340–349, 2018.

---

> ### Author Response · Authors · 2023-11-22
> **Additional concerns**
>
> Thank you for sharing your unresolved concerns about the lack of details, particularly in handling unreliable rays and the absence of camera correction in sparse views.
>
> **Q1**. Lack of details in handling unreliable rays
>
> We are sorry for any ambiguity or unclear explanations in our initial manuscript. We have currently uploaded a revised version of our manuscript, including a detailed explanation of how the unreliable rays are distilled. The **specific explanation can be found in Section B.3 of the supplementary materials**. We would really appreciate it if you could spend your time reviewing our revised manuscript.
>
> **Q2**. Absence of camera correction in sparse views.
>
> We would like to re-emphasize that there are two lines of research in training NeRF with sparse views. 1. **Sparse views with accurate poses**(DietNeRF[1], InfoNeRF[2], RegNeRF[3]).  2. **Sparse views with noisy poses**(Sparf[4]). We **follow prior works of few-shot NeRF that tackle the task of training NeRF with sparse views with accurate poses**(DietNeRF[1], InfoNeRF[2], RegNeRF[3]) and make a quantitative and qualitative comparison with the prior works in our manuscript. Additionally, Sparf incorporates a camera correction strategy by assuming their initial given camera poses are noisy by adding noise to the ground truth camera poses. After the step of optimizing noisy camera poses, they **compare the refined camera pose to the initial ground truth camera poses** to evaluate how well the camera poses have been corrected. As our work and prior works tackling the task of training NeRF with **sparse views with accurate poses** start with the accurate poses, we would like to clarify that **there is no need for any camera pose correction in this line of research.**
>
> We hope that we have resolved your concerns, and if you have any remaining questions please feel free to ask, and we will get back to you as soon as possible.
>
> [1] Ajay Jain, Matthew Tancik, and Pieter Abbeel. Putting nerf on a diet: Semantically consistent few-shot view synthesis. In Proceedings of the IEEE/CVF International Conference on Computer Vision, pp. 5885–5894, 2021.
>
> [2] Mijeong Kim, Seonguk Seo, and Bohyung Han. Infonerf: Ray entropy minimization for few-shot neural volume rendering. In *Proceedings of the IEEE/CVF Conference on Computer Vision and Pattern Recognition*, pp. 12912–12921, 2022.
>
> [3] Michael Niemeyer, Jonathan T Barron, Ben Mildenhall, Mehdi SM Sajjadi, Andreas Geiger, and Noha Radwan. Regnerf: Regularizing neural radiance fields for view synthesis from sparse inputs. In *Proceedings of the IEEE/CVF Conference on Computer Vision and Pattern Recognition*, pp. 5480–5490, 2022.
>
> [4] Prune Truong, Marie-Julie Rakotosaona, Fabian Manhardt, and Federico Tombari. Sparf: Neural radiance fields from sparse and noisy poses. In Proceedings of the IEEE/CVF Conference on Computer Vision and Pattern Recognition (CVPR), pp. 4190–4200, June 2023.

---

### Official Review · Reviewer_BBPH · 2023-10-29

**Soundness:** 3 good
**Presentation:** 3 good
**Contribution:** 1 poor
**Rating:** 3
**Confidence:** 3

**Summary:**

This paper introduces a teacher-student network designed to address the neural rendering task using a limited set of training view images. The authors present a technique for estimating ray reliability. Utilizing this estimation, they perform ray distillation and subsequently iterate the process multiple times, augmenting the training view images with pseudo ground truth.

**Strengths:**

1. The teacher-student idea for few-shot NeRF can inspire many follow-up works.
2. The authors provide many detailed analysis in the appendix, which can help understand the method better.

**Weaknesses:**

1. The method of estimating ray reliability raises concerns. The authors use a 2D-CNN to obtain super-pixel features, meaning that the features for similarity computation are derived from a pixel region, not individual pixels. This approach seems misaligned with the NeRF setting, which relies only on per-pixel information instead of per-region data. Such a discrepancy makes the ray reliability estimation seem less accurate and contradicts the proposed method.

2. There's a noticeable absence of an ablation study for the unreliable ray distillation. The proposed method for this distillation isn't entirely convincing. An ablation study demonstrating the effects of removing this module would have been beneficial in validating its effectiveness.

3. The performance improvement appears to be marginal. According to Table 1, the PSNR value for the NeRF Synthetic dataset and NeRF base model only saw a slight enhancement from 19.38 to 20.53. Given this quality level, the baseline model's performance is somewhat volatile, potentially fluctuating between 2-4dB with changes like altering the selected view. The paper itself even acknowledges this. A more significant achievement would have been, for example, if the authors could elevate the performance from 30dB to 32dB.

4. The research lacks comparisons to other few-shot NeRF models. Several methods, including PixelNeRF, MSVNeRF, and IBRNet, address the few-shot neural rendering challenge. Integrating the proposed method with these established works would have provided valuable insights into its effectiveness. Unfortunately, such experiments are absent.

5. There's a discrepancy in the results presented in two different tables. For the LLFF dataset, 3-view setting, and the K-plane baseline model, outcomes are documented in both Table 1 and Table 7. While the results for SE-NeRF(K-Planes) are consistent across the two tables (e.g., 16.30dB for the PSNR metric), the baseline model's results differ: Table 1 shows 14.18dB, while Table 7 indicates 15.77dB. This inconsistency raises questions about the data's accuracy.

**Questions:**

Please see the weaknesses above.

---

> ### Author Response · Authors · 2023-11-17
> **Response to Reviewer BBPH (1/3)**
>
> **General reply:** Thank you for your constructive review and helpful suggestions! We give a detailed response to your questions and comments below. If any of our responses do not adequately address your concerns, please let us know, and we will get back to you as soon as possible.
>
> ### Weakness1. Estimating ray reliability using a 2D-CNN
>
> Thank you for your valuable feedback. We appreciate your concerns regarding the method of estimating ray reliability. We agree that the feature map our framework relies on should contain pixel-level information, not pixel-region information. We would like to clarify that we **use a pixel-level feature map**. Specifically, feature maps are extracted prior to the first 4 pooling layers, upsampled using bilinear interpolation, and concatenated to form **latent vectors of size of the original image.** With the process of upsampling feature maps extracted from different levels of the 2D CNN network, **we can achieve a pixel-level feature map.** More details can be found in Section B.2 of the supplementary materials. Please note that we qualitatively demonstrate that our feature map operates on pixel-level in Figure 7 and Figure 8.
>
> In addition, the process of making pixel-level feature maps by upsampling different sizes of feature maps has been leveraged in pixelNeRF[1], successfully learning the prior of 2D features and rendering novel viewpoints in the task of Generalized NeRF settings. We also show a comparison of performance using U-Net[2] as the feature extractor backbone, which is designed to extract a pixel-level feature map, and VGG-19[3], which is used as our default feature extractor.
>
> | $\text{Methods}$ | $\text{Average}$ | $\text{chair}$ | $\text {drums}$ | $\text{ficus}$ | $\text{hotdog}$ | $\text{lego}$ | $\text{mater.}$ | $\text{ship}$ | $\text{mic}$ |
> | --- | --- | --- | --- | --- | --- | --- | --- | --- | --- |
> | $\text{K-Planes}$ | $15.45$ | $15.61$ | $13.23$ | $18.29$ | $12.45$ | $14.67$ | $16.30$ | $13.35$ | $19.74$ |
> | $\text{SE-NeRF (K-Planes) with U-Net}$ | $17.25$$\scriptsize\color{green}(+1.80)$ | $19.81$$\scriptsize\color{green}(+4.20)$ | $13.37$$\scriptsize\color{green}(+0.14)$ | $18.33$$\scriptsize\color{green}(+0.04)$ | $20.19$$\scriptsize\color{green}(+7.74)$ | $16.29$$\scriptsize\color{green}(+1.62)$ | $16.74$$\scriptsize\color{green}(+0.44)$ | $13.47$$\scriptsize\color{green}(+0.12)$ | $19.78$$\scriptsize\color{green}(+0.04)$ |
> | $\text{SE-NeRF (K-Planes) with VGG-19}$ | $17.49$$\scriptsize\color{green}(+2.04)$ | $20.54$ $\scriptsize\color{green}(+4.93)$ | $13.38$ $\scriptsize\color{green}(+0.15)$ | $18.33$ $\scriptsize\color{green}(+0.04)$ | $20.14$ $\scriptsize\color{green}(+7.69)$ | $16.65$ $\scriptsize\color{green}(+1.98)$ | $17.01$ $\scriptsize\color{green}(+0.71)$ | $13.72$ $\scriptsize\color{green}(+0.37)$ | $20.13$ $\scriptsize\color{green}(+0.39)$ |
>
> [1] Alex Yu, Vickie Ye, Matthew Tancik, and Angjoo Kanazawa. pixelnerf: Neural radiance fields from one or few images. In Proceedings of the IEEE/CVF Conference on Computer Vision and Pattern Recognition, pp. 4578–4587, 2021.
>
> [2] Olaf Ronneberger, Philipp Fischer, and Thomas Brox. U-net: Convolutional networks for biomedical image
> segmentation. In Nassir Navab, Joachim Hornegger, William M. Wells, and Alejandro F. Frangi (eds.),
> Medical Image Computing and Computer-Assisted Intervention – MICCAI 2015, pp. 234–241, Cham, 2015.
> Springer International Publishing. ISBN 978-3-319-24574-4.
>
> [3] Karen Simonyan and Andrew Zisserman. Very deep convolutional networks for large-scale image recognition. *arXiv preprint arXiv:1409.1556*, 2014.

---

> ### Author Response · Authors · 2023-11-17
> **Response to Reviewer BBPH (2/3)**
>
> ### Weakness2. Ablation study for the unreliable ray distillation
>
> We appreciate your insightful question and apologize for any confusion that may have arisen. To address your concern, please refer to **Table 3** in our paper, which provides the necessary details.
>
> In this table, the method labeled as **$K$-Planes + Reliable** represents the results obtained when the **unreliable ray distillation is removed from our proposed method**. As can be observed from these results, the reliable ray distillation significantly contributes to the performance improvement of our method. Furthermore, we find that additional performance enhancements can be achieved by incorporating the unreliable ray distillation.
>
> We acknowledge that the notation used in the table may have led to some confusion, and we will make revisions to ensure clarity. We hope this addresses your concern and demonstrates the effectiveness of the unreliable ray distillation in our proposed method. Thank you for bringing this to our attention.
>
> ### Weakness3. Performance improvements appears to be marginal
>
> Thank you for sharing your concerns. We would like to emphasize that training NeRF with sparse views in a complex scene setting is inherently challenging, and the scene knowledge itself is largely dependent on view selection. This is the fundamental reason why NeRF performance fluctuates in terms of view selection. We are aware of this problem and have carefully designed our evaluation protocol to minimize its effect. Specifically, for fair comparison, we randomly select views in each scene and **keep them fixed while evaluating all comparison models**.  Therefore, **our score improvement is independent of the fluctuating effect** and can be seen as a **general improvement**. Furthermore, we propose a new evaluation protocol for fair comparison called “NeRF Synthetic Extreme,” where all views are selected from one side of the scene. While it may seem that our performance improvement is somewhat modest, it's important to note that the scale of the performance value is tenfold, as a 1db increase in PSNR indicates 10 times less noise in the image.

---

> ### Author Response · Authors · 2023-11-17
> **Response to Reviewer BBPH (3/3)**
>
> ### Weakness4. Comparisons to other few-shot NeRF models including PixelNeRF, MSVNeRF, and IBRNet
>
> Thank you for your suggestions! First, we point out that our method is a **prior-free few-shot NeRF model**, following prior works such as DietNeRF [1], InfoNeRF [2], and RegNeRF [3], which **does not utilize** any encoder networks trained across different scenes as priors for few-shot NeRF, but relies on regularization methods to successfully model a single-scene through per-scene optimization. In contrast, the works you have mentioned, such as pixelNeRF [4], IBRNet [5], MVSNet [6] are **prior-based methods** relying on the **learned generalized priors** acquired by training a feature predictor across numerous scenes for few-shot reconstruction, which introduces numerous constraints such as the requirement of large 3D training data, lengthy training time, and shows low performance when the scenes for inference are not included in the domain in of the training data. Although the comparison with prior-based methods as a pre-trained baseline in conjunction with prior-free optimization methods is possible, we would like to emphasize that these **two types of methods take an orthogonal approach** to tackle the few-shot NeRF problem.
>
> ### Weakness5. Discrepancy in the results
>
> We would like to express our gratitude for your thorough review of our experimental results. As we have mentioned in the paper, the performance of the NeRF trained with sparse views shows volatile results and for a fair comparison of multiple methods, we conducted multiple experiments in a random view setting and reported the average values. Although the values in both tables are correct, the 15.77dB in Table 7 is the results after being averaged, and the 14.18dB in Table 1 is one of the results of our experiment that was used to calculate the averaged results. We apologize for any confusion caused by the results. We will correct the values in Table 1 and upload the revised paper. The correct values in Table 1 for the LLFF 3-view setting are as follows:
>
> | $\text{Methods (LLFF)}$  | $\text{PSNR}\uparrow$ | $\text {SSIM}\uparrow$ | $\text{LPIPS}\downarrow$ | $\text{Avg.}\downarrow$ |
> | --- | --- | --- | --- | --- |
> | $\text{K-Planes}$ | $15.77$ | $0.44$ | $0.46$ | $0.41$ |
>
> [1] Ajay Jain, Matthew Tancik, and Pieter Abbeel. Putting nerf on a diet: Semantically consistent few-shot view synthesis. In Proceedings of the IEEE/CVF International Conference on Computer Vision, pp. 5885–5894, 2021.
>
> [2] Mijeong Kim, Seonguk Seo, and Bohyung Han. Infonerf: Ray entropy minimization for few-shot neural volume rendering. In *Proceedings of the IEEE/CVF Conference on Computer Vision and Pattern Recognition*, pp. 12912–12921, 2022.
>
> [3] Michael Niemeyer, Jonathan T Barron, Ben Mildenhall, Mehdi SM Sajjadi, Andreas Geiger, and Noha Radwan. Regnerf: Regularizing neural radiance fields for view synthesis from sparse inputs. In *Proceedings of the IEEE/CVF Conference on Computer Vision and Pattern Recognition*, pp. 5480–5490, 2022.
>
> [4] Alex Yu, Vickie Ye, Matthew Tancik, and Angjoo Kanazawa. pixelnerf: Neural radiance fields from one or few images. In Proceedings of the IEEE/CVF Conference on Computer Vision and Pattern Recognition, pp. 4578–4587, 2021.
>
> [5] Qianqian Wang, Zhicheng Wang, Kyle Genova, Pratul P Srinivasan, Howard Zhou, Jonathan T Barron, Ricardo Martin-Brualla, Noah Snavely, and Thomas Funkhouser. Ibrnet: Learning multi-view image-based rendering. In Proceedings of the IEEE/CVF Conference on Computer Vision and Pattern Recognition, pp. 4690–4699, 2021.
>
> [6] Yao Yao, Zixin Luo, Shiwei Li, Tian Fang, and Long Quan. Mvsnet: Depth inference for unstructured multi-view stereo. In Proceedings of the European conference on computer vision (ECCV), pp. 767–783, 2018.

---

### Official Review · Reviewer_qLAL · 2023-10-31

**Soundness:** 3 good
**Presentation:** 3 good
**Contribution:** 2 fair
**Rating:** 5
**Confidence:** 4

**Summary:**

This work considers the problem of training a NeRF with sparse views. In this scenario, there are two issues: 1) the estimated cameras will cause errors; 2) the views are few and cannot support enough supervision. Taking these questions as consideration, this work presented a self-training strategy: it first train a NeRF directly, and then a consistency-based measurement is proposed to obtain ray reliability. Given reliability, the Nerf can be boosted using the reliable region to guide the training. The above procedure is conducted iteratively.

**Strengths:**

- The paper raises an interesting problem and an interesting strategy to boost NeRF training from sparse views.
- A new reliability estimation method is proposed, based on a consistency-based measurement.
- The experiments verifies the effectiveness of the proposed strategy where the artifacts are reduced.

**Weaknesses:**

- The major issues of NeRF with sparse views are two aspects: 1) few constraints for optimization; 2) wrong camera estimations. But, I don't see any designs for camera correction. Evenly the reliable regions are detected, but the cameras are still wrong. More discussions on this are needed.

- Another major concern is about the quality: although artifacts are reduced, the visual quality are still far from satisfactory, seeing from Fig 1, 4, and 5. I am wondering why we need such a self-training strategy. To me, sparse-view NeRF is a very challenging task, resorting to prior model might be a better way to involving self-training strategy.

**Questions:**

See weakness part.

---

> ### Author Response · Authors · 2023-11-17
> **Response to Reviewer qLAL (1/2)**
>
> **General reply:** Thank you for taking the time to thoroughly review our paper. We are glad that you found our work interesting and that the proposed strategy is effective in boosting NeRF training from sparse views. We have thoroughly examined your feedback and provided detailed responses below. Please inform us if any of our responses fail to address your concerns
>
> ### Weakness1. About designs for camera correction
>
> Thank you for sharing your concerns. We would like to first clarify that in the field of few-shot NeRF, there are two lines of research of training 1. NeRF with sparse views (DietNeRF[1], InfoNeRF[2], RegNeRF[3]) together **with accurate poses** and 2. NeRF with sparse views together with **noisy poses** (Sparf[4]). The current task we are tackling in this paper is the setting **where sparse images with accurate poses are given**. This can be achieved using external sensors or pre-calibrated fixed camera arrays to obtain accurate poses. We **follow prior works** (DietNeRF, InfoNeRF, RegNeRF) that have been developed under this setting and compare our works to these works in our paper. However, as you pointed out, NeRF’s application will be more practical when it becomes possible to optimize the cameras together and we believe that **our work can also be extended to these settings** by estimating the reliability with the inaccurately estimated pose information taken into account. We currently leave extending our work to pose-free or noisy poses as future work.
>
> [1] Ajay Jain, Matthew Tancik, and Pieter Abbeel. Putting nerf on a diet: Semantically consistent few-shot view synthesis. In Proceedings of the IEEE/CVF International Conference on Computer Vision, pp. 5885–5894, 2021.
>
> [2] Mijeong Kim, Seonguk Seo, and Bohyung Han. Infonerf: Ray entropy minimization for few-shot neural volume rendering. In *Proceedings of the IEEE/CVF Conference on Computer Vision and Pattern Recognition*, pp. 12912–12921, 2022.
>
> [3] Michael Niemeyer, Jonathan T Barron, Ben Mildenhall, Mehdi SM Sajjadi, Andreas Geiger, and Noha Radwan. Regnerf: Regularizing neural radiance fields for view synthesis from sparse inputs. In *Proceedings of the IEEE/CVF Conference on Computer Vision and Pattern Recognition*, pp. 5480–5490, 2022.
>
> [4] Prune Truong, Marie-Julie Rakotosaona, Fabian Manhardt, and Federico Tombari. Sparf: Neural radiance fields from sparse and noisy poses. In Proceedings of the IEEE/CVF Conference on Computer Vision and Pattern Recognition (CVPR), pp. 4190–4200, June 2023.

---

> ### Author Response · Authors · 2023-11-17
> **Response to Reviewer qLAL (2/2)**
>
> ### Weakness2. Quality & Resorting to prior models
>
> Thank you for your suggestions! We agree that training NeRF in sparse views is a challenging task. As you suggested, prior works using generative priors have been proposed, such as 2D generative prior (NeRDi[1]), 2D CLIP[2] prior (DietNeRF[3]), 2D normalizing flow model (RegNeRF[4]) and show the improvement in visual quality. However, we would like to emphasize that 1. using generative priors as additional guidance and 2. solely modeling the 3D scenes with only the given limited information are **orthogonal approaches**. Using an entirely different approach, compared to RegNeRF, we **achieve much higher results** of 4db and 7db in the NeRF Synthetic Extreme and the NeRF Synthetic dataset and achieve **competitive results** on the LLFF dataset. Our work aligns with the prior-free works adopting carefully designed regularization methods such as the entropy minimization used in InfoNeRF[5].
>
> In addition, we would like to point out that when resorting to prior models in a sparse view setting, the overall architecture needs to be carefully designed. In extreme settings such as sparse three views, these priors often act stronger than the rendering loss, guiding the network to **generate completely new objects that do not maintain the original geometry.** This can be seen in the ‘Lego’ figure generated by DietNeRF in Figure 4 of our paper, making the Lego object seem toy-like due to the prior knowledge of CLIP. Furthermore, we found that even with the incorporated priors, **existing works still suffer from overfitting to known views**, as shown in Figure 1. Our work is the first to tackle this fundamental overfitting problem by leveraging additional data rendered from the teacher NeRF and carefully designed reliability estimation methods. We show that we successfully take the **first step of resolving the overfitting problem from existing works** by showing a much gentle graph of PSNR as the views get further from known viewpoints and a **general improvement in performance** **entirely in a self-supervised manner** even when applied to different methods.
>
> Also, we agree that the adaptation of prior models can support our framework to synthesize images of higher quality, and as the improvements shown in the paper are solely from our ‘general’ plug-and-play manner framework, we believe that the incorporation of prior models has the potential of higher quality results.
>
> [1] Congyue Deng, Chiyu Jiang, Charles R Qi, Xinchen Yan, Yin Zhou, Leonidas Guibas, Dragomir Anguelov, et al. Nerdi: Single-view nerf synthesis with language-guided diffusion as general image priors. In Proceedings of the IEEE/CVF Conference on Computer Vision and Pattern Recognition, pp. 20637–20647, 2023.
>
> [2] Alec Radford, Jong Wook Kim, Chris Hallacy, Aditya Ramesh, Gabriel Goh, Sandhini Agarwal, Girish Sastry, Amanda Askell, Pamela Mishkin, Jack Clark, et al. Learning transferable visual models from natural language supervision. In *International conference on machine learning*, pp. 8748–8763. PMLR, 2021.
>
> [3] Ajay Jain, Matthew Tancik, and Pieter Abbeel. Putting nerf on a diet: Semantically consistent few-shot view synthesis. In Proceedings of the IEEE/CVF International Conference on Computer Vision, pp. 5885–5894, 2021
>
> [4] Michael Niemeyer, Jonathan T Barron, Ben Mildenhall, Mehdi SM Sajjadi, Andreas Geiger, and Noha Radwan. Regnerf: Regularizing neural radiance fields for view synthesis from sparse inputs. In *Proceedings of the IEEE/CVF Conference on Computer Vision and Pattern Recognition*, pp. 5480–5490, 2022.
>
> [5] Mijeong Kim, Seonguk Seo, and Bohyung Han. Infonerf: Ray entropy minimization for few-shot neural volume rendering. In *Proceedings of the IEEE/CVF Conference on Computer Vision and Pattern Recognition*, pp. 12912–12921, 2022.

---

### Official Review · Reviewer_qNfa · 2023-11-03

**Soundness:** 3 good
**Presentation:** 3 good
**Contribution:** 3 good
**Rating:** 6
**Confidence:** 5

**Summary:**

This paper tailors a self-training framework for NeRF reconstruction with few views, dubbed SE-NeRF. Technically, SE-NeRF iteratively optimizes NeRF parameters via reliability-aware pseudo-label supervision, which are generated by the teacher NeRF model obtained from the previous iteration. The reliability is computed by thresholding the feature similarity between epipolar correspondences. The reliable points are supervised directly using the teacher NeRF while the unreliable points are supervised by the Gaussian average over the adjacent rays.

**Strengths:**

+ The paper is clearly written and illustrated. The proposed method, which employs a self-training framework to maximize the utilization of training views, is not only intuitive but also well-motivated (Sec. 3.2).

+ The self-training framework has been carefully tailored to the NeRF setting. The approach to adaptively estimate ray reliability and the supervision scheme for both reliable and unreliable rays is elegant, straightforward, and efficient to implement.

+ The empirical improvements over the baseline methods are substantial (as shown in Tab. 2). The comprehensive ablation studies presented in Fig. 6 and Tab. 3 effectively establish the effectiveness of the iterative training and the two-fold supervision schemes employed in SE-NeRF. Furthermore, the appendix is well-prepared with necessary discussions and visualizations.

**Weaknesses:**

- A major concern arises from the apparent similarity to the existing work, Self-NeRF [1], which also employs a self-training framework in a few-shot setting, incorporating pseudo-label supervision and uncertainty estimation. Do authors acknowledge Self-NeRF as an independent concurrent work?

- More details need to be provided especially for unreliable ray distillation. What is the formulation for $\tilde{\sigma}$ and what is the definition of the Gaussian mask and the associated hyper-parameters, such as the kernel size etc.? See more in Question section.

[1] Bai et al., Self-NeRF: A Self-Training Pipeline for Few-Shot Neural Radiance Fields

**Questions:**

1. What is the reason behind SE-NeRF generally achieving greater improvements on vanilla NeRF compared to K-Planes (as indicated in Tab. 2)?

2. Would it be possible for the authors to provide results based on other NeRF representations, such as the more prevalent Instant-NGP [1] or Gaussian Splatting [2]?

3. While the concept of estimating unreliable rays via adjacent reliable rays is intuitive, it becomes apparent that an unreliable ray might not find a reliable ray within a local region, as illustrated in Fig. 15. How does the approach handle supervision for an unreliable ray when the neighboring reliable rays form an empty set?

4. Why is point-wise density only distilled rather than including color information? Can authors provide an ablation study on this technical design?


[1] Muller et al., Instant Neural Graphics Primitives with a Multiresolution Hash Encoding

[2] Kerbl et al., 3D Gaussian Splatting for Real-Time Radiance Field Rendering

---

> ### Author Response · Authors · 2023-11-17
> **Response to Reviewer qNfa (1/6)**
>
> **General reply:**
>
> Thank you for your thorough review of our paper. We are greatly encouraged by your assessment of our work as not only intuitive but also well-motivated, obtaining substantial improvement over baseline methods by employing a tailored self-training framework for the NeRF setting. We have carefully considered your questions and comments and have provided detailed responses below. If you have any further concerns or suggestions, please do not hesitate to let us know.
>
> ### Weakness1.  Similarity to the existing work
>
> Yes, we recognize Self-NeRF[1]  as a concurrent work to ours, with its **code not yet revealed.** Due to this reason, we could not give its direct comparison with our work in our main paper. If the code of Self-NeRF is released, we will include the comparison of our model to this work in the final version of our paper. Also, we would like to highlight the key differences between our framework and Self-NeRF in detail.
>
> > **Key Differences with Self-NeRF**
> >
> **1. In terms of Generalizability:**
>
> The authors of Self-NeRF explain their rationale for selecting Mip-NeRF [2] as their baseline model is to incorporate the power of Mip-NeRF’s cone-tracing to capture fine details of the scene and NeRF-W’s [3] uncertainty modeling by designing a new architecture of Mip-NeRF having a second branch to estimate the uncertainty value, uncertainty color, and density. In contrast, we propose an external method of estimating the reliability of rays mentioned in Section 4.2 of our paper, **which can be applied to existing methods in a plug-and-play manner** and show our framework’s versatility by leveraging different baselines(NeRF[4], K-Planes[5]).
>
> [1] Jiayang Bai, Letian Huang, Wen Gong, Jie Guo, and Yanwen Guo. Self-nerf: A self-training pipeline for few-shot neural radiance fields. arXiv preprint arXiv:2303.05775, 2023.
>
> [2] Jonathan T. Barron, Ben Mildenhall, Matthew Tancik, Peter Hedman, Ricardo Martin-Brualla, and Pratul P.Srinivasan. Mip-nerf: A multiscale representation for anti-aliasing neural radiance fields. In Proceedings of the IEEE/CVF International Conference on Computer Vision (ICCV), pp. 5855–5864, October 2021.
>
> [3] Ricardo Martin-Brualla, Noha Radwan, Mehdi SM Sajjadi, Jonathan T Barron, Alexey Dosovitskiy, and Daniel Duckworth. Nerf in the wild: Neural radiance fields for unconstrained photo collections. In Proceedings of the IEEE/CVF Conference on Computer Vision and Pattern Recognition, pp. 7210–7219, 2021.
>
> [4] Ben Mildenhall, Pratul P Srinivasan, Matthew Tancik, Jonathan T Barron, Ravi Ramamoorthi, and Ren Ng. Nerf: Representing scenes as neural radiance fields for view synthesis. Communications of the ACM, 65(1): 99–106, 2021.
>
> [5] Sara Fridovich-Keil, Giacomo Meanti, Frederik Rahbæk Warburg, Benjamin Recht, and Angjoo Kanazawa. K-planes: Explicit radiance fields in space, time, and appearance. In CVPR, 2023.

---

> ### Author Response · Authors · 2023-11-17
> **Response to Reviewer qNfa (2/6)**
>
> > **Key Differences with Self-NeRF (Continued)**
> >
> **2. Distillation Process:**
>
> We find that in order to guide the student network, the learned knowledge of reliable regions from the teacher network, **solely leveraging the photometric loss is not enough**. Why photometric loss is not enough can be understood as follows:
>
> 1. Although the supervision of the photometric loss of the reliable region rendered by the teacher NeRF to the student NeRF by the equation$\displaystyle \sum_{\mathbf{r} \in \mathcal{R}^+}
> M(\mathbf{r})
> \lVert
> {C}(\mathbf{r}; \theta^{\mathbb{T}})- {C}(\mathbf{r}; \theta)
> \rVert_2^2$  can guide the student NeRF to produce the same color estimation, the learned distribution of density can be different.
> 2. In other words, a unique photometric output **does not necessarily correspond one-to-one with a unique density distribution.**
> 3. Let us consider a straightforward experiment: volumetric rendering fundamentally calculates the final color through a weighted sum of the sampled point colors. Assuming the student predicts colors in sequence as (0.5, 0.5, 0), (1, 0, 0), (0, 1, 0) for a sample point count of 3, it becomes evident that two weight vectors, namely (1, 0, 0) and (0, 0.5, 0.5), yield the same final color (0.5, 0.5, 0).
> 4. Regarding NeRF, we **must consider a much higher degree of freedom** since NeRF samples hundreds of points. When distilled exclusively through the photometric loss, the student NeRF model still confronts considerable ambiguity in capturing geometric information.
> 5. As we intend to guide the student NeRF to **correctly model both the geometry and appearance** of reliable regions, we propose a novel reliability-based ray distillation loss and distill both the point-wise density loss and the overall appearance loss through the photometric loss.
>
> We present the effectiveness of our point-wise distillation method through ablations and show that this is **crucial to the overall performance gain.** We additionally show the quantitative results when only the photometric loss to reliable regions is used under the NeRF Synthetic Extreme setting.
>
> | $\text{Methods}$ | $\text{Average}$ | $\text{chair}$ | $\text {drums}$ | $\text{ficus}$ | $\text{hotdog}$ | $\text{lego}$ | $\text{mater.}$ | $\text{ship}$ | $\text{mic}$ |
> | --- | --- | --- | --- | --- | --- | --- | --- | --- | --- |
> | $\text{K-Planes}$ | $15.45$ | $15.61$ | $13.23$ | $18.29$ | $12.45$ | $14.67$ | $16.30$ | $13.35$ | $19.74$ |
> | $\text{SE-NeRF (K-Planes) photometric}$ | $16.89$$\scriptsize\color{green}(+1.44)$ | $19.79$ $\scriptsize\color{green}(+4.18)$ | $12.97$ $\scriptsize\color{red}(-0.26)$ | $18.04$ $\scriptsize\color{red}(-0.25)$ | $18.92$ $\scriptsize\color{green}(+6.47)$ | $16.33$ $\scriptsize\color{green}(+1.66)$ | $16.22$ $\scriptsize\color{red}(-0.08)$ | $12.96$ $\scriptsize\color{red}(-0.39)$ | $19.94$ $\scriptsize\color{green}(+0.20)$ |
> | $\text{SE-NeRF (K-Planes)}$ | $17.49$ $\scriptsize\color{green}(+2.04)$ | $20.54$ $\scriptsize\color{green}(+4.93)$ | $13.38$ $\scriptsize\color{green}(+0.15)$ | $18.33$ $\scriptsize\color{green}(+0.04)$ | $20.14$ $\scriptsize\color{green}(+7.69)$ | $16.65$ $\scriptsize\color{green}(+1.98)$ | $17.01$ $\scriptsize\color{green}(+0.71)$ | $13.72$ $\scriptsize\color{green}(+0.37)$ | $20.13$ $\scriptsize\color{green}(+0.39)$ |
>
> | $\text{Methods}$ | $\text{Average}$ | $\text{chair}$ | $\text {drums}$ | $\text{ficus}$ | $\text{hotdog}$ | $\text{lego}$ | $\text{mater.}$ | $\text{ship}$ | $\text{mic}$ |
> | --- | --- | --- | --- | --- | --- | --- | --- | --- | --- |
> | $\text{Instant-NGP}$ | $15.67$ | $17.66$ | $12.75$ | $18.44$ | $13.67$ | $13.17$ | $16.83$ | $13.82$ | $19.05$ |
> | $\text{SE-NeRF (Instant-NGP) photometric}$ | $17.16$ $\scriptsize\color{green}(+1.49)$ | $20.86$ $\scriptsize\color{green}(+3.20)$ | $13.18$ $\scriptsize\color{green}(+0.43)$ | $19.05$ $\scriptsize\color{green}(+0.61)$ | $18.00$ $\scriptsize\color{green}(+4.33)$ | $15.51$ $\scriptsize\color{green}(+2.34)$ | $17.26$ $\scriptsize\color{green}(+0.43)$ | $14.01$ $\scriptsize\color{green}(+0.19)$ | $19.38$ $\scriptsize\color{green}(+0.33)$ |
> | $\text{SE-NeRF (Instant-NGP)}$ | $17.47$ $\scriptsize\color{green}(+1.80)$ | $20.40$ $\scriptsize\color{green}(+2.74)$ | $13.34$ $\scriptsize\color{green}(+0.59)$ | $19.07$ $\scriptsize\color{green}(+0.63)$ | $18.15$ $\scriptsize\color{green}(+4.48)$ | $15.99$ $\scriptsize\color{green}(+2.82)$ | $17.94$ $\scriptsize\color{green}(+1.11)$ | $14.61$ $\scriptsize\color{green}(+0.79)$ | $20.23$ $\scriptsize\color{green}(+1.18)$ |

---

> ### Author Response · Authors · 2023-11-17
> **Response to Reviewer qNfa (3/6)**
>
> > **Key Differences with Self-NeRF (Continued)**
> >
> **3. Feature-based reliability estimation:**
>
> To estimate the reliability of the rendered rays, we leverage the pixel-level feature by **upsampling the feature map extracted from different levels** of a pre-trained VGG-19[1] model. In contrast, Self-NeRF[2] uses warped pseudo views as additional data to guide the network in unknown views. However, prior works have found that **warped-based supervision can easily fail in reflectant non-Lambertian surfaces** where appearance changes greatly in different viewpoints[3]. In order to estimate reliability even in reflectant non-Lambertian surfaces, we designed our reliability estimation module to measure the feature similarity of the projected points to compare the structural consistency and show that our method successfully estimated reliable regions compared to the ground-truth masks. Additional results and comparisons are shown in Section B.2 and Section D in the supplementary materials.
>
> [1] Karen Simonyan and Andrew Zisserman. Very deep convolutional networks for large-scale image recognition. arXiv preprint arXiv:1409.1556, 2014.
>
> [2] Jiayang Bai, Letian Huang, Wen Gong, Jie Guo, and Yanwen Guo. Self-nerf: A self-training pipeline for few-shot neural radiance fields. arXiv preprint arXiv:2303.05775, 2023.
>
> [3] Huangying Zhan, Ravi Garg, Chamara Saroj Weerasekera, Kejie Li, Harsh Agarwal, and Ian Reid. Unsupervised learning of monocular depth estimation and visual odometry with deep feature reconstruction. In Proceedings of the IEEE conference on computer vision and pattern recognition, pp. 340–349, 2018.

---

> ### Author Response · Authors · 2023-11-17
> **Response to Reviewer qNfa (4/6)**
>
> ### Weakness2. More details for unreliable ray distillation
>
> We apologize for the insufficient explanation in the paper and explain in detail how we apply a Gaussian mask to each unreliable ray. We will add a specific explanation of how the unreliable ray is distilled and upload the revised version of our paper.
>
> Given teacher-generated image-size ray weights $\sigma$ of size $(W,H,S)$, where $W$ is the image width, $H$ is the image height, $S$ is the number of samples on each ray, our goal is to generate Gaussian weighted ray weights $\tilde\sigma$.
>
> Here, we use $N \times N$ discretely approximated version of 2D isotropic Gaussian distribution with mean $(c_x, c_y)$ and standard deviation 1, whose probability mass function is $g(x,y, c_x, c_y)$ where the original probability density function is expressed as $G(x,y, c_x, c_y) =
> \frac{1}{\sqrt{2\pi}}
> e^{-\frac{(x-c_x)^2+(y-c_y)^2}{2}}$
>
> Then given a set of points on reliable ray $\mathbb{R}$, each weight $\tilde\sigma(i,j,w)$ is calculated as:
>
> $$
> \tilde\sigma(i,j,w) =
> \sum_{x \in \Omega_x}
> \sum_{y \in \Omega_y}
> \frac{I_\mathbb{R}(x,y,w) \cdot g(x,y,i,j)}{
> \sum_{x \in \Omega_x}
> \sum_{y \in \Omega_y}
> I_\mathbb{R}(x,y,w) \cdot g(x,y,i,j)} \sigma(x,y,w)
> $$
>
> where $I_\mathbb{R}$ is indicator function, $\Omega_x =
> [i - \lfloor \frac{N}{2} \rfloor,
> i + \lfloor \frac{N}{2} \rfloor ] \backslash \\{i\\}$ and $\Omega_y =
> [j -\lfloor \frac{N}{2} \rfloor,
> j + \lfloor \frac{N}{2} \rfloor ] \backslash \\{j\\}$.
>
> The generated Gaussian weighted ray weights $\tilde\sigma$ are then distilled to the student NeRF in the next iteration.
>
> We adopt a Gaussian mask with a kernel size of 3 as the default unreliable ray distillation method in our framework. The technical design for this choice is that although incorporating the depth smoothness prior is intuitive, we find that this assumption can be too strong in cases when modeling high-frequency regions. As we directly distill the point-wise weights of points sampled from the ray, **we must apply the depth-smoothness prior only in highly adjacent regions**. To this end, we have tested using different sizes of kernels of 3 and 5 in the NeRF Synthetic Extreme setting. Although the overall performance boost does not show a big difference, we have noticed that scenes that contain high-frequency regions such as ‘ficus’ of the NeRF Synthetic dataset show suboptimal results. To allow our framework to work robustly in extreme settings, we have chosen three as the default kernel size for unreliable ray distillation.
>
> | $\text{Methods}$ | $\text{chair}$ | $\text {drums}$ | $\text{ficus}$ | $\text{hotdog}$ | $\text{lego}$ | $\text{mater.}$ | $\text{ship}$ | $\text{mic}$ |
> | --- | --- | --- | --- | --- | --- | --- | --- | --- |
> | $\text{SE-NeRF (kernel 3)}$ | $\text{20.54} \scriptsize\color{green}(+0.27)$  | $\text{13.38} \scriptsize\color{green}(+0.09)$ | $\text{18.33} \scriptsize\color{green}(+0.30)$ | $\text{20.14} \scriptsize\color{red}(-0.19)$ | $\text{16.65} \scriptsize\color{green}(+0.05)$ | $\text{17.01} \scriptsize\color{green}(+0.13)$ | $\text{13.72} \scriptsize\color{red}(-0.14)$ | $\text{20.13} \scriptsize\color{red}(-0.34)$ |
> | $\text{SE-NeRF (kernel 5)}$ | $\text{20.27} \scriptsize\color{red}(-0.27)$ | $\text{13.29} \scriptsize\color{red}(-0.09)$ | $18.03 \scriptsize\color{red}(-0.30)$ | $20.33 \scriptsize\color{green}(+0.19)$ | $16.60 \scriptsize\color{red}(-0.05)$ | $16.88 \scriptsize\color{red}(-0.13)$ | $13.86 \scriptsize\color{green}(+0.14)$ | $20.47 \scriptsize\color{green}(+0.34)$ |

---

> ### Author Response · Authors · 2023-11-17
> **Response to Reviewer qNfa (5/6)**
>
> ### Q1. Greater improvements on vanilla NeRF compared to $K$-Planes
>
> Though our framework is a general one that can be applied to any NeRF model, it can not be ignored that **the distillation process is inevitably influenced by the baseline model and its own training strategy.** Our analysis of vanilla NeRF[1] achieving greater improvements is that in the case of the $K$-Planes[2] baseline, the model converges with much fewer steps than NeRF, leading to fewer steps of the distillation process from the teacher to the student which results in different degrees of improvement. However, we show that our framework can be **generally applied to different existing models and enhance the performance across different scenes.**
>
> ### Q2. SE-NeRF on other NeRF representations
>
> Thank you for your suggestions. Our framework can fundamentally be applied to **all ray-based NeRFs** that learn the scene using the volume rendering equation. To demonstrate this, **we have applied our framework to the Instant-NGP[3] you suggested.** We train SE-NeRF on the Instant-NGP for 5 minutes for each iteration on the NeRF-Synthetic Extreme setting using a single RTX 3090 GPU. Below is the quantitative comparison per scene using our framework.
>
> | $\text{Methods}$ | $\text{chair}$ | $\text {drums}$ | $\text{ficus}$ | $\text{hotdog}$ | $\text{lego}$ | $\text{mater.}$ | $\text{ship}$ | $\text{mic}$ |
> | --- | --- | --- | --- | --- | --- | --- | --- | --- |
> | $\text{NeRF}$ | $15.08$ | $11.98$ | $17.16$ | $13.83$ | $16.31$ | $17.31$ | $10.84$ | $16.29$ |
> | $\text{K-Planes}$ | $15.61$ | $13.23$ | $18.29$ | $12.45$ | $14.67$ | $16.30$ | $13.35$ | $19.74$ |
> | $\textbf{Instant-NGP}$ | $17.66$ | $12.75$ | $18.44$ | $13.67$ | $13.17$ | $16.83$ | $13.82$ | $19.05$ |
> | $\text{DietNeRF}$ | $16.60$ | $8.09$ | $18.32$ | $19.00$ | $11.45$ | $16.97$ | $15.26$ | $10.01$ |
> | $\text{InfoNeRF}$ | $15.38$ | $12.48$ | $18.59$ | $19.04$ | $12.27$ | $15.25$ | $7.23$ | $16.76$ |
> | $\text{RegNeRF}$ | $15.92$ | $12.09$ | $14.83$ | $14.06$ | $14.86$ | $10.53$ | $11.44$ | $16.12$ |
> | $\text{SE-NeRF (NeRF)}$ | $19.96$ $\scriptsize\color{green}(+4.88)$ | $14.72$ $\scriptsize\color{green}(+2.74)$ | $19.29$ $\scriptsize\color{green}(+2.13)$ | $16.06$ $\scriptsize\color{green}(+2.23)$ | $16.45$ $\scriptsize\color{green}(+0.14)$ | $17.51$ $\scriptsize\color{green}(+0.20)$ | $14.20$ $\scriptsize\color{green}(+3.36)$ | $21.09$ $\scriptsize\color{green}(+4.80)$ |
> | $\text{SE-NeRF (K-Planes)}$ | $20.54$ $\scriptsize\color{green}(+4.93)$ | $13.38$ $\scriptsize\color{green}(+0.15)$ | $18.33$ $\scriptsize\color{green}(+0.04)$ | $20.14$ $\scriptsize\color{green}(+7.69)$ | $16.65$ $\scriptsize\color{green}(+1.98)$ | $17.01$ $\scriptsize\color{green}(+0.71)$ | $13.72$ $\scriptsize\color{green}(+0.37)$ | $20.13$ $\scriptsize\color{green}(+0.39)$ |
> | $\textbf{SE-NeRF (Instant-NGP)}$ | $20.40$ $\scriptsize\color{green}(+2.74)$ | $13.34$ $\scriptsize\color{green}(+0.59)$ | $19.07$ $\scriptsize\color{green}(+0.63)$ | $18.15$ $\scriptsize\color{green}(+4.48)$ | $15.99$ $\scriptsize\color{green}(+2.82)$ | $17.94$ $\scriptsize\color{green}(+1.11)$ | $14.61$ $\scriptsize\color{green}(+0.79)$ | $20.23$ $\scriptsize\color{green}(+1.18)$ |
>
> Additionally, as the Gaussian Splatting[4] you have mentioned is not a ray-based volumetric rendering approach, and thus the reliability-based ray distillation of our framework, which we tend to be crucial for the performance boost, cannot be applied. However, theoretically, if we can estimate the reliability of the Gaussian splats predicted by the model, we can distill the learned information of Gaussian splats, such as the mean and variance to the student network as additional pseudo labels. We are considering applying our framework to newly proposed methods of 3D reconstruction as future work.
>
> [1] Ben Mildenhall, Pratul P Srinivasan, Matthew Tancik, Jonathan T Barron, Ravi Ramamoorthi, and Ren Ng. Nerf: Representing scenes as neural radiance fields for view synthesis. Communications of the ACM, 65(1): 99–106, 2021.
>
> [2] Sara Fridovich-Keil, Giacomo Meanti, Frederik Rahbæk Warburg, Benjamin Recht, and Angjoo Kanazawa. K-planes: Explicit radiance fields in space, time, and appearance. In CVPR, 2023.
>
> [3] Thomas Müller, Alex Evans, Christoph Schied, and Alexander Keller. Instant neural graphics primitives with a multiresolution hash encoding. ACM Transactions on Graphics (ToG), 41(4):1–15, 2022.
>
> [4] Bernhard Kerbl, Georgios Kopanas, Thomas Leimkühler, and George Drettakis. 3d gaussian splatting for real-time radiance field rendering. ACM Transactions on Graphics (ToG), 42(4):1–14, 2023.

---

> ### Author Response · Authors · 2023-11-17
> **Response to Reviewer qNfa (6/6)**
>
> ### Q3. Handling unreliable rays
>
> Thank you for pointing this out. For cases when the neighboring reliable rays form an empty set, we **do not include the specific unreliable ray in the distillation process**, as regularization from reliable rays is not possible. This is important from the perspective of semi-supervised training, as the incorporation of unreliable rays can **lead to the confirmation bias problem [1].** This is also shown in our experiments of using the unified equation to distill the knowledge of the teacher to the student, which is explained in Section B.3 of the supplementary materials. The unified equation can be understood as applying a soft-thresholding method to the similarity values we have estimated. In this setting, we distill the teacher's knowledge to the student scaled by the similarity value that has been filtered using a sigmoid-like function. Also, we mimic the regularization of unreliable rays by regularizing rays with adjacent rays that have higher values of similarity. However, we found that the more the performance is enhanced as the sigmoid-like function becomes more similar to a step-function (which will make the whole process identical to hard-thresholding). These findings support our design choice of ignoring unreliable rays with no adjacent reliable rays in the distillation process. Please refer to Section B.3 in the supplementary materials for a more detailed explanation.
>
> ### Q4. Point-wise color distillation
>
> Thank you for pointing out this technical design of the distillation process. As you have pointed out, it is indeed possible to distill both color and density from points along a ray. However, the **ambiguity in the process of rendering the final color may hinder the student network from learning effectively if we directly distill the color of the point**. Specifically, as the final color is calculated with the weighted sum of the colors along the ray, some points on a reliable ray that carries small or no weights may not have an accurate color as they do not contribute to the final rendering, leading to small or none supervision from the photometric loss. While an increase in the number of given views might mitigate this issue, it is important to note that in a few-shot setting, particularly during the initial stages of learning, this problem could significantly arise. Therefore, if we distill such point-wise color directly, it could distill incorrect information and potentially disrupt the learning of the student network. We apologize for not providing enough explanation for the technical design of the distillation process.
>
> [1] Eric Arazo, Diego Ortego, Paul Albert, Noel E O’Connor, and Kevin McGuinness. Pseudo-labeling and confirmation bias in deep semi-supervised learning. In *2020 International Joint Conference on Neural Networks (IJCNN)*, pp. 1–8. IEEE, 2020.

---

### Author Response · Authors · 2023-11-17
**General Response**

We would first like to express our gratitude for the insightful reviews and helpful suggestions provided by the reviewers. We are greatly encouraged by their assessment of our self-training-based framework as **innovative** (qLAL, qP3j), with **significant and original** (qP3j) developments, and **well-motivated** (qNfa) with **clear analysis and derivation** (qP3j, qNfa). We are also thankful for their judgment of the **quality** of our work as **high** (qP3j) with a **clear illustration of methodology** (qNfa, qP3j), with its methodology being **elegant**, **straightforward**, and **efficient** to implement (qNfa), as well as verification of our experimental results and clearly **demonstrating effectiveness** (qLAL) and inducing **substantial improvement** (qNfa, qP3j) over previous baselines, achieving **state-of-the-art** (qP3j).

---

### Author Response · Authors · 2023-11-17
**Additional Results on applying our Self-Evolving Neural Radiance Fields framework**

In response to the suggestions from the reviewers, we have conducted several additional ablations. In particular, following the suggestion from reviewer qNfa, **we have applied our framework to an additional model, Instant-NGP [1], and summarized the results.** We show that similar to the baseline models reported in our paper, the vanilla NeRF [2] and $K$-planes [3], applying Instant-NGP as a baseline model also resulted in general performance improvements across all scenes. **We are pleased to report that these additional experiments have further validated the results of our framework and show that our framework can be applied to existing methods in a plug-and-play manner.** Below are the results of applying SE-NeRF to Instant-NGP under the NeRF-Synthetic Extreme setting we have proposed in our paper. We train the model for 5 minutes for each iteration using a single RTX 3090 GPU.

| $\text{Methods}$ | $\text{chair}$ | $\text {drums}$ | $\text{ficus}$ | $\text{hotdog}$ | $\text{lego}$ | $\text{mater.}$ | $\text{ship}$ | $\text{mic}$ |
| --- | --- | --- | --- | --- | --- | --- | --- | --- |
| $\text{NeRF}$ | $15.08$ | $11.98$ | $17.16$ | $13.83$ | $16.31$ | $17.31$ | $10.84$ | $16.29$ |
| $\text{K-Planes}$ | $15.61$ | $13.23$ | $18.29$ | $12.45$ | $14.67$ | $16.30$ | $13.35$ | $19.74$ |
| $\textbf{Instant-NGP}$ | $17.66$ | $12.75$ | $18.44$ | $13.67$ | $13.17$ | $16.83$ | $13.82$ | $19.05$ |
| $\text{DietNeRF}$ | $16.60$ | $8.09$ | $18.32$ | $19.00$ | $11.45$ | $16.97$ | $15.26$ | $10.01$ |
| $\text{InfoNeRF}$ | $15.38$ | $12.48$ | $18.59$ | $19.04$ | $12.27$ | $15.25$ | $7.23$ | $16.76$ |
| $\text{RegNeRF}$ | $15.92$ | $12.09$ | $14.83$ | $14.06$ | $14.86$ | $10.53$ | $11.44$ | $16.12$ |
| $\text{SE-NeRF (NeRF)}$ | $19.96$ $\scriptsize\color{green}(+4.88)$ | $14.72$ $\scriptsize\color{green}(+2.74)$ | $19.29$ $\scriptsize\color{green}(+2.13)$ | $16.06$ $\scriptsize\color{green}(+2.23)$ | $16.45$ $\scriptsize\color{green}(+0.14)$ | $17.51$ $\scriptsize\color{green}(+0.20)$ | $14.20$ $\scriptsize\color{green}(+3.36)$ | $21.09$ $\scriptsize\color{green}(+4.80)$ |
| $\text{SE-NeRF (K-Planes)}$ | $20.54$ $\scriptsize\color{green}(+4.93)$ | $13.38$ $\scriptsize\color{green}(+0.15)$ | $18.33$ $\scriptsize\color{green}(+0.04)$ | $20.14$ $\scriptsize\color{green}(+7.69)$ | $16.65$ $\scriptsize\color{green}(+1.98)$ | $17.01$ $\scriptsize\color{green}(+0.71)$ | $13.72$ $\scriptsize\color{green}(+0.37)$ | $20.13$ $\scriptsize\color{green}(+0.39)$ |
| $\textbf{SE-NeRF (Instant-NGP)}$ | $20.40$ $\scriptsize\color{green}(+2.74)$ | $13.34$ $\scriptsize\color{green}(+0.59)$ | $19.07$ $\scriptsize\color{green}(+0.63)$ | $18.15$ $\scriptsize\color{green}(+4.48)$ | $15.99$ $\scriptsize\color{green}(+2.82)$ | $17.94$ $\scriptsize\color{green}(+1.11)$ | $14.61$ $\scriptsize\color{green}(+0.79)$ | $20.23$ $\scriptsize\color{green}(+1.18)$ |

[1] Thomas Müller, Alex Evans, Christoph Schied, and Alexander Keller. Instant neural graphics primitives with a multiresolution hash encoding. ACM Transactions on Graphics (ToG), 41(4):1–15, 2022.

[2] Ben Mildenhall, Pratul P Srinivasan, Matthew Tancik, Jonathan T Barron, Ravi Ramamoorthi, and Ren Ng. Nerf: Representing scenes as neural radiance fields for view synthesis. Communications of the ACM, 65(1): 99–106, 2021.

[3] Sara Fridovich-Keil, Giacomo Meanti, Frederik Rahbæk Warburg, Benjamin Recht, and Angjoo Kanazawa. K-planes: Explicit radiance fields in space, time, and appearance. In CVPR, 2023.

---

### Author Response · Authors · 2023-11-20
**Kind reminder for further discussion.**

Dear reviewers,
We thank all the reviewers again for spending their time thoroughly reviewing our manuscript and showing interest in our work. We have made additional analysis, including our experiment on applying our framework to a new existing method, and added specific explanations on terms that might have been ambiguous or not specific enough in our initial manuscript to address all the raised concerns or questions. We believe we have done our best to answer all the questions and weaknesses shown by the reviews, and we are eager to address any further concerns or questions and engage in an active discussion. As the period for discussion ends soon, we would really appreciate it if the reviewers could take a look at our responses.
Thank you for your time and interest in our work! We look forward to hearing back and will gladly provide further clarifications.

Best regards.

---

### Author Response · Authors · 2023-11-22
**We have uploaded a revised version of our manuscript**

Dear reviewers,

We have thoroughly reviewed all your reviews, and we are sorry for any ambiguity or confusion caused by our initial manuscript. We have revised our manuscript to resolve all the concerns and expressed all the changed parts in red, so we would really appreciate it if you could take your time reviewing our updated manuscript. Specifically, we have

- added a specific and comprehensive explanation and experiments of **how unreliable rays are distilled**, as suggested by reviewers qNfa and qP3j. This can be found in **Section B.3 of the supplementary materials.**
- added comprehensive explanation and experiments about **selecting feature extraction** used in our framework as suggested by reviewers qNfa and qP3j, which can be found in **Section B.1 of the supplementary materials.**
- added results on **applying our framework to another NeRF representation** (Instant-NGP [1]) as suggested by reviewer qNfa. This can be found in **Table 2**.
- fixed any **unclear or ambiguous expressions or explanations and mistaken values** in our table.

We appreciate all the reviewers spending their time to review our work! As the time for discussion ends soon, we would really appreciate it if you could express if we have clearly addressed your concerns.

Best regards,

The authors of paper 953.

[1] Thomas Müller, Alex Evans, Christoph Schied, and Alexander Keller. Instant neural graphics primitives with a multiresolution hash encoding. ACM Transactions on Graphics (ToG), 41(4):1–15, 2022.

---

### Meta-Review · Area_Chair_7Z5T · 2023-12-07

**Metareview:**

The submission received mostly slightly negative reviews. The reviewers' main concerns were on similarity to prior works that were not discussed and/or compared against, clarity and missing ablation on ray distillation, and absence of camera correction components. The reviewers had not provided additional feedback after the authors posted the response. The AC carefully read through the paper, the reviewers' comments, and the authors' rebuttal. While the AC agrees with the authors that the paper focuses on the setup of accurate camera poses, the AC agrees with the reviewers that sufficient discussions on prior methods (e.g. Self-NeRF), as well as experiments comparing against important prior methods with learned priors (pixelNeRF, MVSNeRF etc.), are necessary to be included in the paper. As such, the AC recommends rejection.

**Justification For Why Not Higher Score:**

The reviews are inclined towards rejection, with the most critical part being insufficient comparisons against important baseline methods.

**Justification For Why Not Lower Score:**

N/A

---

### Decision · Program_Chairs · 2024-01-16

Reject